# The aging factor EPS8 induces disease-related protein aggregation through RAC signaling hyperactivation

Seda Koyuncu [1,2] ✉, Yaiza Dominguez-Canterla[3,4], Rafael Alis[3,4], Nassima Salarzai[1,2], Dunja Petrovic[1,2], Nuria Flames[3,4] & David Vilchez [1,2,5,6] ✉

Aging is a major risk factor for neurodegenerative diseases associated with protein aggregation, including Huntington's disease and amyotrophic lateral sclerosis (ALS). Although these diseases involve different aggregation-prone proteins, their common late onset suggests a link to converging changes resulting from aging. In this study, we found that age-associated hyperactivation of EPS8/RAC signaling in *Caenorhabditis elegans* promotes the pathological aggregation of Huntington's disease-related polyglutamine repeats and ALS-associated mutant FUS and TDP-43 variants. Conversely, knockdown of *eps-8* or *RAC* orthologs prevents protein aggregation and subsequent deficits in neuronal function during aging. Similarly, inhibiting EPS8 signaling reduces protein aggregation and neurodegeneration in human cell models. We further identify the deubiquitinating enzyme USP4 as a regulator of EPS8 ubiquitination and degradation in both worms and human cells. Notably, reducing USP-4 upregulation during aging prevents EPS-8 accumulation, extends longevity and attenuates disease-related changes. Our findings suggest that targeting EPS8 and its regulatory mechanisms could provide therapeutic strategies for age-related diseases.

Aging is a primary risk factor for distinct neurodegenerative diseases that remain incurable, including Alzheimer's, Parkinson's, Huntington's and amyotrophic lateral sclerosis (ALS)[1,2]. Familial cases often arise during the fifth decade of life, whereas sporadic cases typically occur within the seventh decade or later[3]. For instance, ALS is rare before the age of 40 years, but its incidence increases exponentially thereafter[4].

In addition to their late onset, a common feature of these neurodegenerative diseases is the accumulation of pathological protein aggregates[3]. However, the specific proteins that aggregate differ across diseases. For instance, Huntington's disease results from mutations in the huntingtin (*HTT*) gene, leading to an expanded polyglutamine (polyQ) repeat that is prone to aggregation. In patients, HTT protein

contains more than 35 polyQ repeats and forms pathological aggregates[5]. Most ALS cases (90%) are sporadic with unknown etiology, whereas the remaining cases are linked to familial mutations in one of over 30 different genes[6]. Among them, TDP-43 and FUS mutations are particularly prevalent, leading to their cytosolic aggregation[6]. Moreover, TDP-43 and FUS also frequently form aggregates in sporadic ALS[6].

Together, the late onset and heterogeneity of protein aggregates indicate that these diseases are linked to converging cellular changes resulting from aging. Indeed, loss of protein homeostasis (proteostasis) is an evolutionary conserved hallmark of aging[2,7]. Thus, defining pathways that delay aging and subsequent protein aggregation could provide therapeutic targets for preventing neurodegenerative diseases.

[1]Institute for Integrated Stress Response Signaling, Faculty of Medicine, University Hospital Cologne, Cologne, Germany. [2]Cologne Excellence Cluster for Cellular Stress Responses in Aging-Associated Diseases (CECAD), University of Cologne, Cologne, Germany. [3]Developmental Neurobiology Unit, Instituto de Biomedicina de Valencia IBV-CSIC, Valencia, Spain. [4]Valencia Biomedical Research Foundation, Centro de Investigación Príncipe Felipe (CIPF), Associated Unit to the Instituto de Biomedicina de Valencia (IBV), Valencia, Spain. [5]Institute for Genetics, University of Cologne, Cologne, Germany. [6]Center for Molecular Medicine Cologne (CMMC), University of Cologne, Cologne, Germany. ✉e-mail: skoyunc2@uni-koeln.de; dvilchez@uni-koeln.de

Along these lines, longevity mechanisms such as reduced insulin/IGF-1 signaling, dietary restriction and cold temperature delay pathological protein aggregation in model organisms[8–10].

With age, animals undergo alterations in proteolytic systems, including the ubiquitin–proteasome system[2,11–16]. Because aggregation-prone proteins such as mutant HTT and TDP-43 can be ubiquitinated, extensive research focuses on how ubiquitinating and deubiquitinating enzymes directly influence their proteasomal degradation[17,18]. In this study, we explored a different approach to define common mechanisms that prevent pathological aggregation across distinct disorders. Beyond directly influencing the levels of disease-related proteins, age-related downregulation of targeted degradation also leads to the accumulation of regulatory proteins, affecting pathways required for normal cell function[2,11–16]. An intriguing question is whether the accumulation of regulatory proteins that escape proteasomal clearance contributes to disease-related protein aggregation during aging.

In *C. elegans*, aging leads to a loss of ubiquitination in EPS-8 protein[14]. Subsequently, EPS-8 cannot be degraded by the proteasome and accumulates with age[14]. Although EPS-8 endows benefits early in life[19,20], its upregulation during aging is detrimental for adult lifespan[14,21]. EPS-8 induces the exchange of GDP for GTP on RAC protein, which then becomes active[22]. The accumulation of EPS-8 hyperactivates RAC signaling across tissues during aging, altering downstream mechanisms. For instance, hyperactivated EPS-8/RAC signaling induces excessive actin polymerization and subsequent destabilization of the actin cytoskeleton with age[14]. Moreover, EPS-8/RAC hyperactivates protein kinase JNK, shortening lifespan[14]. Conversely, knockdown of EPS-8 prevents these age-related changes and extends longevity in *C. elegans*[14,21]. Likewise, knockout of *Eps8* in mice also extends lifespan[23].

In the present study, we found that reducing EPS-8/RAC signaling attenuates pathological protein aggregation in *C. elegans* models of Huntington's disease and ALS during aging, preventing subsequent deficits in neuronal function. Moreover, we discovered that the deubiquitinating enzyme (DUB) USP-4 promotes EPS-8 deubiquitination and accumulation during aging. Conversely, knockdown of *usp-4* after development extends lifespan and prevents disease-related changes. In addition, we found that the USP4/EPS8/RAC pathway also influences disease-related aggregation and neurodegeneration in human cell models. Because the effects of USP4 and EPS8 are evolutionary conserved, our results can have implications for disease prevention.

## Results

### Lowering EPS-8/RAC signaling decreases pathological protein aggregation in *C. elegans*

With age, elevated EPS-8 levels hyperactivate RAC signaling, thereby contributing to organismal aging in *C. elegans*[14]. Thus, we investigated whether preventing RAC hyperactivation can attenuate disease-related protein aggregation. To this end, we used worm models expressing expanded polyQ repeats throughout the nervous system. These worms recapitulate pathological phenotypes of Huntington's disease, including protein aggregation, with a pathogenic threshold at polyQ40 repeats[24]. Accordingly, worms expressing polyQ67 in neurons are a well-established model for disease-related protein aggregation, which can be quantified using a filter trap assay (Fig. 1a)[9,25–27].

We found that post-developmental knockdown of *eps-8* prevents polyQ67 aggregation in the neurons of day 5 adult worms, without decreasing the levels of polyQ67 peptides (Fig. 1a). Likewise, loss-of-function *eps-8* mutants also exhibited lower levels of polyQ67 aggregates (Extended Data Fig. 1a). To further validate our filter trap results, we used a western blot approach that can detect both polyQ monomers and SDS-insoluble polyQ aggregates retained at the top of the gel[28]. Using this method, we confirmed that *eps-8* knockdown reduces insoluble polyQ67 levels (Extended Data Fig. 1b).

Similarly, loss of RAC orthologs (*mig-2* and *rac-2*) also reduces polyQ67 aggregation without decreasing polyQ67 levels (Fig. 1a). To assess whether *mig-2* and *rac-2* have redundant effects on polyQ aggregation, we applied diluted RNA interference (RNAi) treatments. We observed that the combination of diluted RNAi against *mig-2* and *rac-2* further decreases polyQ67 aggregation compared with diluted *rac-2* alone, suggesting that both RAC orthologs have at least partially redundant effects on polyQ aggregation (Extended Data Fig. 1c).

Similar to EPS-8, the intermediate filament IFB-2 is a proteasome target that undergoes reduced ubiquitination and degradation with age[14]. Although IFB-2 knockdown during adulthood also extends lifespan[14], it did not reduce polyQ aggregation in any of the tissues tested (Extended Data Fig. 1d–f). These results indicate a specific role for the age-dysregulated proteasome target EPS-8 in pathological protein aggregation. We then asked whether hyperactivated EPS-8/RAC signaling promotes polyQ67 aggregation through its intracellular activity within neurons or via cell non-autonomous mechanisms. We found that neuronal knockdown of *eps-8* or *RAC* orthologs reduces polyQ67 aggregation in neurons (Extended Data Fig. 1g). Although these results suggest that elevated RAC signaling has intracellular effects on polyQ aggregation, the involvement of cell non-autonomous pathways cannot be ruled out.

The accumulation of polyQ aggregates in *C. elegans* neurons impairs neuronal function[24,29]. The most studied phenotype is loss of motility, which correlates with aggregate levels and age[9,24,25,30]. Indeed, polyQ67 worms exhibited a decline in motility compared with control polyQ19 worms at day 5 of adulthood but not at day 1 (Extended Data Fig. 2a). Although knockdown of either *eps-8* or *RAC* orthologs had no effect in young worms, it reduced motility deficits in aged polyQ67 worms (Extended Data Fig. 2a). Previously, we found that lowering EPS-8/RAC signaling not only has effects in neurons but also delays age-related muscle dysfunction in wild-type (WT) animals, preventing motility decline during aging[14]. Consistently, loss of *eps-8* and *RAC* orthologs improved motility in control polyQ19 and WT animals at day 5 of adulthood (Extended Data Fig. 2a,b). Although

**Fig. 1 | Elevated EPS-8/RAC signaling promotes polyQ-expanded aggregation in *C. elegans* neurons. a**, Filter trap of control polyQ19::YFP and expanded polyQ67::YFP (detected by anti-GFP antibody) expressed under neuronal-specific promoter in *C. elegans*. Right: SDS-PAGE with antibodies to GFP and α-tubulin. Graphs represent the relative percentage values of aggregated polyQ67 and total polyQ67 levels (corrected for α-tubulin loading control) to Q67 + Vector RNAi (mean ± s.e.m., *n* = 5 independent experiments). **b**, Percentage of nose touch responses/total trials per worm at days 1, 5 and 7 of adulthood (*n* = 80 worms per condition; each worm was tested 10 times to determine the response percentage). The box plots represent the 25th–75th percentiles, the lines depict the median and the whiskers show the minimum–maximum values. **c**, Chemotaxis index toward 0.5% benzaldehyde at days 1, 5 and 7 of adulthood (mean ± s.e.m., *n* = 3 independent experiments; 65–206 worms were scored per condition for each independent experiment). In **a**–**c**, RNAi was initiated after development. **d**, Filter trap analysis of polyQ67::YFP aggregates (detected by anti-GFP antibody) in worms expressing endogenous WT or Ub-less mutant EPS-8(K524R/K583R/K621R) at days 1 and 3 of adulthood. Right: SDS-PAGE with antibodies to GFP and α-tubulin. Graphs represent the relative percentage values of aggregated and total polyQ67 levels (corrected for α-tubulin) to day 1 adult Q67;EPS-8 (WT) worms (mean ± s.e.m., *n* = 4 independent experiments). **e**, Percentage of nose touch responses/total trials per worm at days 1 and 3 of adulthood (*n* = 50 worms per condition). The box plots represent the 25th–75th percentiles, the lines depict the median and the whiskers show the minimum–maximum values. **f**, Chemotaxis index toward 0.5% benzaldehyde at days 1 and 3 of adulthood (mean ± s.e.m., *n* = 3 independent experiments; 68–204 worms were scored per condition for each independent experiment). Statistical comparisons were made by one-way analysis of variance (ANOVA) with Dunnett's multiple comparisons test (**a**), two-way ANOVA with Sidak's multiple comparisons test (**b,c**) and two-way ANOVA with Fisher's least significant difference (LSD) test (**d**–**f**).

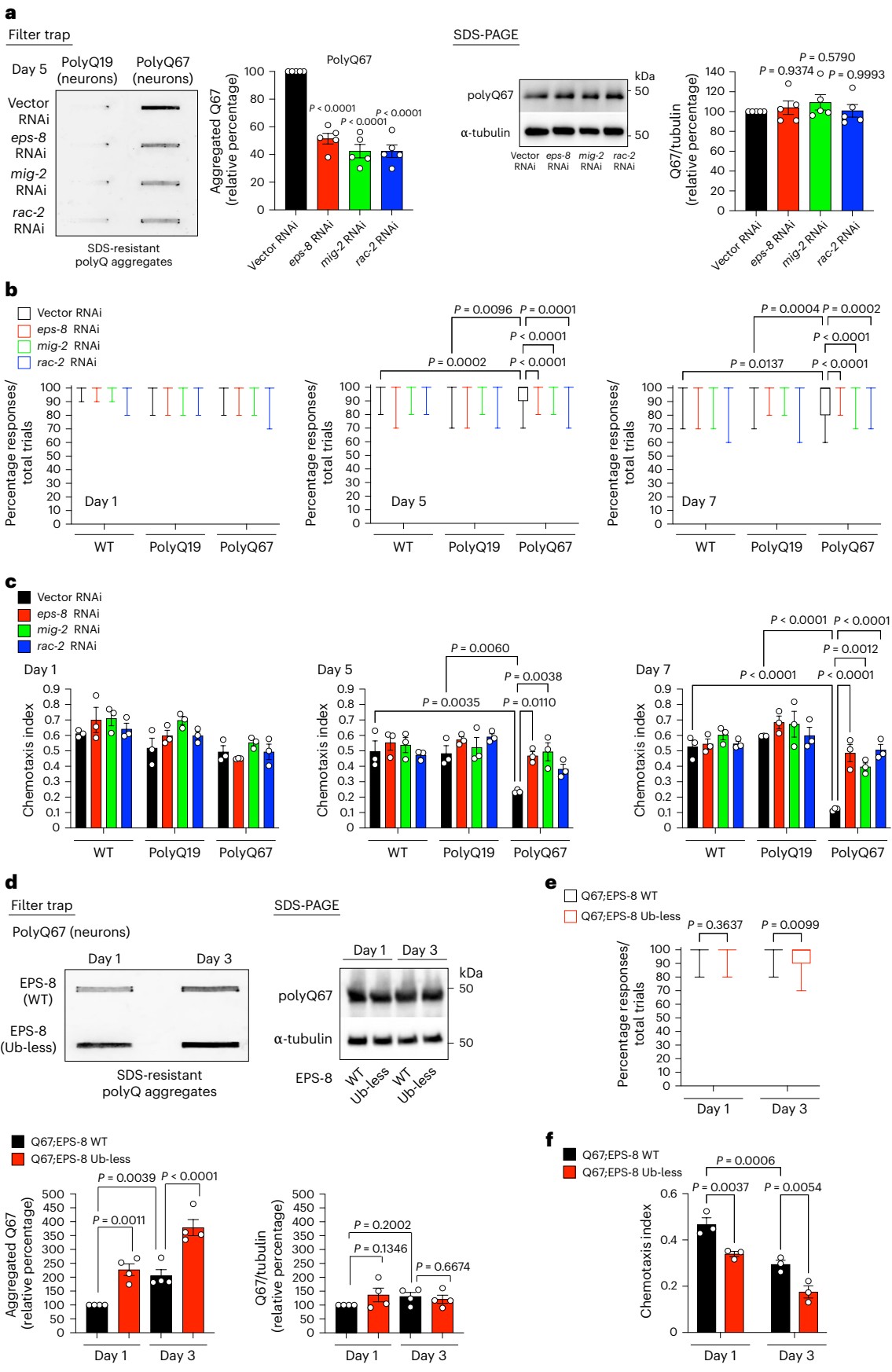

*eps-8* and RAC knockdown rescued motility deficits to levels similar to control Q19 worms under the same treatment (Extended Data Fig. 2a), these results were difficult to interpret due to the beneficial effects of EPS-8/RAC downregulation in aging control animals (Extended Data Fig. 2a,b).

In addition to motility deficits, the accumulation of polyQ aggregates also shortened lifespan (Extended Data Fig. 2c). We observed that *eps-8* knockdown extends lifespan in polyQ67-expressing worms (Extended Data Fig. 2c). However, loss of *eps-8* also extends lifespan in WT[14] and control polyQ19 worms (Extended Data Fig. 2c). Given that aging hastens disease-related phenotypes and *eps-8* knockdown delays aging, it is difficult to ascribe a specific effect of lowering EPS-8/RAC signaling on preventing disease-related phenotypes such as shortened lifespan and motility deficits. Thus, to better assess the link among hyperactivated EPS-8/RAC signaling, polyQ aggregation and neuronal dysfunction, we tested different behavioral assays.

Nose touch avoidance behavior is mediated by sensory neurons located in the head of the worm. On the first day of adulthood, polyQ67-expressing worms responded to nose touch similarly to control polyQ19 and WT worms (Fig. 1b). However, polyQ67 worms exhibited a decline in nose touch response compared with control animals at older ages (Fig. 1b). Notably, knockdown of *eps-8* or RAC orthologs rescued this age-related functional decline in polyQ67 worms but had no effect on aging control animals (Fig. 1b). PolyQ aggregation also induces neurotoxicity in chemosensory neurons, leading to impaired chemotaxis responses[31]. Although polyQ67 worms exhibited normal chemotaxis toward benzaldehyde on day 1 of adulthood, they developed chemotaxis deficits with age (Fig. 1c). However, knockdown of *eps-8* and RAC orthologs mitigated this decline in polyQ67 worms without affecting chemotaxis behavior in control animals (Fig. 1c). Similarly, *eps-8* knockout mutation ameliorated the age-related decline in nose touch responses and chemotaxis caused by polyQ67 expression (Extended Data Fig. 2d,e).

To further confirm a role of elevated EPS-8 levels in polyQ aggregation, we tested ubiquitin (Ub)-less EPS-8 mutant animals. In these worms, the ubiquitinated lysine sites of endogenous EPS-8 are replaced by arginine, blocking its ubiquitination and proteasomal degradation[14]. As a result, these animals exhibit upregulated EPS-8 protein levels from day 1 of adulthood, leading to hyperactivated RAC signaling in young adults[14]. We observed that the expression of Ub-less EPS-8 accelerates polyQ67 aggregation and disease-related behavioral changes from day 1 of adulthood (Fig. 1d–f and Extended Data Fig. 2f). These results establish a direct link between impaired ubiquitination and subsequent EPS-8 accumulation with polyQ aggregation.

With age, EPS-8 levels increase not only in neurons but also in other tissues, such as the intestine and muscle[14]. We found that lowering EPS-8/RAC signaling also prevents aggregation in worms expressing polyQ-expanded peptides specifically in the intestine or muscle (Fig. 2a,b). As in neurons, this decrease in polyQ aggregation was not accompanied by changes in total polyQ levels (Fig. 2a,b). Aggregation of polyQ peptides within muscle cells impairs muscle function, reducing organismal motility[9,25,32]. Accordingly, knockdown of either *eps-8* or RAC orthologs reduced age-related motility deficits in worms expressing polyQ-expanded repeats in muscle tissue but had no effect on day 1 of adulthood (Fig. 2c). Conversely, expression of Ub-less EPS-8 accelerated both aggregation and motility deficits in these worms (Extended Data Fig. 3a,b). However, unlike neuronal polyQ-expanded models, where we observed effects from day 1 of adulthood (Fig. 1d and Extended Data Fig. 2f), the detrimental impact of Ub-less EPS-8 on aggregation and motility in muscle cells started from day 3 of adulthood (Extended Data Fig. 3a,b).

Prompted by these results, we asked whether inhibition of elevated EPS-8/RAC activity prevents aggregation of other disease-related proteins. To this end, we used *C. elegans* models expressing ALS-related mutant variants of human FUS (P525L and R522G) and TDP-43 (M337V) in the nervous system, which recapitulate protein aggregation and neurotoxicity phenotypes[33,34]. Notably, knockdown of *eps-8* or RAC orthologs mitigated aggregation of ALS-related mutant FUS and TDP-43 variants (Fig. 2d,e). Moreover, reducing EPS-8/RAC signaling rescued age-related behavioral deficits in these ALS worm models, including loss of nose touch response and chemotaxis (Fig. 2f–i and Extended Data Fig. 3c–e).

**Fig. 2 | Reducing EPS-8/RAC signaling prevents disease-related changes in distinct *C. elegans* models. a**, Filter trap of day 5 adult *C. elegans* expressing polyQ44::YFP in the intestine (detected by anti-GFP antibody). Right: SDS–PAGE with antibodies to GFP and α-tubulin. Graphs represent the relative percentage values of aggregated polyQ44 and total polyQ44 levels (corrected for α-tubulin) to Vector RNAi (mean ± s.e.m., *n* = 4 independent experiments). **b**, Filter trap of day 5 adult *C. elegans* expressing polyQ40::YFP in the muscle (detected by anti-GFP antibody). Right: SDS–PAGE with antibodies to GFP and α-tubulin. Graphs represent the relative percentage values of aggregated polyQ40 and total polyQ40 levels (corrected for α-tubulin) to Vector RNAi (mean ± s.e.m., *n* = 4 independent experiments). **c**, Body bends per second in worms expressing polyQ40 in the muscle at days 1, 5 and 7 of adulthood (day 1 (D1) + Vector RNAi: *n* = 41 worms; D1 + *eps-8* RNAi: *n* = 41; D1 + *mig-2* RNAi: *n* = 46; D1 + *rac-2* RNAi: *n* = 44; D5 + Vector RNAi: *n* = 58 worms; D5 + *eps-8* RNAi: *n* = 52; D5 + *mig-2* RNAi: *n* = 48; D5 + *rac-2* RNAi: *n* = 59; D7 + Vector RNAi: *n* = 42 worms; D7 + *eps-8* RNAi: *n* = 49; D7 + *mig-2* RNAi: *n* = 54; D7 + *rac-2* RNAi: *n* = 49). The box plots represent the 25th–75th percentiles, the line depicts the median and the whiskers show the minimum–maximum values. **d**, Knockdown of *eps-8* or RAC orthologs ameliorates aggregation of ALS-related mutant FUS[P525L] variant in the neurons of day 5 adult *C. elegans* (detected by anti-FUS antibody). Right: SDS–PAGE with antibodies to FUS and α-tubulin. Graphs represent the relative percentage values of aggregated and total FUS[P525L] protein levels (corrected for α-tubulin) to Vector RNAi (mean ± s.e.m., *n* = 4 independent experiments). **e**, Knockdown of *eps-8* or RAC orthologs ameliorates aggregation of ALS-related mutant TDP-43[M337V] variant in the neurons of day 5 adult worms (detected by anti-TDP-43 antibody). Right: SDS–PAGE with antibodies to TDP-43 and α-tubulin. Graphs represent the relative percentage values of aggregated and total TDP-43[M337V] protein levels (corrected for α-tubulin) to Vector RNAi (mean ± s.e.m., *n* = 4 independent experiments). **f**, Percentage of nose touch responses/total trials in WT worms and transgenic worms expressing WT FUS or ALS-related mutant FUS[R522G] and FUS[P525L] variants (D1: *n* = 40 worms per condition; D5: *n* = 80; D7: *n* = 80). The box plots represent the 25th–75th percentiles, the line depicts the median and the whiskers show the minimum–maximum values. **g**, Chemotaxis index of FUS-ALS worm models toward 0.5% benzaldehyde at day 5 of adulthood (mean ± s.e.m., *n* = 3 independent experiments; 72–148 worms were scored per condition for each independent experiment). **h**, Percentage of nose touch responses/total trials in WT worms and transgenic worms expressing WT TDP-43 or ALS-related mutant TDP-43[M337V] variant (D1: *n* = 30 worms per condition; D5: *n* = 70; D7: *n* = 70). The box plots represent the 25th–75th percentiles, the line depicts the median and the whiskers show the minimum–maximum values. **i**, Chemotaxis index of TDP-43 ALS worm models toward 0.5% benzaldehyde at day 5 of adulthood (mean ± s.e.m., *n* = 3 independent experiments; 48–439 worms were scored per condition for each independent experiment). **j**, Number of GABAergic neurons (*unc-25p*::GFP) in the nerve cord of TDP-43 ALS worms at day 5 of adulthood (mean ± s.e.m., TDP-43(WT) + Vector RNAi: *n* = 88 worms from two independent experiments; TDP-43(WT) + *eps-8* RNAi: *n* = 50; TDP-43(WT) + *mig-2* RNAi: *n* = 49; TDP-43(WT) + *rac-2* RNAi: *n* = 49; TDP-43(M337V) + Vector RNAi: *n* = 73; TDP-43(M337V) + *eps-8* RNAi: *n* = 45; TDP-43(M337V) + *mig-2* RNAi: *n* = 53; TDP-43(M337V) + *rac-2* RNAi: *n* = 49). The box plots represent the 25th–75th percentiles, the line depicts the median and the whiskers show the minimum–maximum values. **k**, Graph represents the percentage of worms displaying discontinuities in the nerve cord at day 5 of adulthood (percentage from 49–88 worms per condition from two independent experiments). Statistical comparisons were made by one-way ANOVA with Dunnett's multiple comparisons test (**a**,**b**,**d**,**e**), two-way ANOVA with Sidak's multiple comparisons test (**c**,**f**–**j**) and two-sided Fisher's exact test from contingency table analysis of number of worms displaying discontinuities in the nerve cord (**k**).

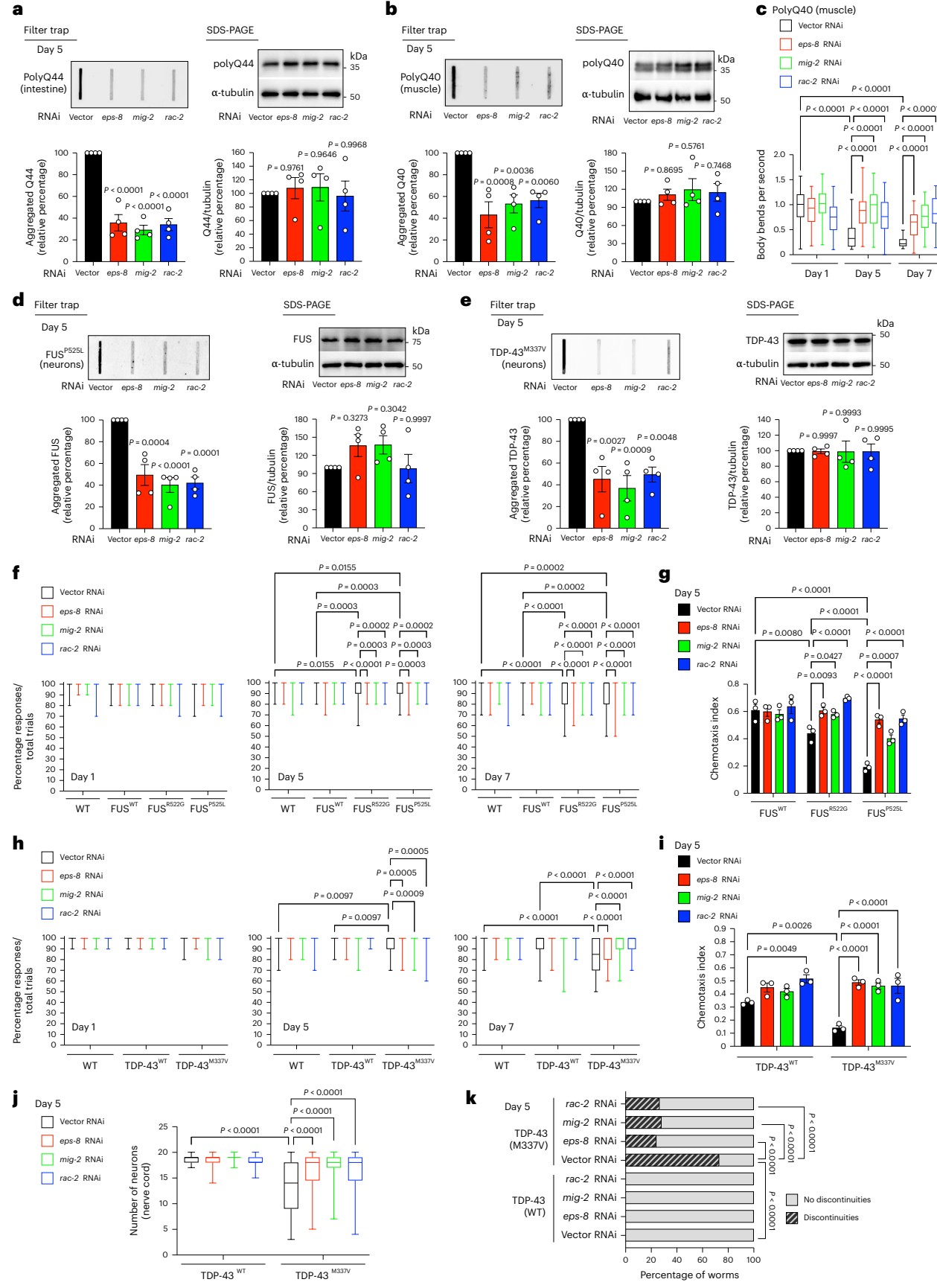

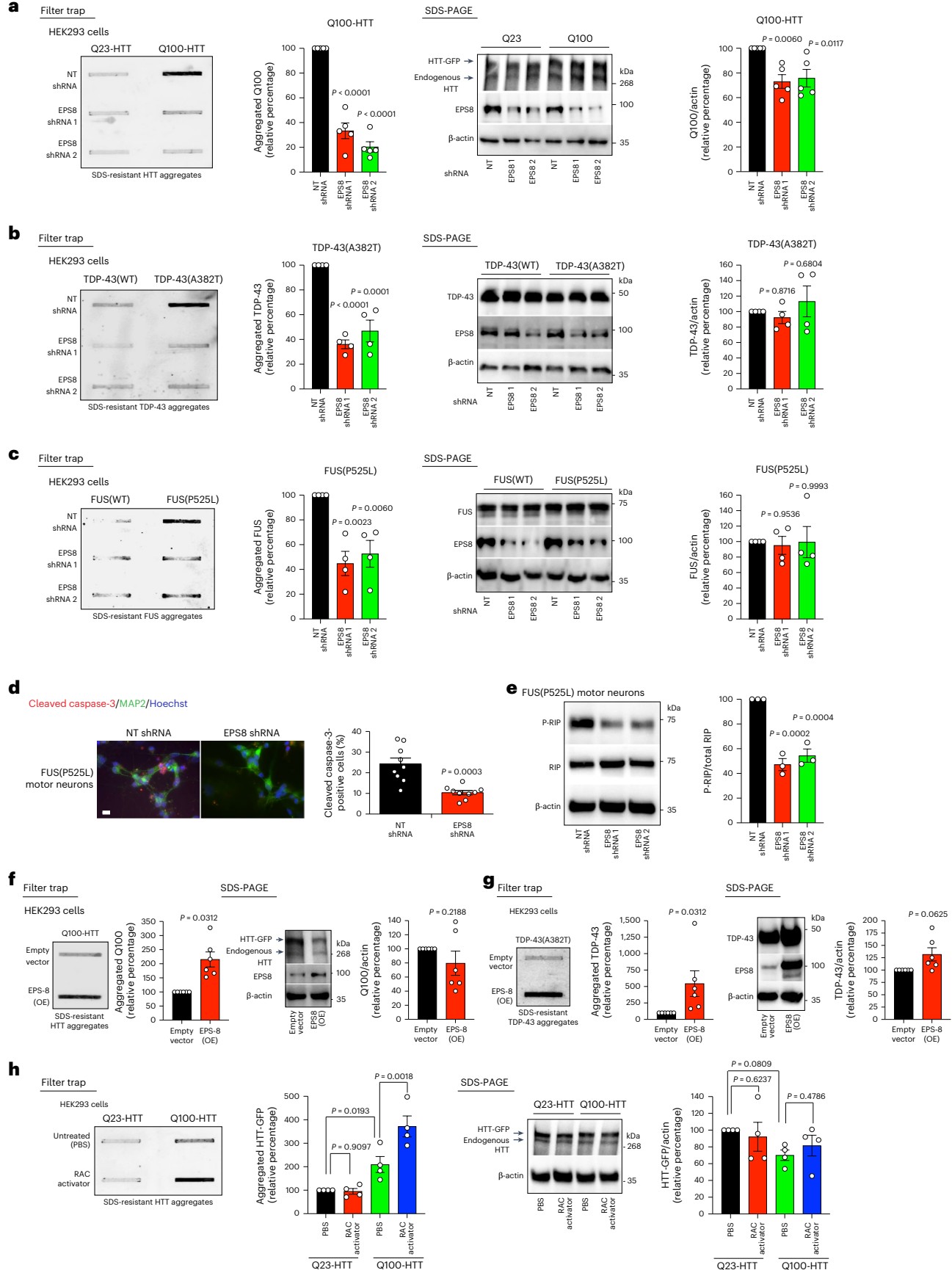

**Fig. 3 | EPS8/RAC signaling modulates disease-related protein aggregation in human cells. a**, Filter trap with anti-GFP antibody of HEK293 human cells expressing Q23-HTT-GFP or Q100-HTT-GFP treated with either non-targeting (NT) shRNA or independent shRNA constructs against EPS8. Right: SDS-PAGE with antibodies to HTT, EPS8 and β-actin loading control. Graphs represent the relative percentage values of aggregated and total Q100-HTT protein levels (corrected for β-actin) to NT shRNA Q100-HTT cells (mean ± s.e.m., $n = 5$ independent experiments). **b**, Filter trap with anti-TDP-43 antibody of HEK293 cells expressing WT TDP-43 or ALS-related mutant TDP-43$^{A382T}$. Right: SDS-PAGE with antibodies to TDP-43, EPS8 and β-actin loading control. Graphs represent the relative percentage values of aggregated and total TDP-43$^{A382T}$ protein levels (corrected for β-actin) to NT shRNA TDP-43$^{A382T}$ cells (mean ± s.e.m., $n = 4$ independent experiments). **c**, Filter trap with anti-FUS antibody of HEK293 cells expressing WT FUS or ALS-related mutant FUS$^{P525L}$. Right: SDS-PAGE with antibodies to FUS, EPS8 and β-actin loading control. Graph represents the relative percentage values of aggregated and total FUS$^{P525L}$ protein levels (corrected for β-actin) to NT shRNA FUS$^{P525L}$ cells (mean ± s.e.m., $n = 4$ independent experiments). **d**, Immunocytochemistry of FUS(P525L) ALS iPSC-derived motor neurons with anti-cleaved caspase-3 (red), anti-MAP2 (green) and Hoechst (nucleus, blue). Scale bar, 10 µm. Graph represents the percentage of cleaved caspase-3-positive cells/total nuclei (mean ± s.e.m. of nine biological replicates from two independent experiments, NT shRNA: 383 total nuclei and EPS8 shRNA 2: 192 total nuclei). **e**, Western blot analysis of FUS(P525L) ALS

iPSC motor neurons with antibodies to phosphorylated RIP (P-RIP) at Ser166, total RIP and β-actin loading control. Graph represents the relative percentage ratio of P-RIP/total RIP levels to NT shRNA (mean ± s.e.m., $n = 3$ independent experiments). **f**, Increased aggregation of Q100-HTT-GFP (detected by anti-GFP antibody) in HEK293 cells overexpressing (OE) EPS8. Right: SDS-PAGE with antibodies to HTT, EPS8 and β-actin. Graphs represent the relative percentage values of aggregated and total Q100-HTT (corrected for β-actin) levels to Q100-HTT cells + empty vector (mean ± s.e.m., $n = 6$ independent experiments). **g**, Overexpression of EPS8 increases aggregation of mutant TDP-43$^{A382T}$ (detected by anti-TDP-43 antibody) in HEK293 cells. Right: SDS-PAGE with antibodies to TDP-43, EPS8 and β-actin. Graphs represent the relative percentage values of aggregated and total TDP-43$^{A382T}$ protein levels (corrected for β-actin) to TDP-43$^{A382T}$ cells + empty vector (mean ± s.e.m., $n = 6$ independent experiments). **h**, Filter trap with anti-GFP antibody of HEK293 human cells expressing control Q23-HTT-GFP or aggregation-prone Q100-HTT-GFP. The treatment with 2 U ml$^{-1}$ RAC activator (6 hours) hastens aggregation of Q100-HTT-GFP. Right: SDS-PAGE with antibodies to HTT and β-actin. Graphs represent the relative percentage of aggregated and total HTT-GFP levels (corrected for β-actin) to Q23-HTT-GFP (PBS vehicle control) cells (mean ± s.e.m., $n = 4$ independent experiments). Statistical comparisons were made by one-way ANOVA with Dunnett's multiple comparisons test (**a**–**c**,**e**), two-sided $t$-test for unpaired samples (**d**), two-tailed Wilcoxon signed-rank test (**f**,**g**) and two-way ANOVA with Fisher's LSD test (**h**).

A previous study reported significant degeneration of GABAergic neurons in the nerve cords of worms expressing mutant TDP-43 variants[33]. Indeed, TDP-43$^{M337V}$ worms exhibit a loss of GABAergic neuronal cell bodies and disruptions in nerve cord continuity of neurons compared with those expressing WT TDP-43 (Fig. 2j,k and Extended Data Fig. 4a)[33]. Notably, knockdown of either *eps-8* or *RAC* orthologs reduced GABAergic degeneration in TDP-43$^{M337V}$ worms (Fig. 2j,k and Extended Data Fig. 4a). Although polyQ67 and FUS-ALS models display a decline in neuronal function, they did not exhibit GABAergic neurodegeneration (Extended Data Fig. 4b–g). Together, our data indicate that inhibiting EPS-8/RAC signaling during aging can alleviate pathological phenotypes in *C. elegans* induced by distinct disease-related mutant proteins.

## EPS8/RAC signaling modulates disease-related protein aggregation in human cells

Because EPS-8/RAC signaling induces disease-related protein aggregation in *C. elegans*, we asked whether these effects are conserved in human cells. To assess this, we used human HEK293 cell models expressing either control (Q23) or polyQ-expanded (Q100) HTT. Although control Q23-HTT does not form aggregates, mutant Q100-HTT accumulates into insoluble aggregates[9,26] (Fig. 3a). Notably, EPS8 knockdown decreased the accumulation of Q100-HTT aggregates in human cells (Fig. 3a). In contrast, EPS8 knockdown did not change the intracellular distribution of mutant HTT aggregates (Extended Data Fig. 5a). In addition to high-molecular-weight insoluble aggregates, the accumulation

of small, soluble assemblies of pathological proteins also contributes to disease-related changes[35,36]. Using native gel electrophoresis, we did not detect changes in the levels of polyQ-expanded soluble oligomers upon EPS8 knockdown (Extended Data Fig. 5b), indicating that this pathway specifically influences the assembly of insoluble aggregates. To further examine the role of EPS8 in protein aggregation, we generated HEK293 cell models expressing either WT or ALS-related mutant TDP-43$^{A382T}$ and FUS$^{P525L}$ variants. We observed that both mutant TDP-43 and FUS form insoluble aggregates in HEK293 cells, but EPS8 knockdown prevented their aggregation (Fig. 3b,c).

Intrigued by these findings, we investigated whether knockdown of EPS8 can attenuate disease-related neurodegeneration. ALS is characterized by the selective loss of motor neurons[37]. Along these lines, motor neurons differentiated from induced pluripotent stem cells (iPSCs) expressing the severe ALS-linked FUS$^{P525L}$ mutation exhibit elevated apoptotic rates compared with isogenic controls[9,37]. However, EPS8 knockdown ameliorated apoptosis in these cells (Fig. 3d). Besides apoptosis, other mechanisms, such as necroptosis, contribute to moto-neuronal death in ALS[38]. Notably, we observed that EPS8 knockdown also reduces phosphorylation and subsequent activation of RIP kinase in ALS motor neurons (Fig. 3e), a marker of necroptotic cell death[39]. Together, these results suggest that lowering EPS8 levels mitigates ALS-related neurodegeneration.

We then tested whether increasing EPS8 levels is sufficient to promote disease-related protein aggregation. Indeed, EPS8

**Fig. 4 | Excessive actin polymerization through EPS-8/RAC hyperactivation promotes disease-related protein aggregation. a**, Filter trap of polyQ67::YFP aggregates (detected by anti-GFP antibody) in day 5 adult worms treated with 10 µM CytoD or DMSO vehicle control for 6 hours before lysis. Right: SDS-PAGE with antibodies to GFP and α-tubulin. Graphs represent the relative percentage values of aggregated polyQ67 and total polyQ67 levels (corrected for α-tubulin loading control) to Q67 + Vector RNAi + DMSO (mean ± s.e.m., $n = 3$ independent experiments). **b**, Filter trap with anti-GFP antibody of HEK293 human cells expressing Q23-HTT-GFP or Q100-HTT-GFP treated with 2 µM CytoD or DMSO vehicle control for 4 hours before lysis. Right: SDS-PAGE with antibodies to HTT and β-actin loading control. Graphs represent the relative percentage of aggregated and total HTT-GFP levels (corrected for β-actin) to Q23-HTT-GFP + DMSO (mean ± s.e.m., $n = 3$ independent experiments). **c**, Filter trap of mutant FUS$^{P525L}$ aggregates (detected by anti-FUS antibody) in day 5 adult worms treated with 10 µM CytoD for 6 hours. Right: SDS-PAGE with antibodies to FUS and α-tubulin. Graphs represent the relative percentage values

of aggregated and total FUS levels (corrected for α-tubulin loading control) to FUS$^{P525L}$ + DMSO (mean ± s.e.m., $n = 6$ independent experiments). **d**, Filter trap of mutant TDP-43$^{M337V}$ aggregates (detected by anti-TDP-43 antibody) in day 5 adult worms treated with 10 µM CytoD for 6 hours. Right: SDS-PAGE with antibodies to TDP-43 and α-tubulin. Graphs represent the relative percentage values of aggregated and total TDP-43 (corrected for α-tubulin) levels to TDP-43$^{M337V}$ + DMSO (mean ± s.e.m., $n = 6$ independent experiments). **e**, Filter trap analysis of polyQ67::YFP (detected by anti-GFP antibody) in day 3 adult worms expressing endogenous WT EPS-8 or Ub-less mutant EPS-8 treated with 10 µM CytoD (6 hours). Right: SDS-PAGE with antibodies to GFP and α-tubulin. Graphs represent the relative percentage values of aggregated polyQ67 and total polyQ67 levels (corrected for α-tubulin) to Q67;EPS-8(WT) + DMSO (mean ± s.e.m., $n = 3$ independent experiments). Statistical comparisons were made by two-way ANOVA with Sidak's multiple comparisons test (**a**), two-way ANOVA with Fisher's LSD test (**b**,**e**) and two-tailed Wilcoxon signed-rank test (**c**,**d**).

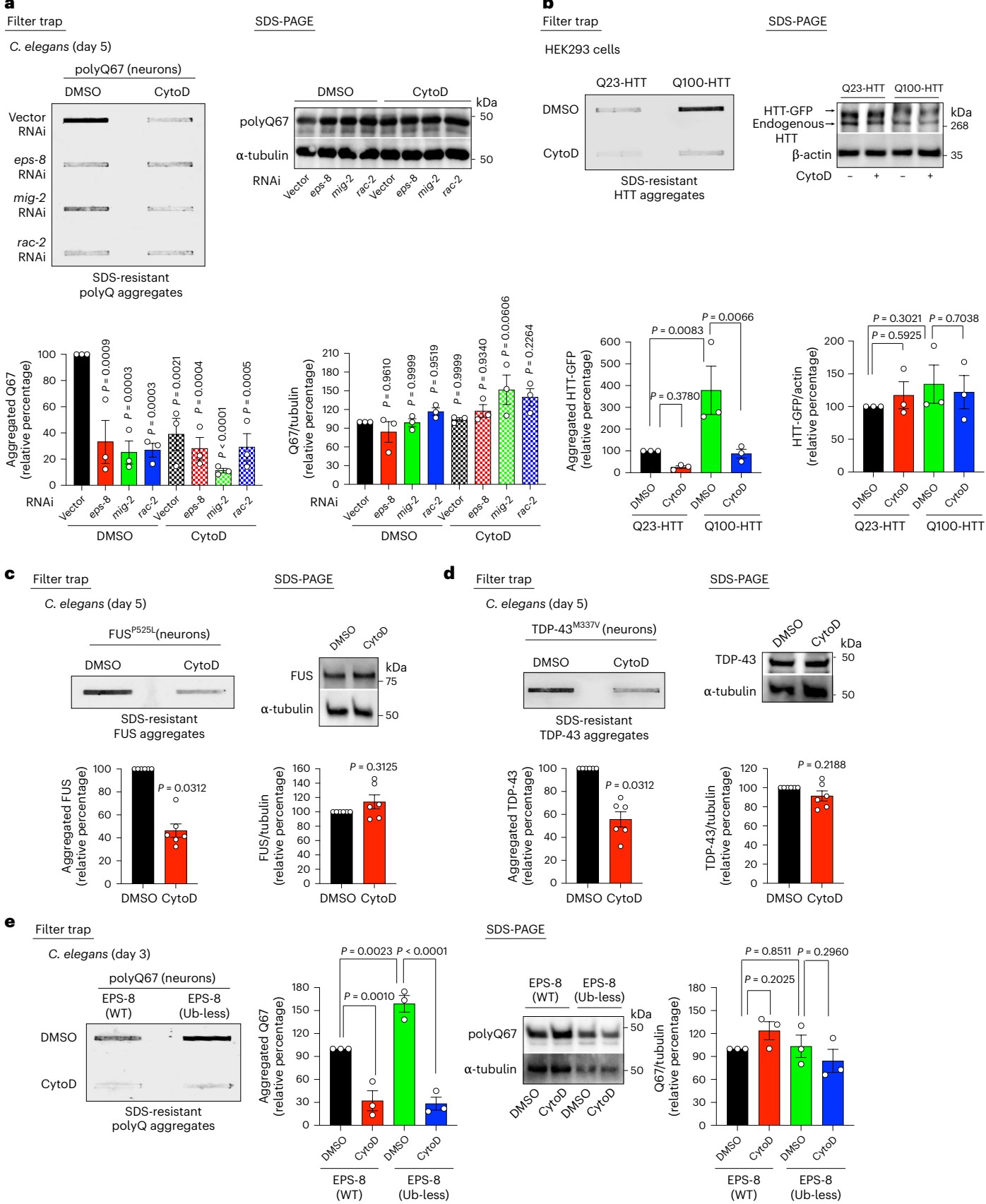

overexpression hastened aggregation of disease-related mutant proteins in HEK293 human cells (Fig. 3f,g). Likewise, treatment with a RAC activator increased disease-related protein aggregation in human cells (Fig. 3h). Collectively, these results indicate an evolutionary conserved role of EPS8/RAC signaling in disease-related protein aggregation.

## Elevated actin polymerization and JNK signaling promote protein aggregation

Our findings suggest that lowering EPS8 levels attenuates pathological protein aggregation without decreasing the levels of disease-related proteins. Consistent with this, knockdown of EPS8 in both worms and human cells did not enhance proteasome activity or LC3 lipidation, a marker of autophagosomes (Extended Data Fig. 6a–f). Thus, we asked whether EPS8/RAC signaling influences protein aggregation through other mechanisms. EPS8/RAC signaling promotes actin cytoskeleton polymerization and remodeling[40]. During aging, hyperactivation of EPS8/RAC signaling leads to excessive actin polymerization, destabilizing the cytoskeleton[14]. To test whether excessive actin polymerization contributes to disease-related aggregation, we treated day 5 adult worms with the actin polymerization inhibitor cytochalasin D (CytoD) for 6 hours (Fig. 4a). Indeed, CytoD treatment reduced polyQ67 aggregation in day 5 adult worms. In contrast, this treatment did not further decrease the already low aggregation levels in worms subjected to RNAi against eps-8 or RAC orthologs (Fig. 4a). Similarly, CytoD treatment for 4 hours reduced polyQ100-HTT aggregation in HEK293 human cells (Fig. 4b).

Besides polyQ-expanded proteins, the treatment with CytoD also decreased aggregation of ALS-related mutant FUS and TDP-43 variants in day 5 adult worms (Fig. 4c,d). To further assess the link among EPS-8/RAC hyperactivation, excessive actin polymerization and disease-related protein aggregation, we treated Ub-less mutant EPS-8 worms with CytoD. Notably, CytoD treatment diminished the accelerated polyQ67 aggregation triggered by Ub-less mutant EPS-8 in young worms at day 3 of adulthood (Fig. 4e). Similarly, CytoD treatment decreased the elevated aggregation of polyQ100-HTT induced by EPS8 overexpression in human cells (Extended Data Fig. 7).

In addition to inducing actin polymerization, the EPS8/RAC pathway also regulates JNK activity[41]. During aging, EPS-8/RAC hyperactivates JNK signaling in C. elegans[14]. In previous work, we found that knocking down the worm JNK homolog kgb-1 after development extends longevity in WT animals and alleviates the shortened lifespan of Ub-less EPS-8 mutants[14]. Notably, kgb-1 knockdown reduced aggregation of polyQ-expanded repeats (Fig. 5a). Similarly, loss of jnk-1, another C. elegans JNK homolog, also decreased polyQ aggregation in day 5 adults (Extended Data Fig. 8a). Moreover, knockdown of either kgb-1 or jnk-1 mitigated the accelerated aggregation of polyQ67 in Ub-less EPS-8 mutants at younger ages (Fig. 5b and Extended Data Fig. 8b).

Besides polyQ peptides, kgb-1 knockdown effectively prevented aggregation of FUS (R522G and P525L) and TDP-43 (M337V) mutant variants in day 5 adult worms (Fig. 5c,d). Although to a lesser extent, loss of jnk-1 also decreased aggregation of TDP-43[M337V] and the severe

FUS[P525L] mutant variant, but it did not significantly prevent aggregation of mutant FUS[R522G] (Extended Data Fig. 8c,d). These results suggest that the JNK homolog kgb-1 has stronger effects on disease-related aggregation than jnk-1. Altogether, our data indicate that hyperactivation of EPS8/RAC-regulated pathways, such as JNK activity and actin polymerization, contributes to disease-related protein aggregation.

## USP4 inhibits proteasomal degradation of EPS8 and triggers disease-related protein aggregation

In C. elegans, EPS-8 undergoes increased deubiquitination during aging, which prevents its degradation by the proteasome[14]. Aging induces elevated levels of various DUBs, leading to the dysregulation of distinct biological processes[14]. Conversely, treating aged worms with a broad-spectrum DUB inhibitor restores ubiquitination levels and extends lifespan[14]. Notably, we observed that DUB inhibitor also attenuates protein aggregation and behavioral deficits in C. elegans disease models (Fig. 6a–e and Extended Data Fig. 9a–f). However, DUB inhibitor did not decrease the total levels of disease-related proteins (Fig. 6a,b). These results support that elevated DUB activity promotes pathological protein aggregation by upregulating pro-aging factors such as EPS-8. Thus, we aimed to identify the specific DUB responsible for preventing EPS-8 degradation.

Among the DUBs upregulated with age[14], single knockdown of usp-4, csn-6 or F07A11.4 extended lifespan in WT worms (Fig. 6f and Supplementary Table 1). However, only the loss of usp-4 reduced polyQ67 aggregation (Fig. 6g). Similar to eps-8 knockdown, lowering usp-4 levels did not decrease the total amount of polyQ67 peptides (Fig. 6g). In addition to polyQ-expanded peptides, usp-4 knockdown also prevented the aggregation of ALS-related mutant proteins (Fig. 7a,b). Consistent with this reduction in aggregation, usp-4 knockdown rescued both nose touch response and chemotaxis deficits in polyQ and ALS models during aging while having no effect on these behavioral responses in control worms (Fig. 7c–g and Extended Data Fig. 10a). Moreover, loss of usp-4 not only extended longevity and alleviated age-related motility deficits in control worms but also prevented the detrimental effects of disease-related mutant proteins on lifespan and motility (Extended Data Fig. 10b–e).

Given that knockdown of usp-4 phenocopies the effects of eps-8 RNAi (that is, lifespan extension and prevention of disease-related changes), we hypothesized that this DUB contributes to the age-associated decline in proteasomal degradation and subsequent accumulation of EPS-8. Indeed, usp-4 knockdown was sufficient to decrease the protein levels of EPS-8 in aged worms (Fig. 7h). By contrast, loss of usp-4 did not reduce the protein levels of Ub-less EPS-8 mutant variant (Extended Data Fig. 10f). Accordingly, knockdown of usp-4 extended lifespan in worms expressing WT EPS-8 but not the short lifespan of Ub-less EPS-8 mutants (Fig. 7i and Supplementary Table 1). Likewise, usp-4 RNAi did not suppress polyQ67 aggregation in Ub-less EPS-8 mutants (Fig. 7j).

In human cells, reducing USP4 levels also promoted EPS8 degradation, a process blocked by proteasome inhibition (Fig. 8a). Moreover,

**Fig. 5 | Elevated JNK signaling via EPS-8/RAC hyperactivation promotes disease-related protein aggregation. a**, Knockdown of kgb-1 after development prevents polyQ67::YFP aggregation (detected by anti-GFP antibody) in the neurons of day 5 adult C. elegans. Right: SDS-PAGE with antibodies to GFP and α-tubulin. Graphs represent the relative percentage values of aggregated and total polyQ67 (corrected for α-tubulin loading control) to Vector RNAi (mean ± s.e.m., n = 7 independent experiments). **b**, Filter trap analysis of polyQ67::YFP aggregates (detected by anti-GFP antibody) in day 3 adult worms expressing endogenous WT or Ub-less mutant EPS-8 on kgb-1 RNAi treatment. Right: SDS-PAGE with antibodies to GFP and α-tubulin. Graphs represent the relative percentage values of aggregated polyQ67 and total polyQ67 levels (corrected for α-tubulin loading control) to Q67;EPS-8(WT) + Vector RNAi (mean ± s.e.m., n = 3 independent experiments). **c**, Knockdown of kgb-1 after development ameliorates aggregation

of ALS-related mutant FUS variants in the neurons of day 5 adult worms (detected by anti-FUS antibody). Right: SDS-PAGE with antibodies to FUS and α-tubulin. Graphs represent the relative percentage values of aggregated and total FUS levels (corrected for α-tubulin loading control) to WT FUS + Vector RNAi (mean ± s.e.m., n = 3 independent experiments). **d**, Knockdown of kgb-1 after development decreases TDP-43[M337V] aggregation in the neurons of day 5 adult worms (detected by anti-TDP-43 antibody). Right: SDS-PAGE with antibodies to TDP-43 and α-tubulin. Graphs represent the relative percentage values of aggregated TDP-43 and total TDP-43 levels (corrected for α-tubulin loading control) to WT TDP-43 + Vector RNAi (mean ± s.e.m., n = 3 independent experiments). Statistical comparisons were made by two-tailed Wilcoxon signed-rank test (**a**), two-way ANOVA with Fisher's LSD test (**b,d**) and two-way ANOVA with Sidak's multiple comparisons test (**c**).

co-immunoprecipitation (co-IP) experiments revealed that USP4 interacts with EPS8 in human cells (Fig. 8b). Prompted by these findings, we assessed whether lowering USP4 levels prevents disease-related protein aggregation in HEK293 cells. Indeed, we found that knockdown of USP4

decreases polyQ-expanded mutant HTT aggregation in these cells (Fig. 8c). In contrast to *C. elegans*, loss of USP4 also reduced the total levels of mutant HTT in human cells (Fig. 8c). However, overexpression of EPS8 counteracted the effects of USP4 knockdown on mutant HTT

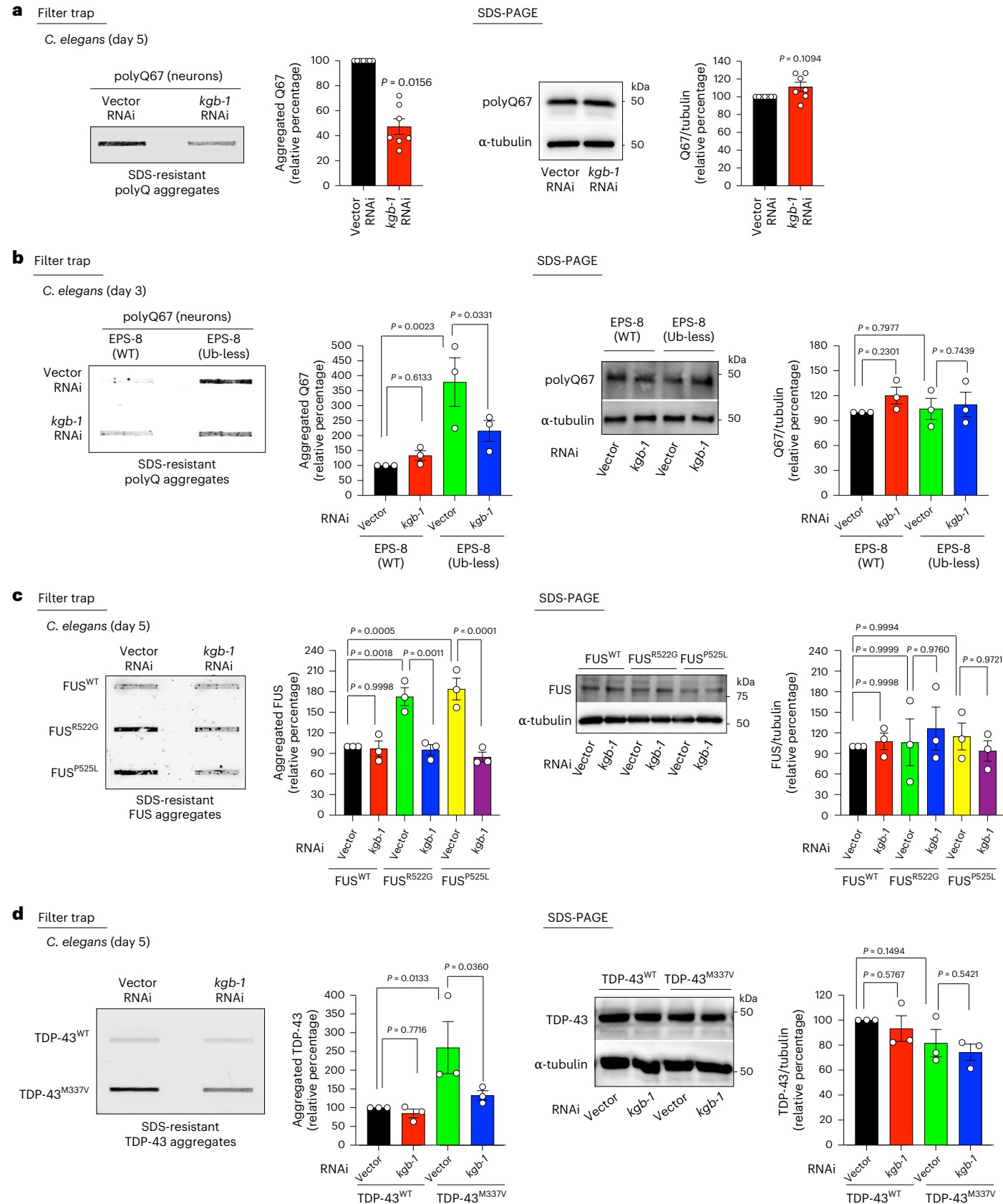

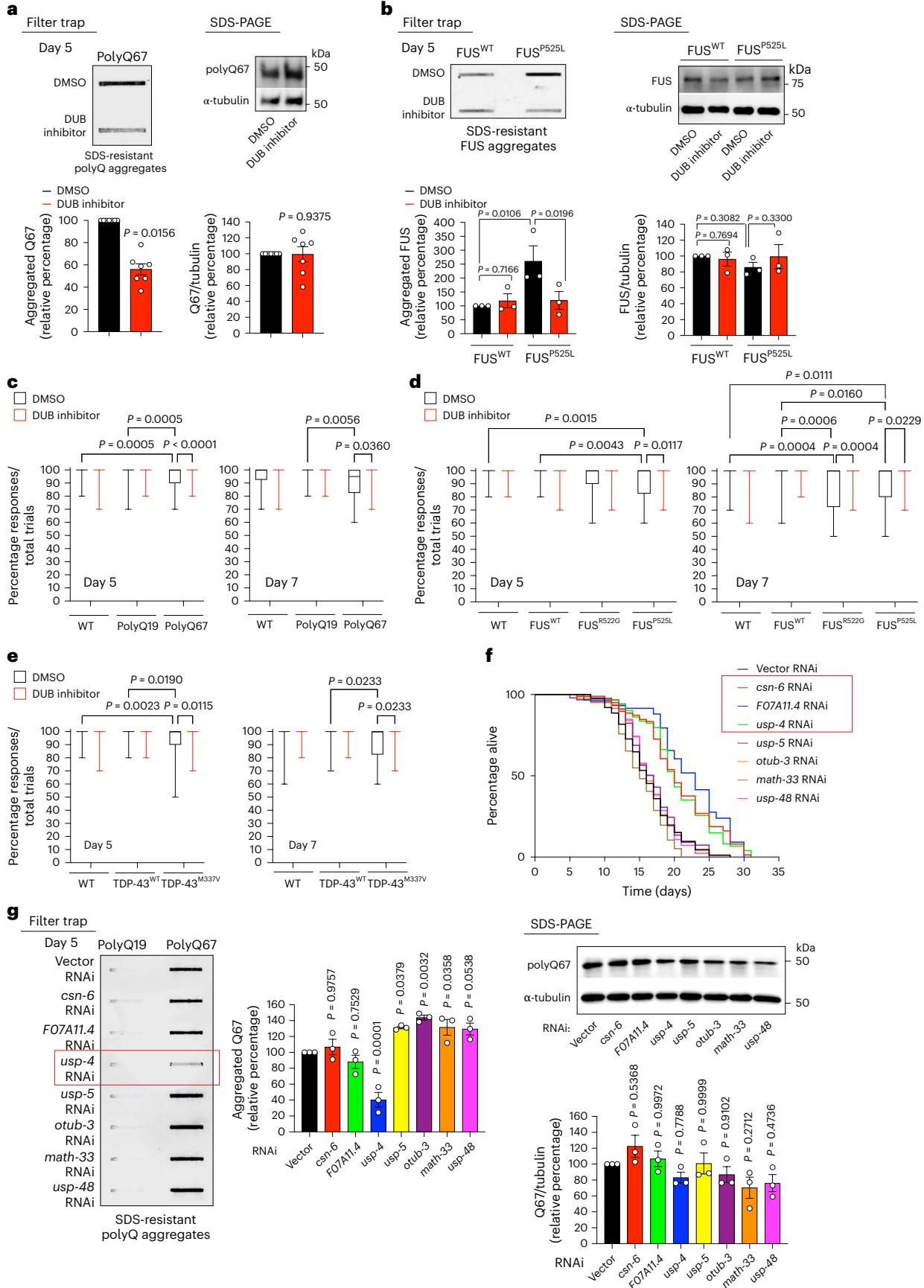

**Fig. 6 | DUB inhibition in adult worms reduces disease-related protein aggregation. a**, Inhibition of elevated DUB activity in day 5 adult worms ameliorates aggregation of neuronal polyQ67::YFP (detected by anti-GFP antibody). Right: SDS-PAGE with antibodies to GFP and α-tubulin loading control. Graphs represent the relative percentage values of aggregated and total polyQ67 levels (corrected for α-tubulin) to Q67 + DMSO vehicle control (mean ± s.e.m., $n = 7$ independent experiments). **b**, Inhibition of elevated DUB activity decreases aggregation of mutant FUS$^{P525L}$ in *C. elegans* neurons (detected by anti-FUS antibody). Right: SDS-PAGE of total FUS protein levels with anti-FUS antibody. Graphs represent the relative percentage values of aggregated and total FUS protein levels (corrected for α-tubulin) to WT FUS + DMSO vehicle control (mean ± s.e.m., $n = 3$ independent experiments). **c**, Percentage of nose touch responses/total trials in WT worms and transgenic worms expressing polyQ19 or polyQ67 in neurons (day 5 (D5): $n = 80$ worms per condition (except WT + DUB inhibitor, $n = 79$); D7: $n = 40$ worms per condition). **d**, Percentage of nose touch responses/total trials in WT worms and transgenic worms expressing WT FUS or ALS-related mutant FUS$^{R522G}$ and FUS$^{P525L}$ variants ($n = 40$ worms per condition). **e**, Percentage of nose touch responses/total trials in WT worms and transgenic worms expressing WT TDP-43 or ALS-related mutant TDP-43$^{M337V}$ variant ($n = 40$ worms per condition). In **c–e**, the box plots represent the 25th–75th percentiles, the lines depict the median and the whiskers show the minimum–maximum

values. Worms were treated with 13.7 µg ml$^{-1}$ PR-619 (broad-spectrum DUB inhibitor) or vehicle control (DMSO) for 4 hours on day 5 of adulthood (**a**) or for 24 hours on day 4 of adulthood (**b–e**) and analyzed at the indicated ages. **f**, Single knockdown after development of *csn-6* (mean ± s.e.m.: 20.56 days ± 0.62, $P < 0.0001$), *F07A11.4* (mean ± s.e.m.: 22.16 ± 0.71, $P < 0.0001$) and *usp-4* (mean ± s.e.m.: 20.42 ± 0.59, $P < 0.0001$) extends lifespan in WT worms compared with Vector RNAi controls (mean ± s.e.m.: 16.22 ± 0.44). By contrast, knockdown of *usp-5* (mean ± s.e.m.: 16.66 ± 0.45, $P = 0.4589$), *otub-3* (mean ± s.e.m.: 16.59 ± 0.46, $P = 0.6532$), *math-33* (mean ± s.e.m.: 15.31 ± 0.36, $P = 0.0552$) or *usp-48* (mean ± s.e.m.: 16.67 ± 0.36, $P = 0.8363$) does not affect lifespan. $P$ values: two-sided log-rank test, $n = 96$ worms per condition. Supplementary Table 1 contains statistical analysis and replicate data from independent lifespan experiments. **g**, Knockdown of *usp-4* after development prevents polyQ67::YFP aggregation in day 5 adult worms (detected by anti-GFP antibody). Right: SDS-PAGE with antibodies to GFP and α-tubulin. Graph represents the relative percentage values of aggregated and total polyQ67 protein levels (corrected for α-tubulin) to Q67 + Vector RNAi (mean ± s.e.m., $n = 3$ independent experiments). Statistical comparisons were made by two-tailed Wilcoxon signed-rank test (**a**), two-way ANOVA with Fisher's LSD test (**b**), two-way ANOVA with Sidak's multiple comparisons test (**c–e**), two-sided log-rank test (**f**) and one-way ANOVA with Dunnett's multiple comparisons test (**g**).

---

aggregation (Fig. 8d), suggesting that USP4 regulates mutant HTT proteostasis through EPS8 levels.

In addition, USP4 knockdown and the subsequent degradation of EPS8 prevented aggregation of ALS-related mutant FUS and TDP-43 variants in human cells, without affecting their total protein levels (Fig. 8e,f). Similar to EPS8 knockdown, loss of USP4 ameliorated the neurodegeneration phenotype induced by ALS-linked FUS$^{P525L}$ in iPSC-derived motor neurons (Fig. 8g). These results highlight an evolutionarily conserved role of USP4 in regulating EPS8 levels and its impact on age-associated neurodegenerative disorders.

## Discussion

The late onset and heterogeneity of protein aggregates characteristic of distinct neurodegenerative diseases suggest common underlying cellular changes associated with aging. *C. elegans* models of Huntington's disease and ALS have proven to be invaluable tools for identifying modifiers of disease-related protein aggregation and its physiological consequences, including components of the proteostasis network and environmental interventions[9,16,24–27,33,42–49]. Using these models, our study provides insight into the intricate interplay among aging, protein aggregation and age-related neurodegenerative diseases. Specifically,

we demonstrate that elevated EPS-8/RAC signaling during aging promotes aggregation of disease-related proteins in Huntington's disease and ALS *C. elegans* models. Similar to *C. elegans*, we found that lowering EPS8/RAC signaling reduces disease-related changes in human cell lines and iPSC-derived neurons, highlighting the evolutionary conservation of these effects. Although ALS iPSC-derived motor neurons exhibit disease-related alterations, such as increased cell death[50–52], they lack hallmarks of aging[53,54]. This limitation arises because the reprogramming process to generate iPSCs resets cellular age to an embryonic-like state[53,54]. Therefore, although our results demonstrate a role for EPS8/RAC activity in protein aggregation and neurodegeneration in human cells, they cannot provide a direct link between aging and EPS8/RAC signaling in these cellular models.

Our data indicate that EPS8/RAC signaling contributes to protein aggregation through different pathways. EPS8/RAC modulates cellular processes such as actin polymerization and JNK signaling, both of which have been implicated in neurodegenerative diseases[55–58]. We found that excessive actin polymerization, driven by hyperactivated EPS8/RAC signaling, contributes to disease-related protein aggregation. Additionally, elevated EPS8/RAC hyperactivates JNK signaling, further promoting protein aggregation. However, the precise mechanisms by

---

**Fig. 7 | Knockdown of *usp-4* prevents EPS-8 upregulation and disease-related changes during aging in *C. elegans*. a**, Knockdown of *usp-4* ameliorates mutant FUS aggregation in the neurons of day 5 adult *C. elegans* (detected by anti-FUS antibody). Right: SDS-PAGE with antibodies to FUS and α-tubulin. Graphs represent the relative percentage of aggregated and total FUS levels (corrected for α-tubulin) to WT FUS + Vector RNAi (mean ± s.e.m., $n = 3$ independent experiments). **b**, Knockdown of *usp-4* decreases mutant TDP-43$^{M337V}$ aggregation in day 5 adult worms (detected by anti-TDP-43 antibody). Right: SDS-PAGE with antibodies to TDP-43 and α-tubulin. Graphs represent the relative percentage of aggregated and total TDP-43 levels (corrected for α-tubulin) to TDP-43(WT) + Vector RNAi (mean ± s.e.m., $n = 3$ independent experiments). **c**, Percentage of nose touch responses/total trials in WT worms and transgenic worms expressing polyQ19 or polyQ67 in neurons (day 1 (D1): $n = 40$ worms per condition; D5: $n = 80$; D7: $n = 80$). **d**, Chemotaxis index of neuronal polyQ-expressing worms toward 0.5% benzaldehyde (mean ± s.e.m., $n = 3$ independent experiments; 56–215 worms were scored per condition for each independent experiment). **e**, Percentage of nose touch responses/total trials in WT worms and transgenic worms expressing WT FUS or ALS-related mutant FUS$^{R522G}$ and FUS$^{P525L}$ variants (D1: $n = 40$ worms per condition; D5: $n = 80$; D7: $n = 80$). **f**, Chemotaxis index of FUS-ALS worm models toward 0.5% benzaldehyde (mean ± s.e.m., $n = 3$ independent experiments; 78–150 worms were scored per condition for each independent experiment). **g**, Percentage of nose touch responses/total trials in WT worms and transgenic worms expressing WT TDP-43 or ALS-related mutant

TDP-43$^{M337V}$ variant (D1: $n = 30$ worms per condition; D5: $n = 70$; D7: $n = 70$). In **c,e,g**, the box plots represent the 25th–75th percentiles, the lines depict the median and the whiskers show the minimum–maximum values. **h**, Western blot with antibody to EPS-8 of day 10 adult worms on *usp-4* knockdown. RNAi was initiated after development. Graph: relative percentage values of EPS-8 protein levels (corrected for α-tubulin) to Vector RNAi (mean ± s.e.m., $n = 6$ independent experiments). **i**, Knockdown of *usp-4* after development prolongs lifespan in worms expressing WT EPS-8 ($P < 0.01$) but not the short lifespan of Ub-less EPS-8 mutants ($P = 0.6798$). EPS-8(WT) + Vector RNAi: 21.01 days ± 0.46, EPS-8(WT) + *usp-4* RNAi: 23.27 ± 0.40, EPS-8(Ub-less) + Vector RNAi: 17.51 ± 0.55, EPS-8(Ub-less) + *usp-4* RNAi: 18.24 ± 0.48. $P$ values: two-sided log-rank test, $n = 96$ worms per condition. Supplementary Table 1 contains statistical analysis and replicate data of independent lifespan experiments. **j**, Knockdown of *usp-4* prevents polyQ67::YFP aggregation (detected by anti-GFP antibody) in worms expressing WT EPS-8 but not in worms expressing Ub-less EPS-8 mutant. Right: western blot with antibodies to GFP and α-tubulin. Graphs represent the relative percentage values of aggregated polyQ67 and total polyQ67 levels (corrected for α-tubulin loading control) to Q67;EPS-8(WT) + Vector RNAi (mean ± s.e.m., $n = 3$ independent experiments). In all the experiments, RNAi was initiated after development. Statistical comparisons were made by two-way ANOVA with Sidak's multiple comparisons test (**a,c,e–g**), two-way ANOVA with Fisher's LSD test (**b,d,j**), two-tailed Wilcoxon signed-rank test (**h**) and two-sided log-rank test (**i**).

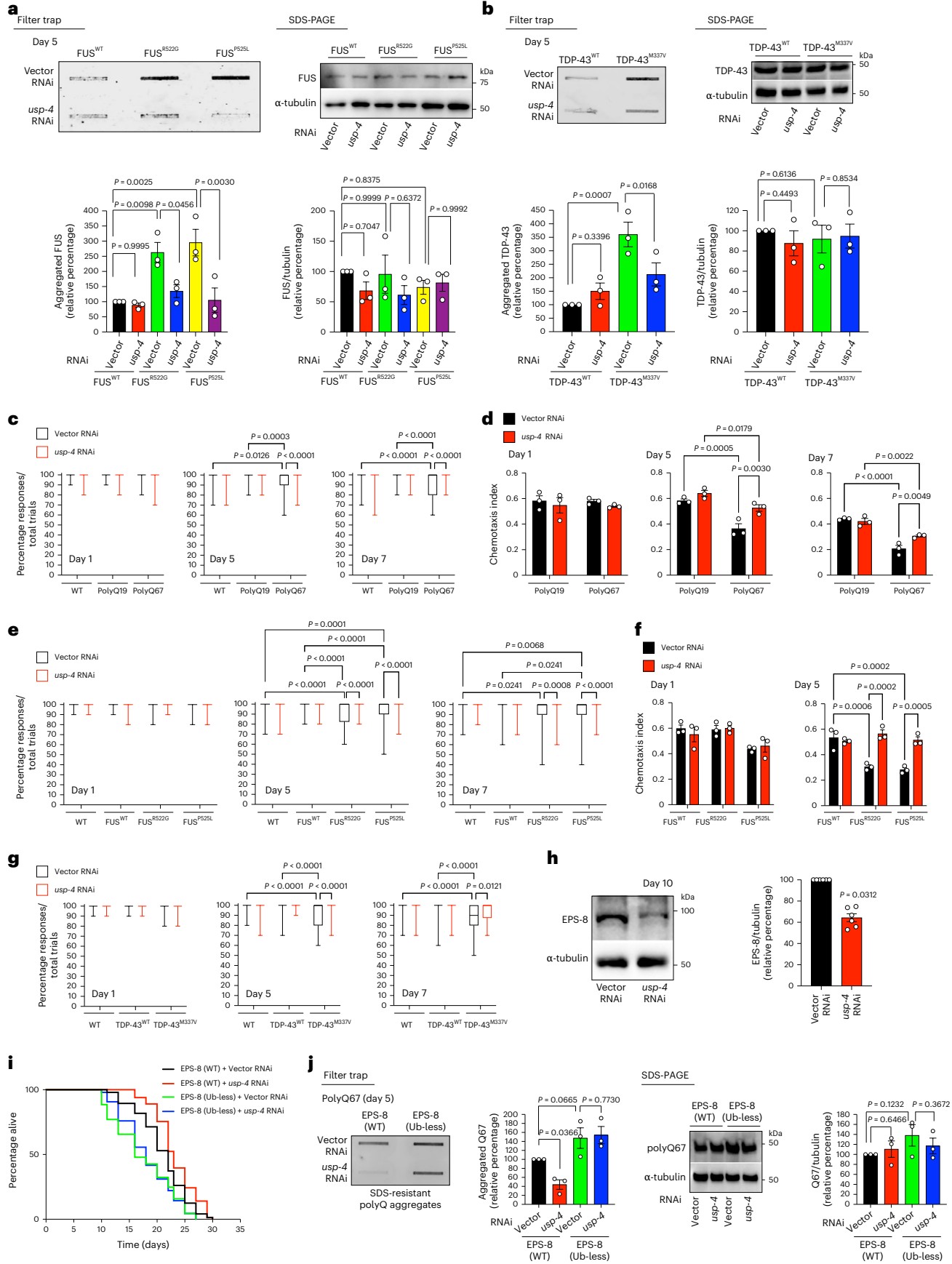

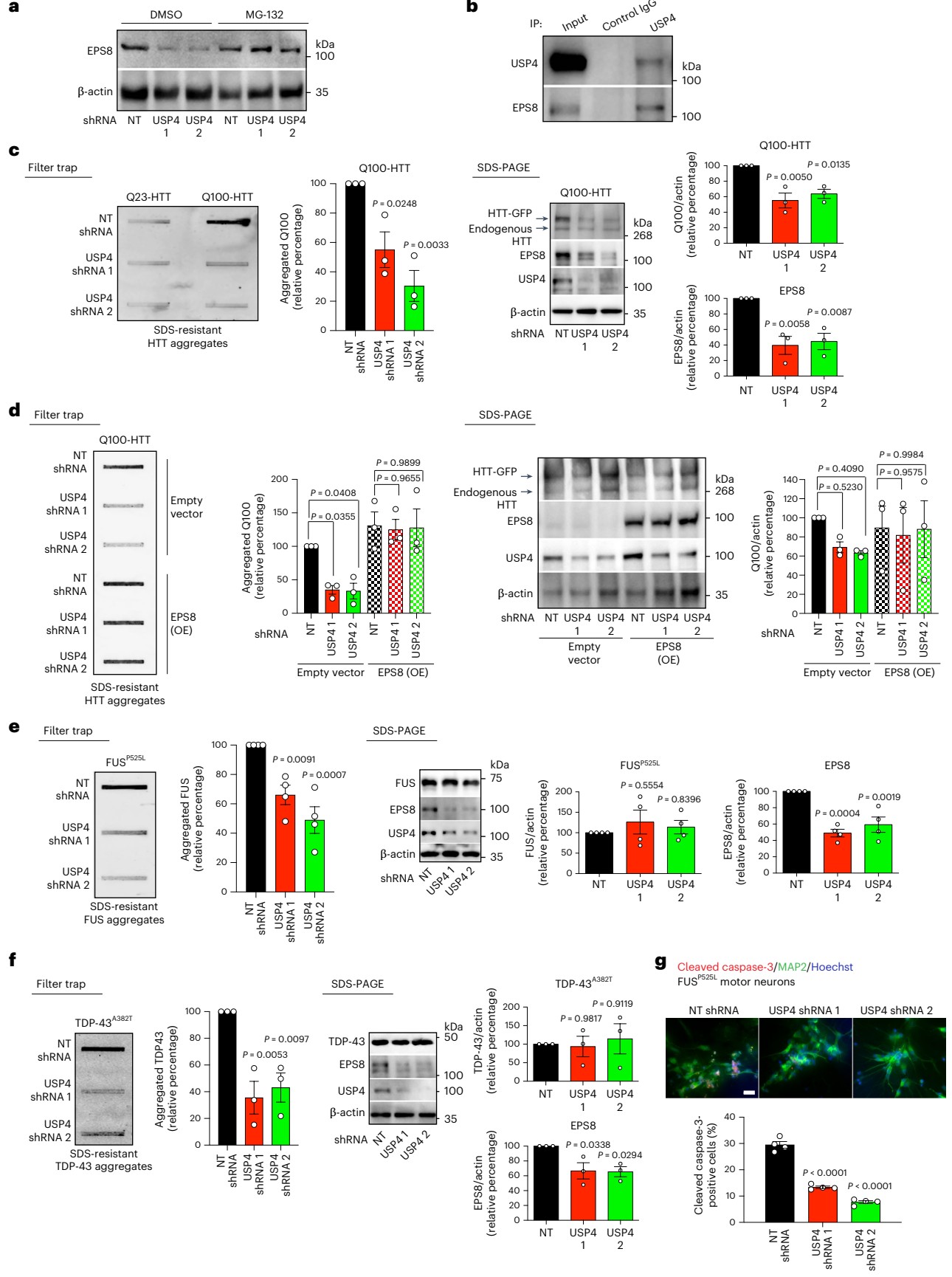

**Fig. 8 | USP4 knockdown decreases EPS8 levels and disease-related protein aggregation in human cells. a**, Western blot analysis of EPS8 levels in HEK293 cells expressing control NT or USP4 shRNA. Cells were treated with 0.5 μM MG-132 proteasome inhibitor or DMSO vehicle control for 16 hours before the lysis. β-Actin is the loading control. Representative of three independent experiments. **b**, Co-IP with control IgG and antibody against USP4 in HEK293 cells. Co-IP was followed by western blot with antibodies to USP4 and EPS8. Representative of two independent experiments. **c**, Filter trap with anti-GFP of HEK293 cells expressing either control Q23-HTT-GFP or aggregation-prone Q100-HTT-GFP upon knockdown of USP4 using two independent shRNAs. Right: SDS-PAGE with antibodies to HTT, EPS8, USP4 and β-actin loading control. Graphs: mean ± s.e.m. relative percentage of aggregated Q100-HTT and total Q100-HTT or EPS8 levels (corrected for β-actin) to NT shRNA Q100-HTT cells (mean ± s.e.m., $n = 3$ independent experiments). **d**, Filter trap of Q100-HTT-GFP aggregation (detected with anti-GFP antibody) in HEK293 cells upon knockdown of USP4 and overexpression of EPS8. Right: SDS-PAGE with antibodies to HTT, EPS8, USP4 and β-actin loading control. Graphs: mean ± s.e.m. relative percentage of aggregated and total Q100-HTT levels (corrected for β-actin) to NT shRNA Q100-HTT cells (mean ± s.e.m., $n = 3$ independent experiments). **e**, Filter trap with anti-FUS of HEK293 cells expressing aggregation-prone FUS[P525L] upon knockdown of USP4. Right: SDS-PAGE with antibodies to FUS, EPS8, USP4 and β-actin loading control. Graphs: mean ± s.e.m. relative percentage of aggregated FUS and total FUS or EPS8 levels (corrected for β-actin) to NT shRNA cells (mean ± s.e.m., $n = 4$ independent experiments). **f**, Filter trap with anti-TDP-43 antibody of HEK293 cells expressing aggregation-prone mutant TDP-43[A382T] upon knockdown of USP4. Right: SDS-PAGE with antibodies to TDP-43, EPS8, USP4 and β-actin loading control. Graphs: mean ± s.e.m. relative percentage of aggregated TDP-43 and total TDP-43 or EPS8 levels (corrected for β-actin) to NT shRNA cells (mean ± s.e.m., $n = 3$ independent experiments). **g**, Immunocytochemistry of FUS(P525L) ALS iPSC-derived motor neurons with anti-cleaved caspase-3 (red), anti-MAP2 (green) and Hoechst (nucleus, blue). Scale bar, 20 μm. Graph represents the percentage of cleaved caspase-3-positive cells/total nuclei (mean ± s.e.m. of four biological replicates from two independent experiments, NT shRNA: 185 total nuclei; USP4 shRNA 1: 149 total nuclei; USP4 shRNA 2: 147 total nuclei). Statistical comparisons were made by one-way ANOVA with Dunnett's multiple comparisons test (**c**,**e**–**g**) and two-way ANOVA with Tukey's multiple comparisons test (**d**).

which excessive actin polymerization and JNK activity drive protein aggregation remain unknown.

Although HTT and ALS-related proteins, including the polyQ-containing protein ataxin-2, regulate actin dynamics[59,60], previous studies indicated that actin filaments and actin-binding factors may also influence pathological protein aggregation[55,61]. For instance, distinct familial ALS cases that exhibit WT TDP-43 aggregates are associated with mutations in actin cytoskeleton regulators such as profilin 1 (ref. 55). We speculate that age-related destabilization of actin filaments may affect protein aggregation by impairing essential cellular processes, thereby reducing the cellular capacity to prevent protein aggregation. In *C. elegans*, knockdown of *anc-1*, which encodes a protein involved in actin binding and cytoskeleton organization, alters the expression of transcription factors and E3 Ub ligases, leading to polyQ aggregation[62].

Importantly, mutant HTT aggregates can co-localize with actin filaments[57]. Although we did not observe changes in the intracellular distribution of mutant HTT aggregates after EPS8 knockdown in human cells, we cannot exclude the possibility that the actin cytoskeleton directly influences aggregation through its interaction with disease-related proteins. For instance, redistribution of the intermediate protein vimentin contributes to the assembly of aggresomes containing cystic fibrosis transmembrane conductance regulator, whereas disruption of microtubules blocks aggresome formation[63]. In previous work, we observed that age-related changes in actin filaments lead to aggregation of actin protein itself[14]. This raises the intriguing possibility that actin aggregates may act as a niche for the accumulation of disease-related proteins. Alternatively, actin aggregates could sequester molecular chaperones and other components of the proteostasis network, leading to its collapse and subsequent aggregation of pathological proteins.

Likewise, hyperactivation of JNK may influence pathological protein aggregation through different mechanisms. The JNK pathway is involved in the response to proteotoxic stresses, such as heat and oxidative stress. Moreover, JNK triggers phosphorylation cascades that modulate distinct regulatory proteins in the mitochondria and nucleus, including SMAD4, p53, c-JUN, ATF2, ELK1 and HSF1 (refs. 64,65). Thus, JNK hyperactivation during aging may lead to cellular alterations, promoting protein aggregation. In addition, these downstream targets of JNK signaling could directly affect the activity of proteostasis mechanisms. Beyond elevated actin polymerization and JNK activity, we cannot discard that other mechanisms regulated by EPS8/RAC signaling contribute to protein aggregation. For instance, RAC regulates additional pathways, including p38 MAPK, PI3K/Akt/mTOR and STAT signaling[41,66,67]. Moreover, RAC influences reactive oxygen species production[68], which could play a role in pathological aggregation.

Our study identified the DUB USP4 as a key regulator of EPS8 ubiquitination and degradation. We found that USP4 knockdown not only decreases EPS8 levels but also prevents aggregation of polyQ-expanded and ALS-related mutant proteins in both *C. elegans* and human cells. Notably, a previous study demonstrated that loss of this DUB also protects against paralysis induced by aggregation of human amyloid-β in the muscle of worm models[69].

By uncovering the role of EPS8/RAC signaling and its regulation by USP4, we provide insights that may contribute to the development of targeted therapies to prevent or delay distinct age-related neurodegenerative diseases. To further evaluate therapeutic implications, it will be interesting to explore whether EPS8 also influences pathological protein aggregation in mammalian models.

## Methods

### *C. elegans* strains

*C. elegans* strains were cultured at 20 °C on standard Nematode Growth Medium seeded with OP50 *Escherichia coli*[70]. On day 1 of adulthood, worms were transferred to plates containing OP50 *E. coli* (or HT115 *E. coli* for RNAi experiments) supplemented with 100 μg ml$^{-1}$ 5-fluoro-2′-deoxyuridine to prevent progeny development, except in lifespan assays. All experiments were conducted using hermaphrodite worms, and the age of the worms is indicated in the corresponding figures and figure legends.

WT (N2) and AM141 (*rmIs133*[*unc-54p*::Q40::YFP]) strains were obtained from the *Caenorhabditis* Genetics Center (CGC), supported by the National Institutes of Health Office of Research Infrastructure Programs (P40 OD010440). RB751 (*eps-8(ok539)*) was generated by the *C. elegans* Gene Knockout Consortium and acquired from the CGC. AM23 (*rmIs298*[*F25B3.3p*::Q19::CFP]) and AM716 (*rmIs284*[*F25B3.3p*::Q67::YFP]) strains were gifted by Richard I. Morimoto[24]. MAH602 (*sqIs61*[*vha-6p*::Q44::YFP + *rol-6*(*su1006*)]) was provided by Malene Hansen[71]. ZM5838 (*hpIs223*[*rgef-1p*::FUS[WT]::GFP]), ZM5844 (*hpIs233*[*rgef-1p*::FUS[P525L]::GFP]) and ZM5842 (*hpIs228*[*rgef-1p*::FUS[R522G]::GFP]) were provided by Peter St. George-Hyslop[45]. CK405(*Psnb-1*::TDP-43[WT], *myo-2p*::dsRED) and CK423 (*Psnb-1*::TDP-43[M337V], *myo-2p*::dsRED) were provided by Brian C. Kraemer[33].

From these strains, we generated NFB2862 (*Psnb-1*::TDP-43[WT], *myo-2p*::dsRED;*juIs76*[*unc-25p*::GFP + *lin-15*(+)]II) and NFB2863 (*Psnb-1*::TDP-43[M337V],*myo-2p*::dsRED;*juIs76*[*unc-25p*::GFP + *lin-15*(+)]II). NFB2858 (*rmIs298*[*F25B3.3p*::Q19::CFP];*otIs549*[*unc-25p*::*unc-25*(partial)::mChopti::*unc-54* 3′ untranslated region (UTR) + *pha-1*(+)];*him-5*(*e1490*)V), NFB2859 (*rmIs284*[*F25B3.3p*::Q67::YFP];*otIs549*[*unc-25p*::*unc-25* (partial)::mChopti::*unc-54* 3′ UTR + *pha-1*(+)];*him-5*(*e1490*)V), NFB2860 (*hpIs223*[*rgef-1p*::FUS[WT]::GFP];*otIs549*[*unc-25p*::*unc-25*

(partial)::mChopti::*unc-54* 3' UTR + *pha-1*(+)];*him-5*(*e1490*)V) and NFB2861 (*hpIs233*[*rgef-1p*::FUS$^{P525L}$::GFP];*otIs549*[*unc-25p*::*unc-25* (partial)::mChopti::*unc-54* 3' UTR + *pha-1*(+)];*him-5*(*e1490*)V) were generated by crossing the respective polyQ and FUS-expressing strains with the OH13526 strain[72]. For RNAi in the neurons of polyQ67 worms, we used the DVG196 strain (*rmIs284*[*F25B3.3p*::Q67::YFP];*sid-1*(*pk3321*)V; uIs69[pCFJ90(*myo-2p*::mCherry) + *unc-119p*::*sid-1*]).

Worms expressing endogenous WT EPS-8::3xHA (VDL05, *eps-8*(*syb2901*)IV) or mutant EPS-8(K524R/K583R/K621R::3×HA) (VDL06, *eps-8*(*syb2901*, *syb3149*)IV) were previously generated via CRISPR–Cas9 (ref. [14]). The strains DVG344 (*rmIs284*[*pF25B3.3*::Q67::YFP]);*eps-8*(*syb2901*) and DVG363 (*rmIs133*[*unc-54p*::Q40::YFP]);*eps-8* (*syb2901*) were generated by crossing VDL05 with AM716 and AM141, respectively. DVG345 (*rmIs284*[*pF25B3.3*::Q67::YFP]);*eps-8*(*syb2901*, *syb3149*) and DVG364 (*rmIs133*[*unc-54p*::Q40::YFP]);*eps-8*(*syb2901*, *syb3149*) were generated by crossing VDL06 to AM716 and AM141, respectively. These strains were validated by sequencing using the following primers: *eps-8*(*syb2901*): 5'-TTTGTTCGAAGCATGAACGA-3' and 5'-AGCAGCCCCTGAAATAGTGA-3'; *eps-8*(*syb2901*, *syb3149*): 5'-AACG AGCTAGCAATCCGAAA-3' and 5'-AGTGCTCTGCCGTCATTAAT-3'. DVG365 (*rmIs284*[*pF25B3.3*::Q67::YFP]);*eps-8*(*ok539*)) was generated by crossing RB751 to AM716. The strain was outcrossed two times to AM716 and validated by polymerase chain reaction with 5'-TCTCCACCACCA CAACGTAA-3' and 5'-GCGGAGCAACTCTTCCATAG-3' primers.

## RNAi constructs

Adult worms were fed HT115 *E. coli* carrying either an empty control vector (L4440) or vectors expressing double-stranded RNAi. The RNAi constructs targeting *eps-8*, *ifb-2*, *jnk-1*, *kgb-1*, *mig-2* and *otub-3* were obtained from the Vidal library. The *csn-6*, *F07A11.4*, *math-33*, *rac-2*, *usp-4*, *usp-5* and *usp-48* RNAi constructs were obtained from the Ahringer library. All RNAi constructs were sequence verified. The RNAi sequences are listed in Supplementary Table 2.

## Lifespan assay

Larvae were synchronized using the egg-laying protocol and grown on OP50 *E. coli* at 20 °C until day 1 of adulthood. Adult hermaphrodites were then transferred to plates with HT115 *E. coli* carrying either an empty vector or RNAi constructs for lifespan assays. All lifespan assays were performed at 20 °C. Each condition included 96 worms, scored daily or every other day[73]. Worms that were lost, burrowed into the medium, had a protruding vulva or underwent bagging were censored[73].

## Nose touch assay

Age-synchronized worms were assessed for nose touch response as previously described[74–76]. In brief, worms were placed on a thin bacterial lawn, and an eyelash pick was positioned in front of a forward-moving animal. A lack of response was recorded when the worm continued moving forward to crawl under or over the pick. For each condition, 30–40 animals were tested by monitoring the number of responses to a total of 10 gentle eyelash touches.

## Chemotaxis assay

Freshly prepared agar plates (2% agar, 5 mM KPO$_4$ (pH 6.0), 1 mM CaCl$_2$, 1 mM MgSO$_4$) were divided into four equal quadrants, along with an inner circle measuring approximately 1 cm across diagonally. A test solution (0.5% benzaldehyde (Sigma-Aldrich, B1334) in ethanol + 0.25 M sodium azide) and a control solution (ethanol + 0.25 M sodium azide) were added to two opposing diagonal quadrants. On the indicated days of adulthood (as shown in the corresponding figures), worms were collected in S-Basal medium, washed three times to remove residual bacteria and placed at the center of the chemotaxis plate. The plates were sealed with parafilm and incubated at 20 °C for 90 minutes. The number of worms in each quadrant was counted, excluding those that did not cross the inner circle. The chemotaxis index was calculated using the following formula: chemotaxis index = ((number of animals in test quadrants) − (number of animals in control quadrants)) / total number of animals[77].

## Motility assays

*C. elegans* were synchronized on OP50 *E. coli* using the egg-laying method and grown until day 1 of adulthood and then randomly transferred to plates with HT115 *E. coli* containing either empty vector or RNAi for the remainder of the experiment. For experiments with Ub-less EPS-8 mutants or DUB inhibitor treatment, worms were instead transferred to fresh plates containing OP50 *E. coli*. On the indicated day of adulthood (as shown in the corresponding figures), worms were randomly picked and transferred to a drop of M9 buffer, allowing 30 seconds for recovery[24]. Body bends were then recorded for 30 seconds and analyzed using ImageJ software (version 1.53k) with the wrMTrck plugin (https://www.phage.dk/plugins/)[78,79]. The locomotion velocity data were used to calculate body bends per second.

## Microscopy

For imaging GABAergic neurons, fluorescent reporter worms were anesthetized with a drop of 0.5 M sodium azide (Sigma-Aldrich, 26628-22-8) on 4% agarose pads (diluted in distilled water) placed over a standard microscope glass slide (Rogo-Sampaic, 11854782). These preparations were sealed with 24 × 60-mm coverslips (RS France, BPD025). To score the number of GABAergic neurons and ventral nerve cord projections, we used a Zeiss Axio Imager.M2 microscope with a ×40 objective. Whole-body worm images were acquired using a Leica THUN-DER Imager microscope with Tile Scan function and a ×40 objective.

## Human cell lines

HEK293T/17 cells (American Type Culture Collection (ATCC), CRL-11268) were plated on 0.1% gelatin-coated plates and grown in DMEM (Thermo Fisher Scientific, 11966025), supplemented with 1% MEM non-essential amino acids (Thermo Fisher Scientific, 11140035), 1% GlutaMAX (Life Technologies, 35050038) and 10% FBS (Thermo Fisher Scientific, 10500064) at 37 °C with 5% CO$_2$. ALS-iPSCs (FUS$^{P525L/P525L}$) were kindly provided by Irene Bozzoni and Alessandro Rosa[37]. iPSCs were cultured on Geltrex (Thermo Fisher Scientific, A1413302) using mTeSR1 medium (STEMCELL Technologies, 85850) at 37 °C with 5% CO$_2$. All cell lines were routinely tested for mycoplasma contamination, and no contamination was detected.

## Motor neuron differentiation

Motor neurons were derived from ALS-iPSCs using a monolayer-based differentiation protocol[80]. ALS-iPSCs were seeded on Geltrex-coated plates and maintained in mTeSR1 medium until confluent. Differentiation was initiated using neuron differentiation medium composed of DMEM/F12 and Neurobasal (1:1; Thermo Fisher Scientific, 11330057 and 21103049), supplemented with non-essential amino acids, GlutaMAX (Thermo Fisher Scientific, 35050038), B27 (Thermo Fisher Scientific, 12587010) and N2 (Thermo Fisher Scientific, 17502048).

From day 0 to day 6, the medium was further supplemented with 1 μM retinoic acid (Sigma-Aldrich, R2625), 1 μM smoothened agonist (SAG; Sigma-Aldrich, 566661), 0.1 μM LDN-193189 (Miltenyi Biotec, 130-103-925) and 10 μM SB-431542 (Miltenyi Biotec, 130-105-336). From day 7 to day 14, the neuron differentiation media were supplemented with 1 μM retinoic acid, 1 μM SAG, 4 μM SU-5402 (Sigma-Aldrich, SML0443) and 5 μM DAPT (Sigma-Aldrich, D5942). After day 14, differentiated motor neurons were dissociated and replated on poly-L-ornithine (Sigma-Aldrich, P3655) and laminin-coated (Thermo Fisher Scientific, 23017015) plates in Neurobasal medium, supplemented with non-essential amino acids, GlutaMAX, N2, B27 and neurotrophic factors (10 ng ml$^{-1}$ BDNF (BIOZOL, 450-02) and 10 ng ml$^{-1}$ GDNF (BIOZOL, 450-10)).

## Lentiviral infection of human cells

Lentivirus (LV)-non-targeting short hairpin RNA (shRNA), LV-EPS8 shRNA 1 (TRCN0000061544), LV-EPS8 shRNA 2 (TRCN0000061545), LV-USP4 shRNA 1 (TRCN0000004039) and LV-USP4 shRNA 2 (TRCN0000004040) in the pLKO.1-puro backbone were obtained from Mission shRNA (Sigma-Aldrich). Supplementary Table 2 contains the target sequences of each shRNA construct.

To generate stable shRNA-expressing HEK293 cell lines, cells were transduced with 5 μl of concentrated lentivirus and selected with 2 μg ml⁻¹ puromycin (Thermo Fisher Scientific, A1113803). For lentiviral infection of iPSCs, cells were dissociated using Accutase (Thermo Fisher Scientific, A1110501), and 100,000 cells were seeded on Geltrex-coated plates in mTeSR1 medium supplemented with 10 μM ROCK inhibitor for 1 day. The next day, cells were infected with 5 μl of concentrated lentivirus. Medium was replaced the following day to remove residual virus. Selection for lentiviral integration was performed using 2 μg ml⁻¹ puromycin for 2 days.

## Transfection of HEK293 cells

HEK293 cells (ATCC, CRL-11268) were seeded on 0.1% gelatin-coated plates. When cells reached approximately 40% confluency, they were transfected with 1 μg of one of the following plasmids using FuGENE HD (Promega), according to the manufacturer's instructions: pARIS-mCherry-httQ23-GFP, pARIS-mCherry-httQ100-GFP, pLVX-Puro-TDP-43-WT, pLVX-Puro-TDP-43-A382T, pcDNA3.1-FUS-HA-WT or pcDNA3.1-FUS-HA-P525L. In the indicated experiments, cells were co-transfected with an additional 1 μg of the pCMV3-EPS8-HA plasmid. The cells were collected after 72 hours of incubation in standard medium. The pARIS-mCherry-httQ23-GFP and pARIS-mCherry-httQ100-GFP plasmids were generously provided by Frédéric Saudou[81]. The FUS-HA-WT and FUS-HA-P525L plasmids were a gift from Dorothee Dormann[82]. The pLVX-Puro-TDP-43-WT and pLVX-Puro-TDP-43-A382T plasmids were provided by Shawn Ferguson (Addgene, 133753 and 133756)[83]. The pCMV3-EPS8-HA plasmid was obtained from Sino Biological (HG11153-CY).

## Filter trap and western blot

For filter trap assays, synchronized adult *C. elegans* were collected and washed with M9 buffer, and worm pellets were snap frozen in liquid nitrogen. Frozen pellets were thawed on ice and lysed in non-denaturing buffer (50 mM HEPES (pH 7.4), 150 mM NaCl, 1 mM EDTA, 1% Triton X-100, 2 mM sodium orthovanadate, 1 mM PMSF, protease inhibitor cocktail (Roche)) using a Precellys 24 homogenizer. Lysates were cleared of worm debris by centrifugation (8,000$g$, 5 minutes, 4 °C), and protein concentrations were determined using the BCA assay (Thermo Fisher Scientific). To assess protein levels by western blot, 30 μg of total protein was separated by SDS-PAGE and transferred to polyvinylidene difluoride membranes (Millipore). To assess aggregated proteins by filter trap, 100 μg of total protein was supplemented with SDS to a final concentration of 0.5% and loaded onto a cellulose acetate membrane assembled in a slot-blot apparatus (Bio-Rad). Then, the membrane was washed with 0.2% SDS, and SDS-resistant aggregates were detected by immunoblotting.

If lysates were used solely for western blot, worms were lysed with a Precellys 24 homogenizer in buffer containing 50 mM Tris-HCl (pH 7.8), 150 mM NaCl, 1% Triton X-100, 0.25% sodium deoxycholate, 1 mM EDTA, 25 mM N-ethylmaleimide, 2 mM sodium orthovanadate, 1 mM PMSF and protease inhibitor cocktail. Lysates were cleared at 10,600$g$ for 10 minutes at 4 °C, and 30 μg of protein was used for western blot experiments. For analysis of polyQ monomers and SDS-insoluble polyQ aggregates, age-synchronized worms were lysed by sonication in native buffer (50 mM Tris (pH 8), 150 mM NaCl, 5 mM EDTA, 1 mM PMSF, protease inhibitor cocktail). Then, 30 μg of total protein was mixed with SDS to a final concentration of 0.4% and resolved by 12.5% SDS-PAGE.

For both filter trap and western blot analyses of *C. elegans*, immunoblotting was performed with antibodies against GFP (AMSBIO, TP401, dilution 1:5,000), FUS (Abcam, ab154141, clone CL0190, 1:1,000) and TDP-43 (Abcam, ab225710, 1:1,000). Additionally, for western blot experiments, immunoblotting was conducted with anti-EPS8L2 (Abcam, ab85960, 1:1,000), anti-LGG-1 (ref. [84], 1:2,000) and α-tubulin (Sigma-Aldrich, T6199, 1:5,000).

For filter trap and western blot analysis of HEK293 cell lines, the cells were collected in lysis buffer (50 mM HEPES (pH 7.4), 150 mM NaCl, 1 mM EDTA, 1% Triton X-100, 2 mM sodium orthovanadate, 1 mM PMSF, protease inhibitor cocktail), followed by homogenization through a 27-gauge syringe needle. Lysates from cells expressing pARIS-mCherry-httQ23-GFP, pARIS-mCherry-httQ100-GFP or without any overexpression were centrifuged at 8,000$g$ for 5 minutes at 4 °C. Lysates from cells expressing FUS-HA-WT, FUS-HA-P525L, pLVX-Puro-TDP-43-WT or pLVX-Puro-TDP-43-A382T were centrifuged at 1,000$g$ for 5 minutes at 4 °C. The supernatants were collected, and protein concentrations were measured with the BCA assay. For western blot, 30 μg of protein was analyzed as above. For filter trap analysis, 100 μg of total protein was supplemented with SDS to a final concentration of 0.5% and loaded onto a cellulose acetate membrane assembled in a slot-blot apparatus as described above. The membrane was then washed with 0.2% SDS, and SDS-resistant protein aggregates were evaluated by immunoblotting. For filter trap analysis, immunoblotting was conducted with antibodies against GFP (AMSBIO, TP401, 1:5,000), FUS (Abcam, ab154141, clone CL0190, 1:1,000) and TDP-43 (Abcam, ab225710, 1:1,000). For western blot, immunoblotting was conducted with anti-EPS8 (Proteintech, 12455-1-AP, 1:1,000), anti-β-actin (Abcam, 8226, 1:5,000), anti-HTT (Cell Signaling Technology, 5656, 1:1,000), FUS (Abcam, ab154141, clone CL0190, 1:1,000), TDP-43 (Abcam, ab225710, 1:1,000), anti-LC3B (Cell Signaling Technology, 2775, 1:1,000) and anti-USP-4 (Abcam, ab181105, 1:1,000).

For necroptosis analysis, iPSC-derived motor neurons were lysed in RIPA buffer (50 mM Tris-HCl (pH 7.4), 150 mM NaCl, 1% Triton X-100, 1% sodium deoxycholate, 0.1% SDS, 1 mM EDTA, 1 mM PMSF, protease inhibitor cocktail). Immunoblotting was performed using anti-phospho-RIP (Ser166) (Cell Signaling Technology, 65746, clone D1L3S, 1:1,000) and anti-RIP (Cell Signaling Technology, 3493, clone D94C12, 1:1,000). Densitometry of filter trap and western blot assays was performed using ImageJ software (version 1.51).

## Protein immunoprecipitation for interaction analysis

HEK293 cells were collected and lysed in a protein lysis buffer containing 50 mM Tris-HCl (pH 6.7) 150 mM NaCl, 1% NP40, 0.25% sodium deoxycholate, 1 mM EDTA, 1 mM PMSF, 1 mM sodium orthovanadate, 1 mM NaF and protease inhibitor cocktail. Lysates were homogenized through a 27-gauge syringe needle and centrifuged at 13,000$g$ for 15 minutes at 4 °C. Supernatants were incubated on ice for 1 hour with anti-USP-4 antibody (Abcam, ab181105, 1:100). As a negative control, the same amount of protein was incubated with anti-normal rabbit IgG (Cell Signaling Technology, 2729, 1:378). Samples were then incubated with 50 μl of μMACS MicroBeads for 1 hour at 4 °C with overhead shaking. Then, the samples were loaded onto pre-cleared μMACS columns (Miltenyi Biotec, 130-042-701). The beads were washed three times with a buffer containing 50 mM Tris (pH 7.5), 150 mM NaCl, 5% glycerol and 0.05% Triton, followed by five washes with 50 mM Tris (pH 7.5) and 150 mM NaCl. The samples were eluted with 75 μl of boiled 2× Laemmli buffer, boiled for 5 minutes at 95 °C and analyzed by western blotting.

## Native gels analysis

HEK293 cells expressing CMV:mRFP-Q74 (ref. [30]) were lysed in buffer containing 50 mM Tris-HCl (pH 7.4), 150 mM NaCl, 0.5% NP-40, 2 mM EDTA, 1 mM EGTA, 10% glycerol, 2 mM sodium orthovanadate, 1 mM PMSF and protease inhibitor cocktail. Lysates were homogenized using

a 27-gauge syringe needle and centrifuged at 12,000$g$ for 15 minutes at 4 °C. Supernatants were collected, and protein concentrations were determined using the BCA protein assay (Thermo Fisher Scientific). Equal amounts of protein lysates were mixed 1:1 with sample buffer (50 mM Tris-HCl (pH 6.8), 10% glycerol, 0.01% bromophenol blue). Then, 20 µg of total protein was separated using 4–15% Tris-Glycine eXtended protein gels (Bio-Rad) and imaged via fluorescence using LICOR Odyssey M.

### Immunocytochemistry
Cells were fixed with 4% paraformaldehyde in PBS for 20 minutes, followed by permeabilization with 0.2% Triton X-100 in PBS (10 minutes) and blocking with 3% BSA in 0.2% Triton X-100 in PBS (10 minutes). The cells were then incubated with anti-MAP2 (Sigma-Aldrich, M1406, 1:300) and rabbit anti-cleaved caspase-3 (Cell Signalling Technology, 9661S, 1:300) for 2 hours at room temperature. After washing with PBS, cells were incubated with secondary antibodies (Alexa Fluor 488 goat anti-mouse (Thermo Fisher Scientific, A-11029, 1:500) and Alexa Fluor 568 F(ab')2 fragment of goat anti-rabbit IgG (H + L) (Thermo Fisher Scientific, A-21069, 1:500)) and Hoechst 33342 (Life Technologies, 1656104) for 1 hour at room temperature. Finally, the coverslips were rinsed in PBS, followed by a distilled water wash, and then mounted onto microscope slides with FluorSave Reagent (Merck, 345789).

### CytoD, RAC activator and DUB inhibitor treatment
For CytoD treatment, worms were collected and randomly divided equally into M9 solutions containing either 10 µM CytoD (STEMCELL Technologies, 100-0557) or DMSO as a vehicle control. The worms were incubated with CytoD or DMSO for 6 hours on a shaker. For DUB inhibitor experiments, worms were collected and randomly transferred onto plates with OP50 bacteria covered with a final concentration of 13.7 µg ml$^{-1}$ PR-619 (Merck, 662141) or vehicle control (DMSO) for either 4 hours or 1 day as indicated in the corresponding figures.

HEK293 cells were treated with 2 µM CytoD or DMSO for 4 hours before lysis. For RAC activation, cells were treated with 2 U ml$^{-1}$ Rac/Cdc42 Activator II (Cytoskeleton, CN02-A) for 6 hours.

### Proteasome activity
Day 5 adult worms and HEK293 cells were lysed in proteasome activity assay buffer (50 mM Tris-HCl (pH 7.5), 10% glycerol, 5 mM MgCl$_2$, 0.5 mM EDTA, 2 mM ATP, 1 mM DTT) using a Precellys 24 or a 27-gauge syringe, respectively. The samples were centrifuged at 10,000$g$ for 10 minutes at 4 °C, and the supernatants were collected. Protein concentrations were determined using the BCA protein assay kit.

To measure chymotrypsin-like proteasome activity, 25 µg of total protein was incubated with the fluorogenic substrate Suc-Leu-Leu-Val-Tyr-AMC (Enzo Life Sciences, BML-P802) in 96-well plates (BD Falcon). Fluorescence was measured every 5 minutes for 2 hours at 20 °C (*C. elegans*) or 37 °C (human cells) using a microplate fluorometer (PerkinElmer, EnSpire) at 380-nm excitation and 460-nm emission.

### Statistics and reproducibility
For quantification of filter trap and western blot data, we presented the results as relative changes compared with the corresponding control conditions. To average and statistically analyze independent experiments for these assays, we normalized test conditions to their corresponding control groups measured concurrently in each replicate experiment. Given that all the control groups were set to 100, we used a non-parametric Wilcoxon test when comparing two conditions to assess changes in protein aggregation and protein levels. For all other assays, we used parametric tests. Data distribution was assumed to be normal, but this was not formally tested. When more than two conditions or two independent variables were compared, we used one-way or two-way ANOVA followed by multiple comparisons tests. All statistical analyses were performed using GraphPad Prism (version 10.4.1).

For lifespan experiments, we used GraphPad Prism (version 10.4.1) and OASIS (version 1)[85] to determine median and mean lifespan, respectively. The $P$ values were calculated using the log-rank (Mantel–Cox) method and refer to comparisons between experimental and control animals within a single lifespan experiment. Each lifespan graph represents a single, representative experiment. Supplementary Table 1 contains the number of total/censored worms as well as detailed statistical analyses for each replicate lifespan experiment.

No statistical methods were used to predetermine sample size, but our sample sizes are similar to, or greater than, those reported in previous publications using the same procedures[9,14,16,26,30,33,44,46,50,73,75,76,78,86–88]. For motility assays, worms were excluded from analysis if they showed fewer than 0.1 body bends per second or were not recognized by the ImageJ software. No animals or data points were excluded from other analyses. For lifespan assays, worms were randomly picked and transferred from the synchronized population to the different experimental conditions. For all other experiments, worms were randomly distributed into the various experimental groups from single pulls of synchronized populations. Human cells were distributed to the various groups of all experiments from single pulls. Data collection was not randomized. Data collection and analysis were not performed blinded to the conditions of the experiments.

### Reporting summary
Further information on research design is available in the Nature Portfolio Reporting Summary linked to this article.

## Data availability
The authors declare that all data supporting the findings of this study are available within the paper and its Supplementary Information files. Source data are provided with this paper.

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

## Acknowledgements

This work was supported by the Deutsche Forschungsgemeinschaft (DFG) (VI742/4-2 to D.V.; CRC1678 (project B06) to D.V.; and Germany's Excellence Strategy-CECAD, EXC 2030-390661388); by the Fritz Thyssen Foundation (0.22.2.030MN to D.V.); by the Center for Molecular Medicine Cologne (project C16 to D.V.); by the Generalitat Valenciana (CIAICO/2022/195 to N.F.); and by the Spanish Government (PID2020-115635RB-I00 to N.F.). The funders had no role in study design, data collection and analysis, preparation of the manuscript or decision to publish. We are grateful to A. Rosa and I. Bozzoni for providing the ALS-iPSCs and to T. Hoppe for the anti-LGG-1 antibody.

## Author contributions

S.K. and D.V. conceived of and supervised the study. They also designed the experiments. S.K. performed most of the experiments and data analysis. Y.D.-C. and R.A. performed the analysis of neurodegeneration in *C. elegans*. N.S. assisted with lifespan assays. D.P. generated polyQ strains in mutant backgrounds and contributed to other experiments. N.F. contributed with her expertise in *C. elegans* neurodegeneration and behavior assays. D.V. wrote the paper. All authors revised and approved the paper.

## Funding

## Competing interests

The authors declare no competing interests.

## Additional information

**Extended data** is available for this paper at https://doi.org/10.1038/s43587-025-00943-w.

**Correspondence and requests for materials** should be addressed to Seda Koyuncu or David Vilchez.

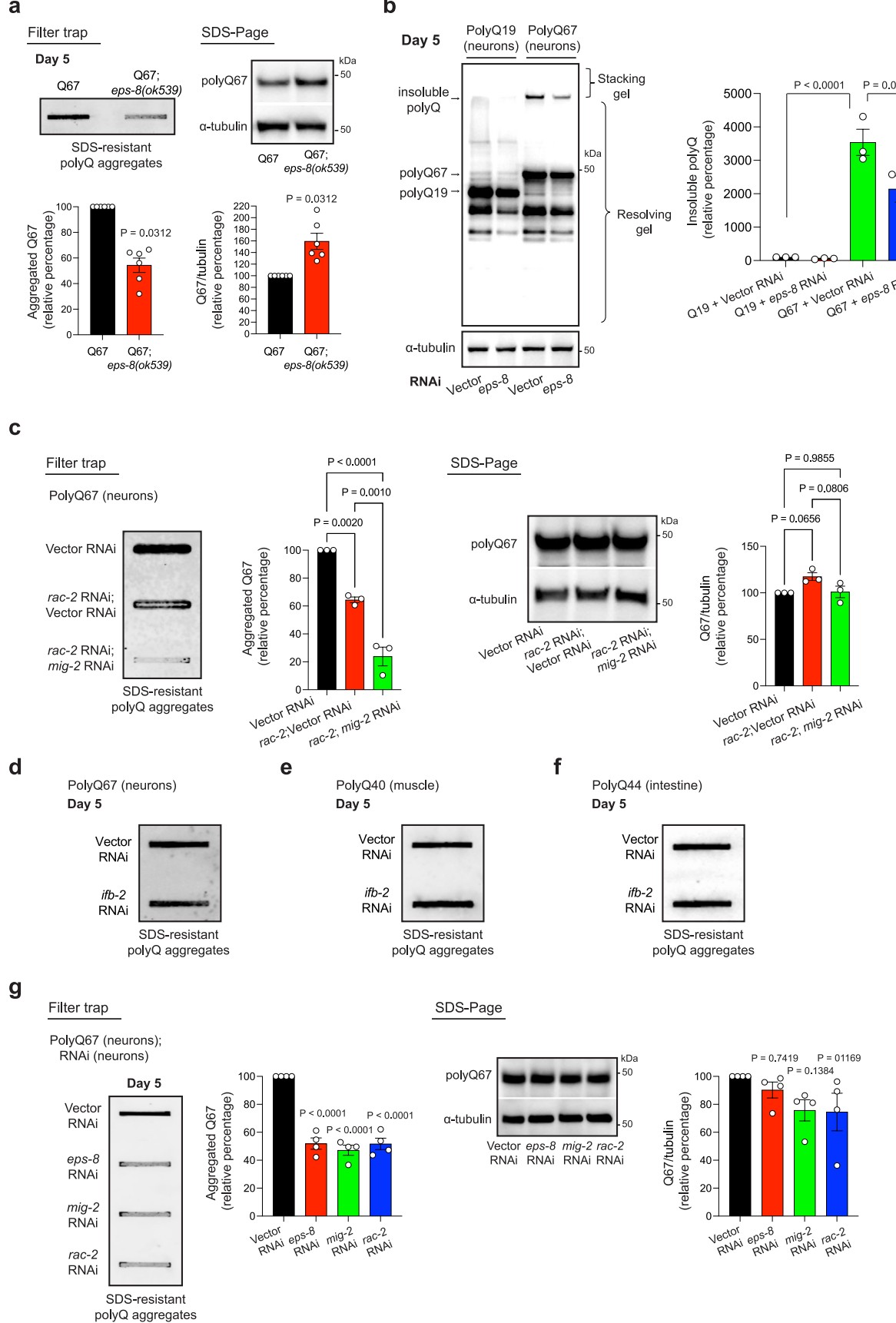

**Extended Data Fig. 1 | See next page for caption.**

**Extended Data Fig. 1 | The age-dysregulated proteasome target EPS-8 promotes polyQ-expanded aggregation in *C. elegans*. a**, *eps-8* knockout reduces polyQ aggregation. Left: Filter trap assay with anti-GFP in day-5 adult worms. Right: SDS–PAGE with antibodies to GFP and α-tubulin loading control. Graphs represent the relative percentage values of aggregated polyQ67 and total polyQ67 levels (corrected for α-tubulin) to Q67 (mean ± s.e.m., $n$ = 6 independent experiments). **b**, Western blot analysis of day-5 adult worms expressing polyQ19::YFP or polyQ67::YFP peptides in neurons, using antibodies against GFP and α-tubulin loading control. Worms were treated with RNAi after development. Graph represents the relative percentage values of insoluble polyQ levels (corrected for α-tubulin) to Q19 + Vector RNAi (mean ± s.e.m., $n$ = 3 independent experiments). **c**, Filter trap assay with anti-GFP in day-5 adult polyQ67::YFP worms following RNAi treatment against *RAC* orthologs after development. Right: SDS–PAGE with antibodies to GFP and α-tubulin. Graphs represent the relative percentage values of aggregated polyQ67 and total polyQ67 levels (corrected for α-tubulin) to Vector RNAi (mean ± s.e.m., $n$ = 3

independent experiments). **d**, Knockdown of IFB-2 after development does not reduce aggregation of polyQ-expanded peptides in the neurons of day-5 adult worms. Representative of 7 independent experiments. **e**, Knockdown of IFB-2 after development does not decrease polyQ-expanded aggregation in the muscle of day-5 adult worms. Representative of 3 independent experiments. **f**, Knockdown of IFB-2 after development does not decrease polyQ-expanded aggregation in the intestine of day-5 adult worms. Representative of 3 independent experiments. **g**, Filter trap of neuronal polyQ67::YFP aggregation with anti-GFP in day-5 adult worms upon neuronal knockdown of *eps-8* and *RAC* orthologs. Right: SDS–PAGE with antibodies to GFP and α-tubulin. Graphs represent the relative percentage values of aggregated polyQ67 and total polyQ67 levels (corrected for α-tubulin) to Vector RNAi (mean ± s.e.m., $n$ = 4 independent experiments). Statistical comparisons were made by two-tailed Wilcoxon signed-rank test (**a**), two-way ANOVA with Fisher's LSD test (**b**), one-way ANOVA with Tukey's multiple-comparison test (**c**), and one-way ANOVA with Dunnett's multiple-comparison test (**g**).

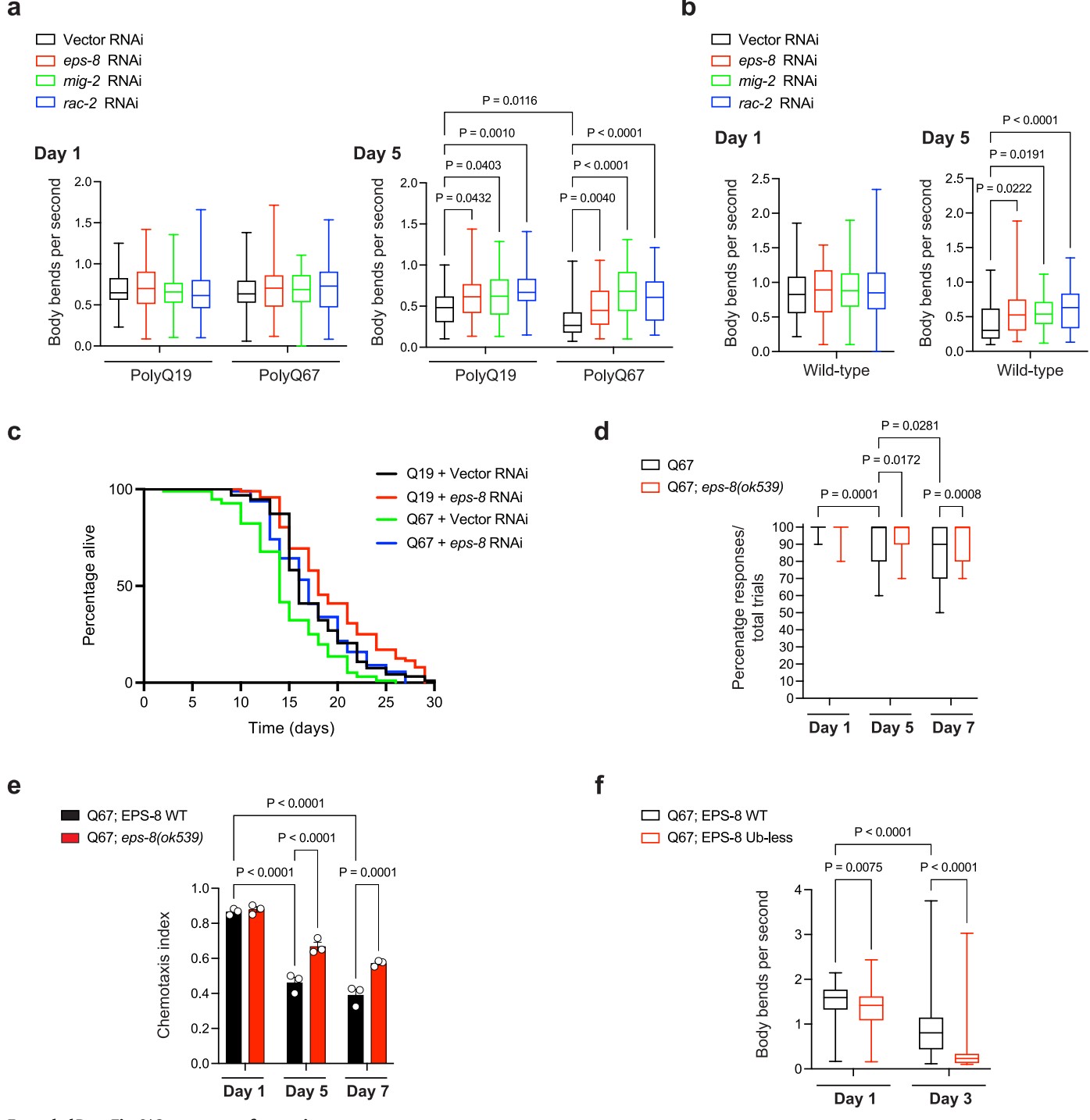

**Extended Data Fig. 2 | See next page for caption.**

**Extended Data Fig. 2 | EPS-8 levels modulate pathological changes in polyQ67 worms during aging. a**, Body bends per second in worms expressing neuronal polyQ peptides at day 1 and 5 of adulthood. Day 1(D1) Q19 + Vector RNAi: *n* = 76 worms; D1 Q19 + *eps-8* RNAi: *n* = 78; D1 Q19 + *mig-2* RNAi: *n* = 97; D1 Q19 + *rac-2* RNAi: *n* = 83; D1 Q67 + Vector RNAi: *n* = 78; D1 Q67 + *eps-8* RNAi: *n* = 72; D1 Q67 + *mig-2* RNAi: *n* = 86; D1 Q67 + *rac-2* RNAi: *n* = 81; D5 Q19 + Vector RNAi: *n* = 49; D5 Q19 + *eps-8* RNAi: *n* = 50; D5 Q19 + *mig-2* RNAi: *n* = 62; D5 Q19 + *rac-2* RNAi: *n* = 49; D5 Q67 + Vector RNAi: *n* = 54; D5 Q67 + *eps-8* RNAi: *n* = 55; D5 Q67 + *mig-2* RNAi: *n* = 49; D5 Q67 + *rac-2* RNAi: *n* = 49. **b**, Body bends per second in wild-type worms at day 1 and 5 of adulthood. D1 + Vector RNAi: *n* = 71 worms; D1 + *eps-8* RNAi: *n* = 88; D1 + *mig-2* RNAi: *n* = 76; D1 + *rac-2* RNAi: *n* = 80; D5 + Vector RNAi: *n* = 57; D5 + *eps-8* RNAi: *n* = 62; D5 + *mig-2* RNAi: *n* = 58; D5 + *rac-2* RNAi: *n* = 69. **c**, Worms expressing polyQ67 peptides have a shorter lifespan compared to control polyQ19 worms (*P* < 0.0001). Knockdown of *eps-8* after development extends lifespan in both polyQ67 (*P* = 0.0003) and polyQ19 worms (*P* = 0.0120). Q19 + vector RNAi mean ± s.e.m.: 17.44 days ± 0.44; Q19 + *eps-8* RNAi: 19.41 ± 0.53; Q67 + vector RNAi: 14.67 ± 0.44; Q67 + *eps-8* RNAi: 17.33 ± 0.46. *P*-values: two-sided log-rank test, n = 96 worms/condition. Supplementary Table 1 contains statistical analysis and replicate data from independent lifespan experiments. **d**, *eps-8* knockout mutation mitigates the age-related decline in nose-touch response induced by polyQ67 expression. Graph represents the percentage of nose-touch responses/total trials per worm at the indicated ages (*n* = 40 worms per condition). **e**, Chemotaxis index towards 0.5% benzaldehyde at the indicated ages (mean ± s.e.m., *n* = 3 independent experiments, 67-164 worms were scored per condition for each independent experiment). **f**, Body bends per second in wild-type (WT) EPS-8 and Ub-less EPS-8 worms expressing polyQ67 in neurons at day 1 and 3 of adulthood. D1 EPS-8(WT): *n* = 85 worms; D1 EPS-8(Ub-less): *n* = 79; D3 EPS-8(WT): *n* = 78; D3 EPS-8(Ub-less): *n* = 61. In **a,b,d,f**, the box plots represent the 25th–75th percentiles, the lines depict the median and the whiskers show the minimum–maximum values. Statistical comparisons were made by two-way ANOVA with Šidák multiple-comparison test (**a,d,e**), two-way ANOVA with Fisher's LSD test (**f**), one-way ANOVA with Dunnett's multiple-comparison test (**b**), and two-sided log-rank test (**c**).

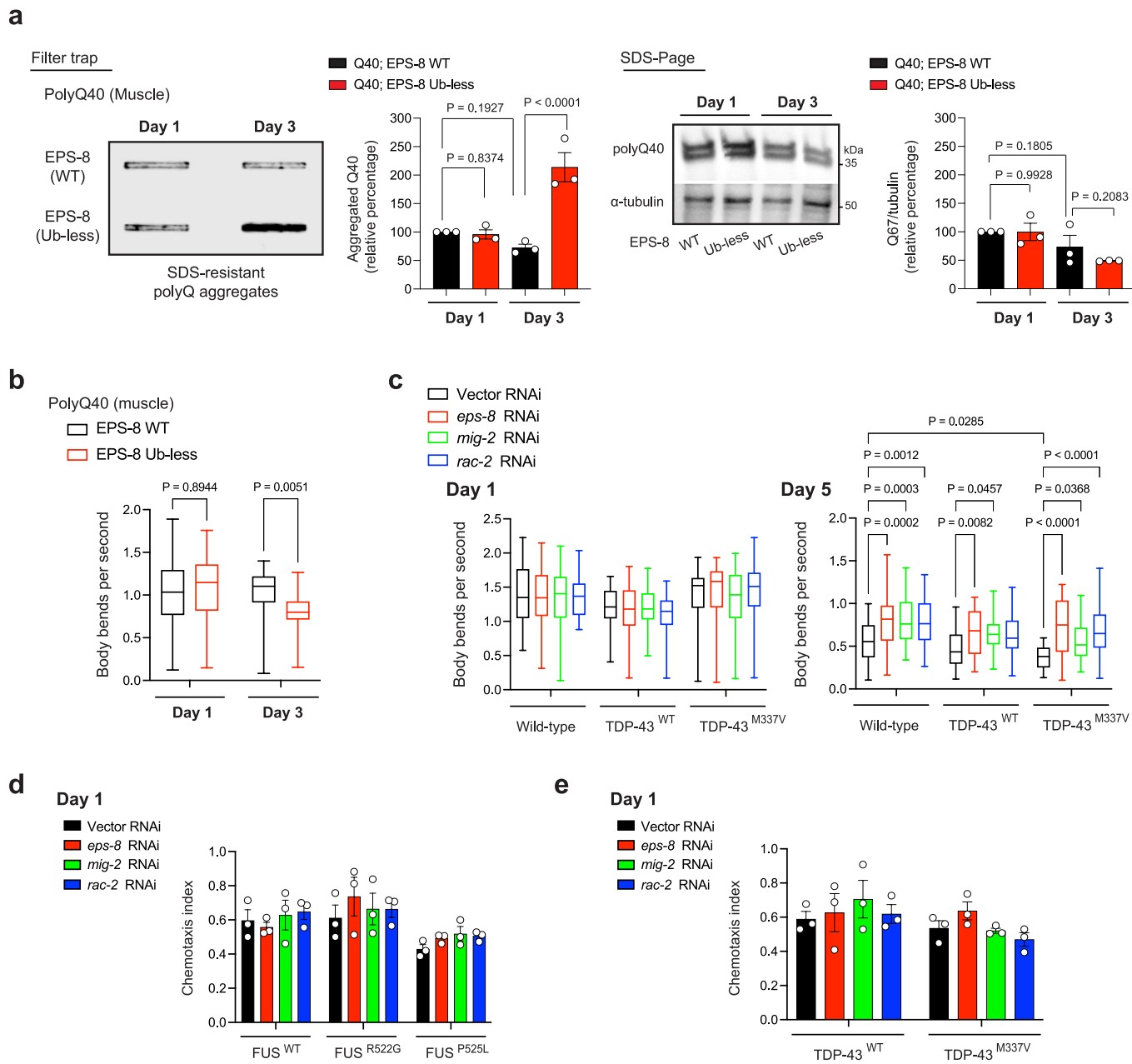

**Extended Data Fig. 3 | EPS-8/RAC signaling influences disease-related changes in *C. elegans* models. a**, Filter trap analysis of muscle polyQ40::YFP aggregates (detected by anti-GFP antibody) in worms expressing endogenous wild-type (WT) or Ub-less mutant EPS-8(K524R/K583R/K621R) at day 1 and 3 of adulthood. Right: SDS–PAGE with antibodies to GFP and α-tubulin. Graphs represent the relative percentage values of aggregated and total polyQ40 levels (corrected for α-tubulin) to day-1 adult Q40;EPS-8 (WT) worms (mean ± s.e.m., *n* = 3 independent experiments). **b**, Body bends per second in wild-type EPS-8 and Ub-less EPS-8 worms expressing polyQ40 in muscle cells. Day 1 (D1) EPS-8(WT): *n* = 46 worms; D1 EPS-8(Ub-less): *n* = 41; D3 EPS-8(WT): *n* = 51; D3 EPS-8(Ub-less): *n* = 45. **c**, Body bends per second in wild-type worms and transgenic worms expressing wild-type TDP-43 (WT) or ALS-related mutant TDP-43^M337V variant. D1 wild-type + Vector RNAi: *n* = 44 worms; D1 wild-type + *eps-8* RNAi: *n* = 47; D1 wild-type + *mig-2* RNAi: *n* = 41; D1 wild-type + *rac-2* RNAi: *n* = 42; D1 TDP-43(WT) + Vector RNAi: *n* = 45; D1 TDP-43(WT) + *eps-8* RNAi: *n* = 41; D1 TDP-43(WT) + *mig-2* RNAi: *n* = 47; D1 TDP-43(WT) + *rac-2* RNAi: *n* = 44; D1 TDP-43(M337V) + Vector RNAi: *n* = 45; D1 TDP-43(M337V) + *eps-8* RNAi: *n* = 44; D1 TDP-43(M337V) + *mig-2* RNAi: *n* = 43; D1 TDP-43(M337V) + *rac-2* RNAi: *n* = 41; D5 wild-type + Vector RNAi:

*n* = 42 worms; D5 wild-type + *eps-8* RNAi: *n* = 35; D5 wild-type + *mig-2* RNAi: *n* = 38; D5 wild-type + *rac-2* RNAi: *n* = 36; D5 TDP-43(WT) + Vector RNAi: *n* = 38; D5 TDP-43(WT) + *eps-8* RNAi: *n* = 39; D5 TDP-43(WT) + *mig-2* RNAi: *n* = 34; D5 TDP-43(WT) + *rac-2* RNAi: *n* = 38; D5 TDP-43(M337V) + Vector RNAi: *n* = 34; D5 TDP-43(M337V) + *eps-8* RNAi: *n* = 39; D5 TDP-43(M337V) + *mig-2* RNAi: *n* = 35; D5 TDP-43(M337V) + *rac-2* RNAi: *n* = 38. In **b,c**, the box plots represent the 25th–75th percentiles, the line depicts the median and the whiskers show the minimum–maximum values. **d**, Chemotaxis index of FUS ALS worm models toward 0.5% benzaldehyde at day 1 of adulthood (mean ± s.e.m., *n* = 3 independent experiments, 66-174 worms were scored per condition for each independent experiment). No significant differences were observed. **e**, Chemotaxis index of TDP-43 ALS worm models toward 0.5% benzaldehyde at day 1 of adulthood (mean ± s.e.m., *n* = 3 independent experiments, 48-257 worms were scored per condition for each independent experiment). No significant differences were observed. In all the experiments, RNAi was initiated after development. Statistical comparisons were by two-way ANOVA with Fisher's LSD test (**a,b**) and two-way ANOVA with Šidák multiple-comparison test (**c,d,e**).

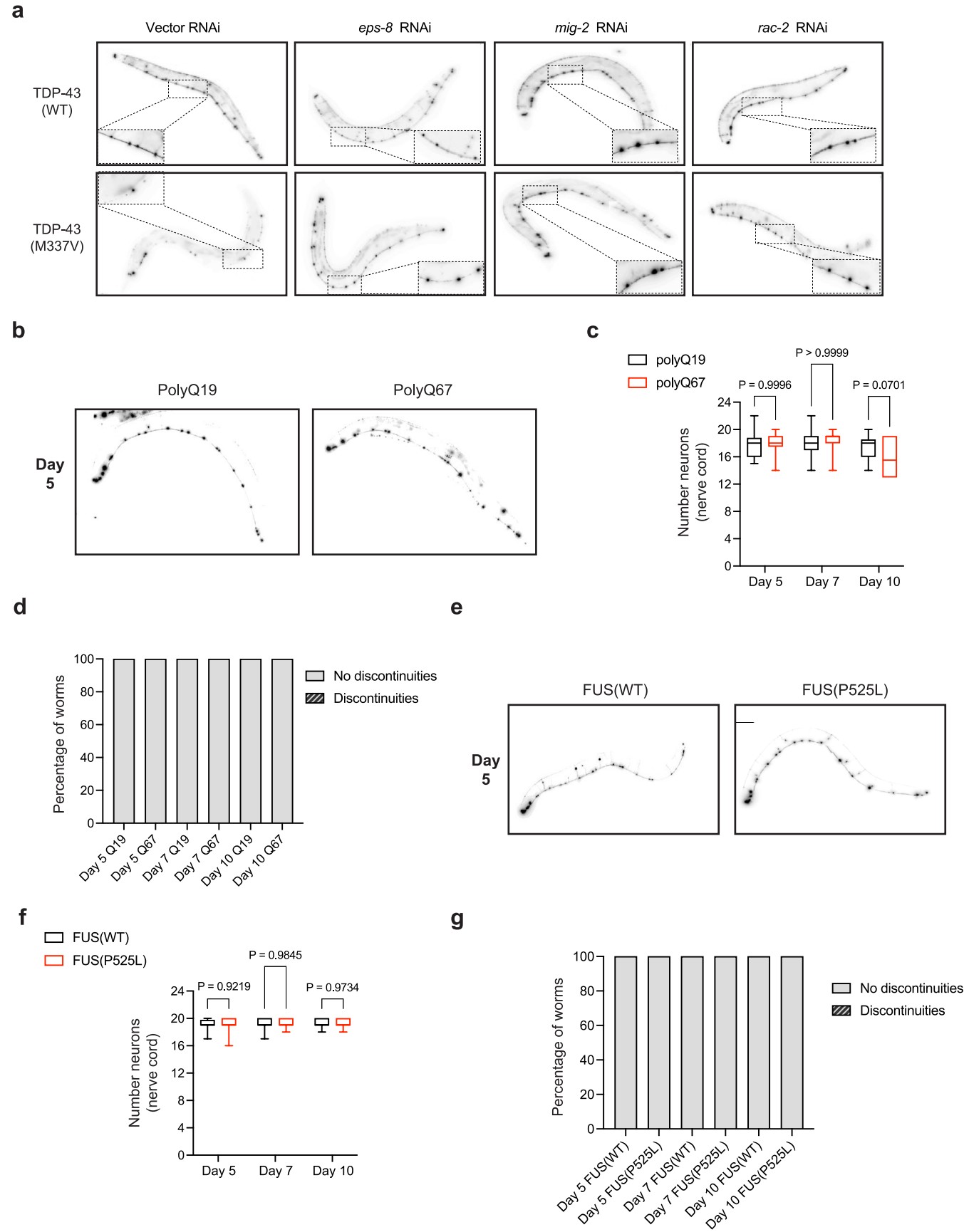

**Extended Data Fig. 4 | See next page for caption.**

**Extended Data Fig. 4 | Lowering EPS-8/RAC signaling ameliorates degeneration of GABAergic neurons in mutant TDP-43 worm models.**
**a**, Representative images of *unc-25* reporter expression in GABAergic neurons of day-5 adult TDP-43 worm models under the indicated RNAi treatments (representative of 2 independent experiments). **b**, Representative images of *unc-25* expression in GABAergic neurons of day-5 adult polyQ worm models. **c**, Number of GABAergic neurons in the nerve cord of polyQ worms at the indicated ages (mean ± s.e.m., Day 5 (D5) Q19: $n$ = 20 worms; D5 Q67: $n$ = 17; D7 Q19: $n$ = 18; D7 Q67: $n$ = 15; D10 Q19: $n$ = 17; D10 Q67: $n$ = 14). The box plots represent the 25th–75th percentiles, the line depicts the median and the whiskers show the minimum–maximum values. **d**, Graph represents the percentage of polyQ worms displaying discontinuities in the nerve cord (percentage from

14-20 worms). We did not observe neurodegeneration in polyQ67 worms. **e**, Representative images of *unc-25* expression in day-5 adult worms expressing wild-type FUS (WT) or mutant FUS[P525L]. **f**, Number of GABAergic neurons in the nerve cord of FUS-expressing worms at the indicated ages (mean ± s.e.m., D5 FUS(WT): $n$ = 20 worms; D5 FUS(P525L): $n$ = 19; D7 FUS(WT): $n$ = 16; D7 FUS(P525L): $n$ = 18; D10 FUS(WT): $n$ = 15; D10 FUS(P525L): $n$ = 18). The box plots represent the 25th–75th percentiles, the line depicts the median and the whiskers show the minimum–maximum values. **g**, Graph represents the percentage of FUS-expressing worms displaying discontinuities in the nerve cord (percentage from 15-20 worms). Statistical comparisons were made by two-way ANOVA with Šidák multiple-comparison test (**c,f**).

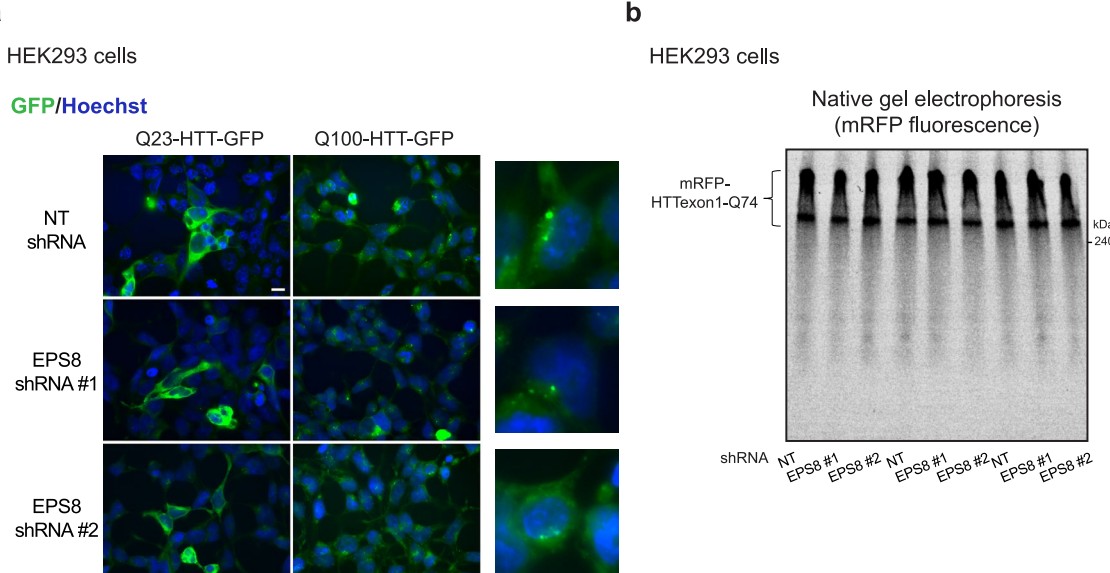

**a**

HEK293 cells

**b**

HEK293 cells

**Extended Data Fig. 5 | Knockdown of EPS8 does not change the intracellular distribution of polyQ aggregates or the levels of polyQ oligomers in human cells. a**, Images of HEK293 cells expressing Q23-HTT or Q100-HTT tagged with GFP. The cells were treated with non-targeting (NT) shRNA or two independent shRNA constructs against *EPS8*. Hoechst staining was used as a nuclear marker.

Scale bar: 10 μm. Representative of two independent experiments. **b**, mRFP fluorescence signal from native gel electrophoresis of lysates from HEK293 human cells expressing mRFP-HTTexon1-Q74. The image shows data from three biological replicates.

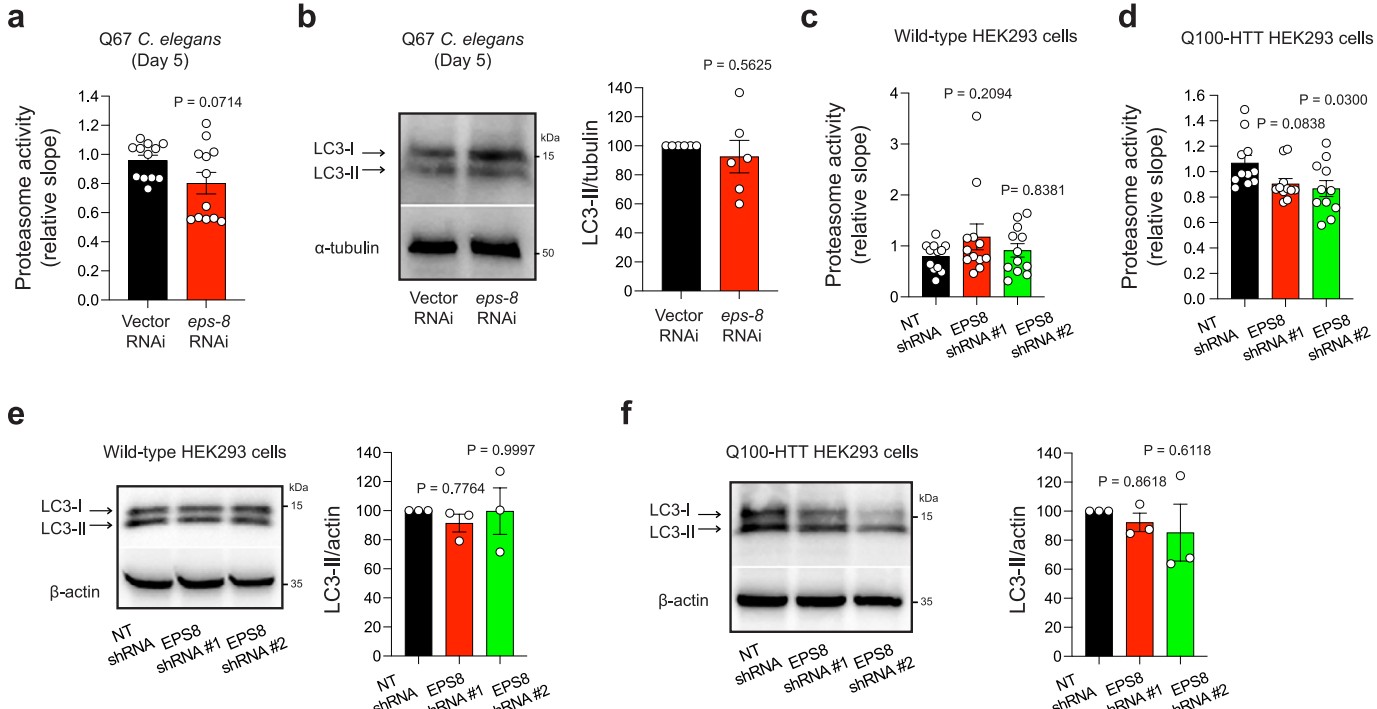

**Extended Data Fig. 6 | Knockdown of EPS8 does not increase proteasome activity or LC3 lipidation in *C. elegans* and human cells. a**, Chymotrypsin-like proteasome activity in polyQ67::YFP *C. elegans* (relative slope to Vector RNAi, mean ± s.e.m., *n* = 12 biological replicates). **b**, Western blot of polyQ67::YFP *C. elegans* with anti-LGG-1/LC3 antibody. LC3-I is conjugated to phosphatidylethanolamine to form LC3-II, whose levels reflect the number of autophagosomes and autophagy-related structures. α-tubulin is the loading control. Graph represents the relative percentage values of LC3-II (corrected for α-tubulin) to Vector RNAi (mean ± s.e.m., *n* = 6 independent experiments). In **a-b**, RNAi was initiated after development and worms were analyzed at day 5 of adulthood. **c**, Chymotrypsin-like proteasome activity in HEK293 cells (relative slope to non-targeting (NT) shRNA, mean ± s.e.m., *n* = 12 biological replicates).

**d**, Chymotrypsin-like proteasome activity in HEK293 cells expressing Q100-HTT-GFP (relative slope to NT shRNA, mean ± s.e.m., *n* = 11 biological replicates). **e**, Western blot of wild-type HEK293 cells with antibodies against LC3 and β-actin loading control. Graph represents the relative percentage values of LC3-II (corrected for β-actin) to NT shRNA (mean ± s.e.m., *n* = 3 independent experiments). **f**, Western blot of HEK293 cells expressing Q100-HTT-GFP with anti-LC3 antibody. Graph represents the relative percentage values of LC3-II (corrected for β-actin) to NT shRNA (mean ± s.e.m., *n* = 3 independent experiments). Statistical comparisons were made by two-tailed Student's *t*-test for unpaired samples (**a**), two-tailed Wilcoxon signed-rank test (**b**), and one-way ANOVA with Dunnett's multiple-comparison test (**c-f**).

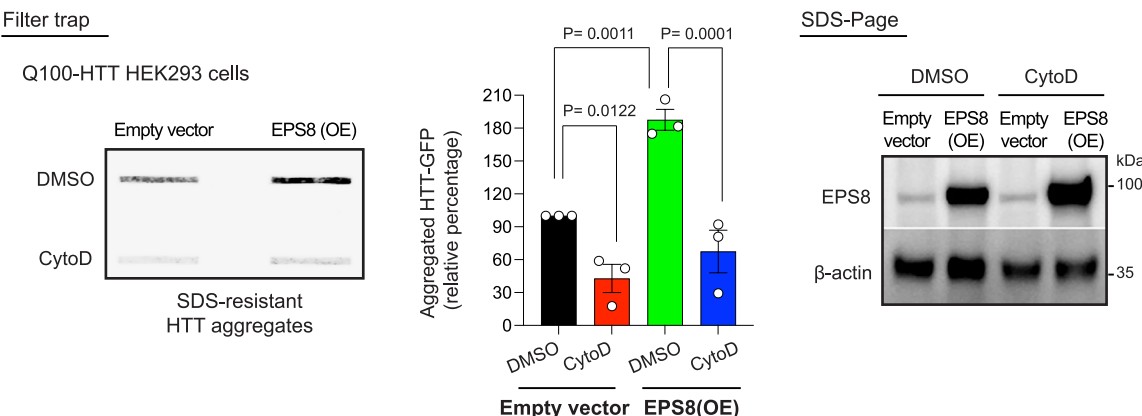

**Extended Data Fig. 7 | Cytochalasin D treatment reduces polyQ100-HTT aggregation induced by EPS8 overexpression in human cells.** Filter trap with anti-GFP antibody of Empty Vector or EPS8 overexpressing (OE) Q100-HTT-GFP HEK293. Cells were treated with 2 µM CytoD or DMSO vehicle control for 4 h before lysis. The graph represents the relative percentage of aggregated Q100-HTT levels to Empty Vector + DMSO (mean ± s.e.m., $n$ = 3 independent experiments). Statistical comparisons were made by two-way ANOVA with Fisher's LSD test. Right: SDS-PAGE with antibodies to EPS8 and β-actin loading control.

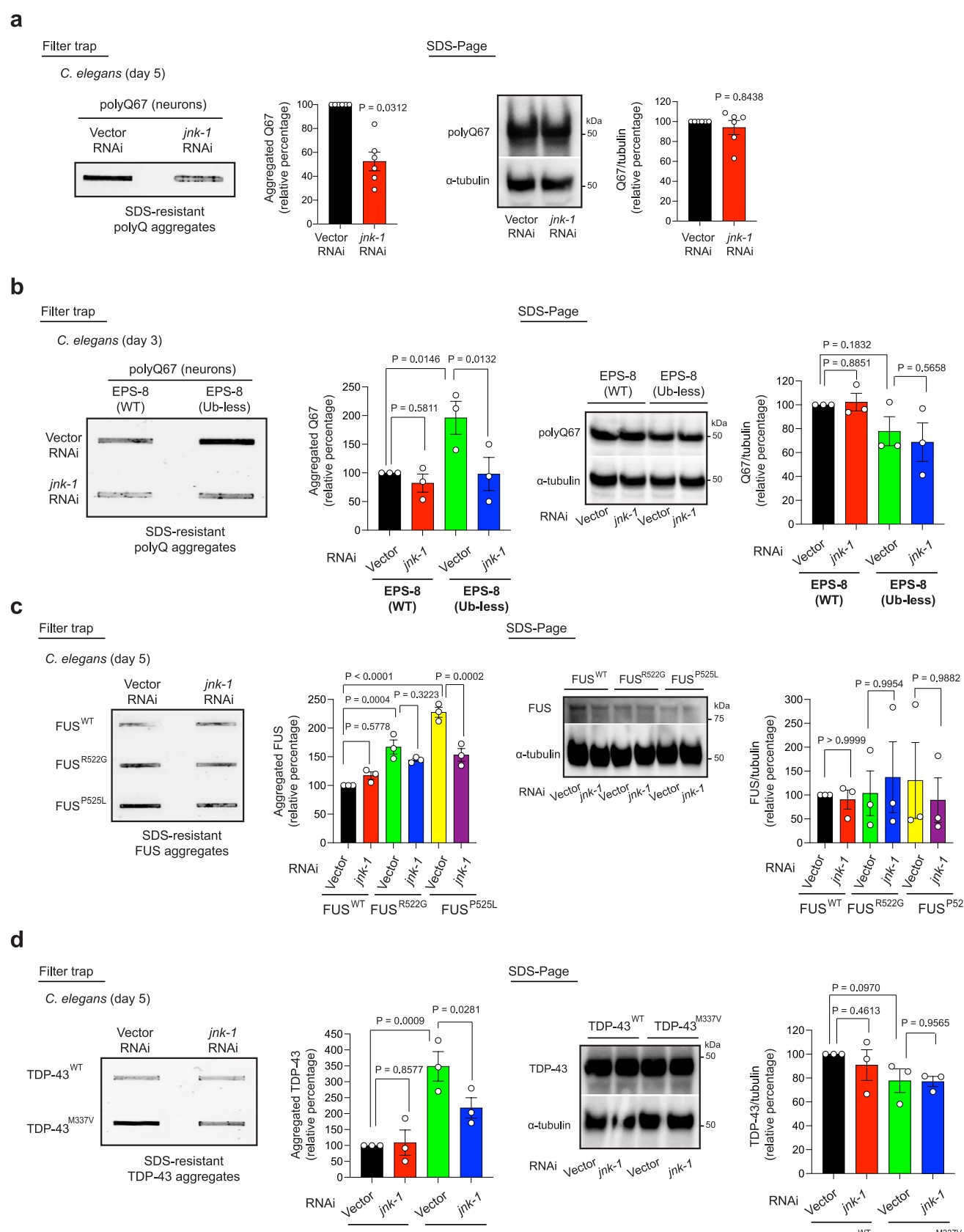

**Extended Data Fig. 8 | See next page for caption.**

**Extended Data Fig. 8 | Knockdown of *jnk-1* after development reduces disease-related protein aggregation in *C. elegans* models. a**, Knockdown of *jnk-1* after development prevents polyQ67::YFP aggregation (detected by anti-GFP antibody) in the neurons of day-5 adult worms. Right: SDS–PAGE with antibodies to GFP and α-tubulin. Graphs represent the relative percentage values of aggregated and total polyQ67 (corrected for α-tubulin loading control) to Vector RNAi (mean ± s.e.m., $n$ = 6 independent experiments). **b**, Filter trap analysis of polyQ67::YFP (detected by anti-GFP antibody) in day-3 adult worms expressing either endogenous wild-type (WT) EPS-8 or Ub-less mutant EPS-8 upon *jnk-1* RNAi. Right: SDS–PAGE with antibodies to GFP and α-tubulin. Graphs represent the relative percentage values of aggregated polyQ67 and total polyQ67 levels (corrected for α-tubulin loading control) to Q67;EPS-8(WT) + Vector RNAi (mean ± s.e.m., $n$ = 3 independent experiments). **c**, Knockdown of *jnk-1* reduces aggregation of the severe FUS$^{P525L}$ mutant variant, but does not significantly affect aggregation of wild-type (WT) FUS or mutant FUS$^{R522G}$ (detected by anti-FUS antibody). Right: SDS-PAGE with antibodies to FUS and α-tubulin. Graphs represent the relative percentage values of aggregated and total FUS levels (corrected for α-tubulin loading control) to FUS$^{WT}$ + Vector RNAi (mean ± s.e.m., $n$ = 3 independent experiments). **d**, Knockdown of *jnk-1* after development decreases TDP-43$^{M337V}$ aggregation in the neurons of day-5 adult worms (detected by anti-TDP-43 antibody). Right: SDS-PAGE with antibodies to TDP-43 and α-tubulin. Graphs represent the relative percentage values of aggregated TDP-43 and total TDP-43 levels (corrected for α-tubulin loading control) to wild-type (WT) TDP-43 + Vector RNAi (mean ± s.e.m., $n$ = 3 independent experiments). Statistical comparisons were made by two-tailed Wilcoxon signed-rank test (**a**), two-way ANOVA with Fisher's LSD test (**b, d**), and two-way ANOVA with Šidák multiple-comparison test (**c**).

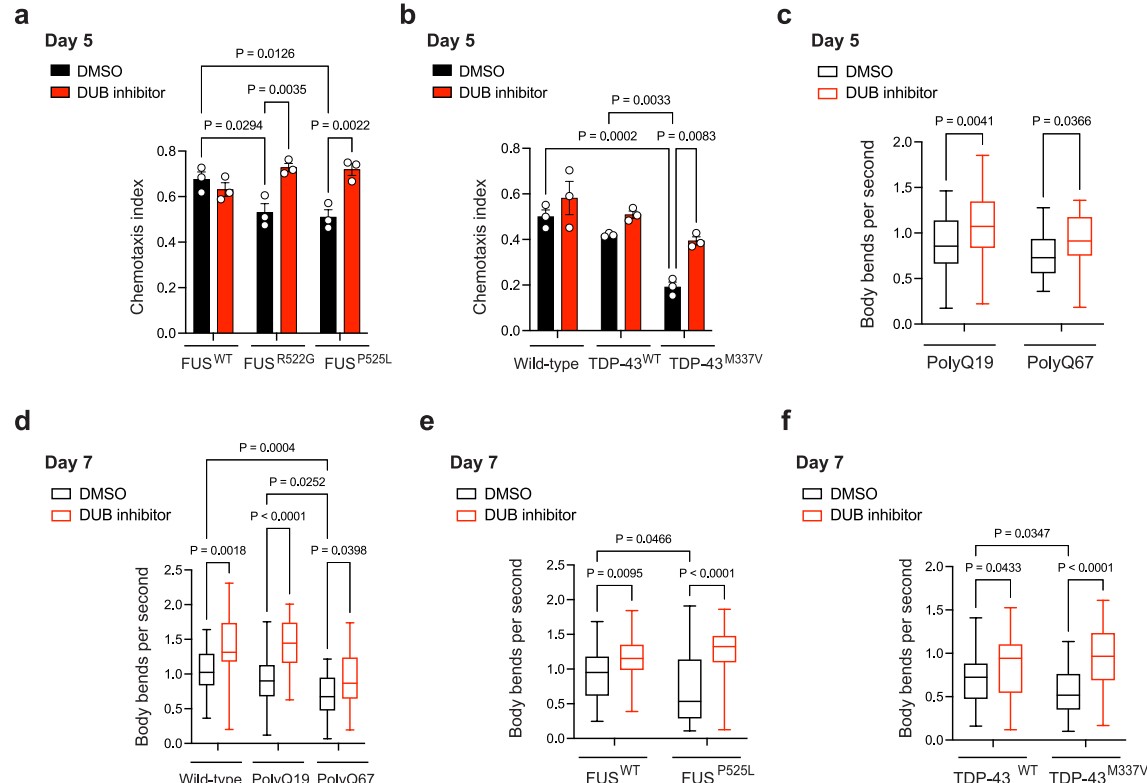

**Extended Data Fig. 9 | DUB inhibitor treatment rescues chemotaxis and motility deficits in *C. elegans* disease models. a**, Chemotaxis index of FUS ALS worm models towards 0.5% benzaldehyde (mean ± s.e.m., *n* = 3 independent experiments, 76-259 worms were scored per condition for each independent experiment). **b**, Chemotaxis index of wild-type worms and worms expressing either wild-type TDP-43 (WT) or ALS mutant TDP-43^M337V towards 0.5% benzaldehyde (mean ± s.e.m., *n* = 3 independent experiments, 42-104 worms were scored per condition for each independent experiment). **c**, Body bends per second in day-5 adult worms expressing neuronal polyQ peptides. Q19 + DMSO: *n* = 34 worms; Q19 + DUB inhibitor: *n* = 31; Q67 + DMSO: *n* = 31; Q67 + DUB inhibitor: *n* = 32. **d**, Body bends per second in day-7 adult wild-type and polyQ worms. Wild-type + DMSO: *n* = 35 worms; wild-type + DUB inhibitor: *n* = 31; Q19 + DMSO: *n* = 39; Q19 + DUB inhibitor: *n* = 40; Q67 + DMSO: *n* = 36;

Q67 + DUB inhibitor: *n* = 36. **e**, Body bends per second in day-7 adult FUS ALS worm models. FUS(WT) + DMSO: *n* = 37 worms; FUS(WT) + DUB inhibitor: *n* = 38; FUS(P525L) + DMSO: *n* = 35; FUS(P525L) + DUB inhibitor: *n* = 39. **f**, Body bends per second in day-7 adult TDP-43 ALS worm models (*n* = 37-42 worms per condition). TDP-43(WT) + DMSO: *n* = 37 worms; TDP-43(WT) + DUB inhibitor: *n* = 42; TDP-43(M337V) + DMSO: *n* = 41; TDP-43(M337V) + DUB inhibitor: *n* = 42. In **c-f**, the box plots represent the 25th–75th percentiles, the lines depict the median and the whiskers show the minimum–maximum values. In all the experiments, worms were treated with 13.7 μg/ml PR-619 (broad-spectrum DUB inhibitor) or vehicle control (DMSO) for 24 h at day 4 of adulthood and analyzed at the indicated ages. Statistical comparisons were made by two-way ANOVA with Šidák multiple-comparison test (**a,b,d**) and two-way ANOVA with Fisher's LSD test (**c,e,f**).

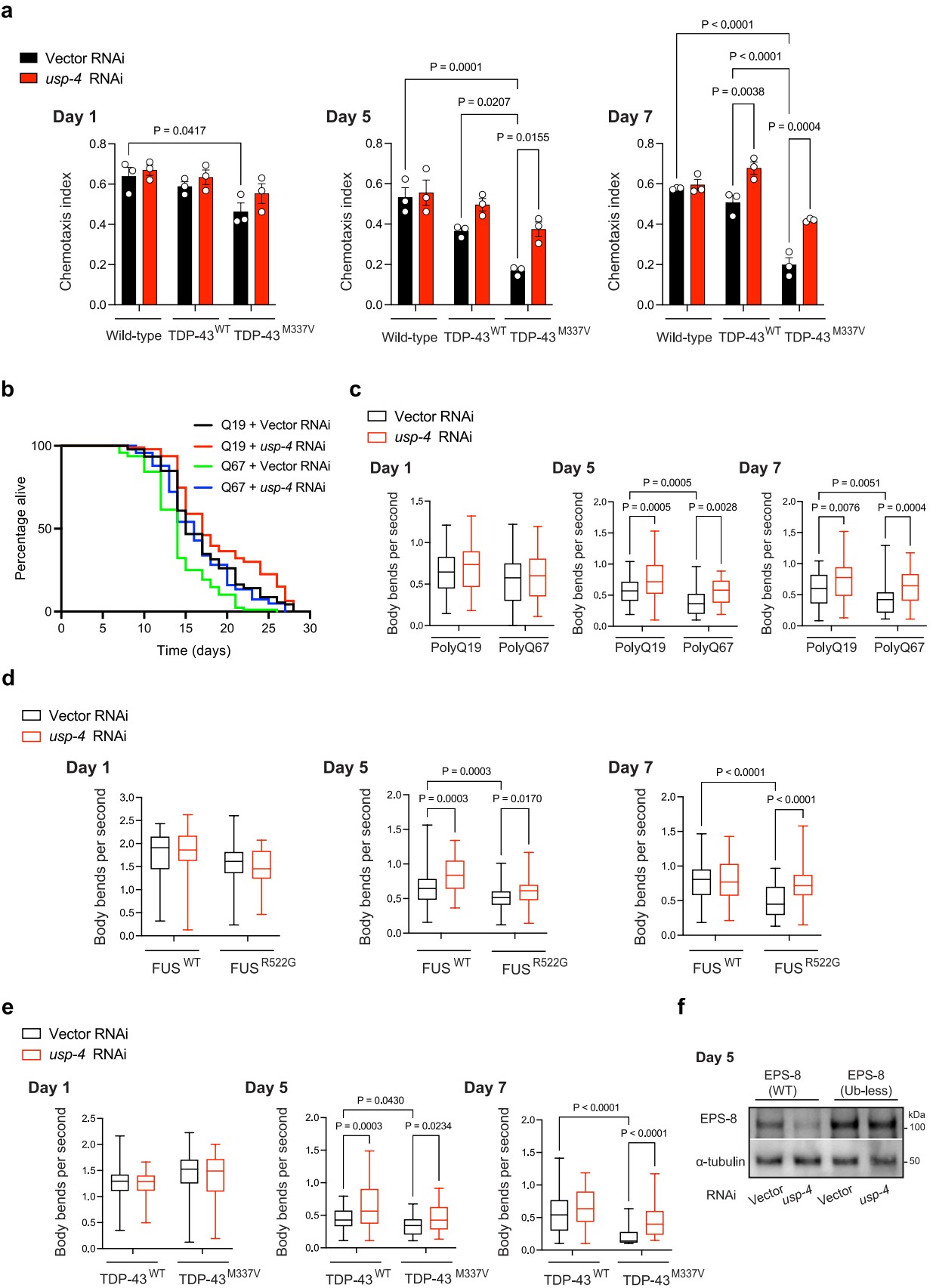

**Extended Data Fig. 10 | See next page for caption.**

**Extended Data Fig. 10 | Knockdown of *usp-4* prevents the detrimental effects caused by disease-related mutant proteins in *C. elegans*. a**, Chemotaxis index of wild-type worms and worms expressing either wild-type TDP-43 (WT) or ALS mutant TDP-43$^{M337V}$ towards 0.5% benzaldehyde at the indicated aged (mean ± s.e.m., *n* = 3 independent experiments, 51-117 worms were scored per condition for each independent experiment). **b**, Knockdown of *usp-4* after development extends lifespan in both polyQ67 (*P* = 0.0006) and polyQ19 worms (*P* = 0.0139). Q19 + vector RNAi mean ± s.e.m.: 16.97 days ± 0.50; Q19 + *usp-4* RNAi: 18.91 ± 0.55; Q67 + vector RNAi: 14.19 ± 0.38; Q67 + *usp-4* RNAi: 16.49 ± 0.47. *P*-values: two-sided log-rank test, n = 96 worms/condition. Supplementary Table 1 contains statistical analysis and replicate data from independent lifespan experiments. **c**, Body bends per second in worms expressing neuronal polyQ peptides (*n* = 41-56 worms per condition). **d**, Body bends per second in

worms expressing wild-type FUS (WT) or ALS-related mutant FUS$^{R522G}$ variant (*n* = 33-42 worms per condition). **e**, Body bends per second in worms expressing wild-type TDP-43 (WT) or ALS-related mutant TDP-43$^{M337V}$ variant (*n* = 33-42 worms per condition). In **c-e**, the box plots represent the 25th-75th percentiles, the lines depict the median and the whiskers show the minimum–maximum values. **f**, Western blot analysis of EPS-8 levels in day-5 adult worms expressing endogenous wild-type EPS-8 (WT) or Ub-less mutant EPS-8(K524R/K583R/K621R). Knockdown of *usp-4* does not decrease the levels of Ub-less EPS-8 mutant variant. Representative of two independent experiments. In all the experiments, RNAi was initiated after development. Statistical comparisons were made by two-way ANOVA with Šidák multiple-comparison test (**a**), two-way ANOVA with Fisher's LSD test (**c-e**), and two-sided log-rank test (**b**).

# Reporting Summary

## Statistics

For all statistical analyses, confirm that the following items are present in the figure legend, table legend, main text, or Methods section.

| n/a | Confirmed | |
|---|---|---|
| ☐ | ☒ | The exact sample size ($n$) for each experimental group/condition, given as a discrete number and unit of measurement |
| ☐ | ☒ | A statement on whether measurements were taken from distinct samples or whether the same sample was measured repeatedly |
| ☐ | ☒ | The statistical test(s) used AND whether they are one- or two-sided *Only common tests should be described solely by name; describe more complex techniques in the Methods section.* |
| ☒ | ☐ | A description of all covariates tested |
| ☐ | ☒ | A description of any assumptions or corrections, such as tests of normality and adjustment for multiple comparisons |
| ☐ | ☒ | A full description of the statistical parameters including central tendency (e.g. means) or other basic estimates (e.g. regression coefficient) AND variation (e.g. standard deviation) or associated estimates of uncertainty (e.g. confidence intervals) |
| ☐ | ☒ | For null hypothesis testing, the test statistic (e.g. $F$, $t$, $r$) with confidence intervals, effect sizes, degrees of freedom and $P$ value noted *Give P values as exact values whenever suitable.* |
| ☒ | ☐ | For Bayesian analysis, information on the choice of priors and Markov chain Monte Carlo settings |
| ☒ | ☐ | For hierarchical and complex designs, identification of the appropriate level for tests and full reporting of outcomes |
| ☒ | ☐ | Estimates of effect sizes (e.g. Cohen's $d$, Pearson's $r$), indicating how they were calculated |

*Our web collection on statistics for biologists contains articles on many of the points above.*

## Software and code

Policy information about availability of computer code

| Data collection | No software was used for data collection. |
|---|---|
| Data analysis | Densitometry of filter trap and western blot assasys was quantified using ImageJ software (version 1.51). For motility assays, body bends were quantified using ImageJ software (version 1.53k) with the wrMTrck plugin (www.phage.dk/plugins). We used GraphPad Prism (version 10.4.1) for statistical analysis of all the data. OASIS software (version 1) was used to determine mean lifespan (Yang, J.S., Nam, H.J., Seo, M., Han, S.K., Choi, Y., Nam, H.G., Lee, S.J. & Kim, S. OASIS: online application for the survival analysis of lifespan assays performed in aging research. PLoS One 6, e23525 (2011)). |

For manuscripts utilizing custom algorithms or software that are central to the research but not yet described in published literature, software must be made available to editors and reviewers. We strongly encourage code deposition in a community repository (e.g. GitHub). See the Nature Portfolio guidelines for submitting code & software for further information.

## Data

Policy information about availability of data

All manuscripts must include a data availability statement. This statement should provide the following information, where applicable:
- Accession codes, unique identifiers, or web links for publicly available datasets
- A description of any restrictions on data availability
- For clinical datasets or third party data, please ensure that the statement adheres to our policy

> The authors declare that all data supporting the findings of this study are available within the paper and its supplementary Information files.

## Research involving human participants, their data, or biological material

Policy information about studies with human participants or human data. See also policy information about sex, gender (identity/presentation), and sexual orientation and race, ethnicity and racism.

| | |
|---|---|
| Reporting on sex and gender | N/A |
| Reporting on race, ethnicity, or other socially relevant groupings | N/A |
| Population characteristics | N/A |
| Recruitment | N/A |
| Ethics oversight | N/A |

Note that full information on the approval of the study protocol must also be provided in the manuscript.

# Field-specific reporting

Please select the one below that is the best fit for your research. If you are not sure, read the appropriate sections before making your selection.

☒ Life sciences ☐ Behavioural & social sciences ☐ Ecological, evolutionary & environmental sciences

For a reference copy of the document with all sections, see nature.com/documents/nr-reporting-summary-flat.pdf

# Life sciences study design

All studies must disclose on these points even when the disclosure is negative.

| | |
|---|---|
| Sample size | Exact sample sizes are provided in the corresponding Figure legends and Extended Data Figure legends.<br>No statistical methods were used to predetermine sample size; however, our sample sizes were selected based on standards established in the field and are similar to, or greater than, those reported to be sufficient in previous publications using the same procedures (i.e. lifespan, nose-touch, chemotaxis, motility, percentage of GABAergic neurodegeneration in C. elegans, percentage of activated caspase-3 in human neurons, filter trap assays, western blotting, and proteasome activity measurements):<br>- Koyuncu S et al, Nature 596:285-290 (2021)<br>- Llamas et al, Aging Cell 20: e13446<br>- Lee HL et al; Nature Metabolism 1: 790-810 (2019)<br>- Lee HL et al; Nature Aging 3:546-566 (2023)<br>- Koyuncu S et al, Nature Communications 9: 2886 (2018)<br>- Amrit FR et al, Methods 68: 465–475 (2014)<br>- Fatima A et al, Communications Biology 3: 262<br>- Alirzayeva H et al, Cell Reports 43: 114626 (2024)<br>- Segref A et al, Nature Communications 13: 5874 (2022))<br>- Koopman M et al, MicroPublication Biology: 10.17912/micropub.biology.000769 (2023)<br>- Fernandez-Abascal J, Neuron 110:470–485 (2022)<br>- Vilchez D et al, Nature 489:263-8 (2012)<br>- Vilchez D et al, Nature 489:304-8 (2012)<br>- Hart AC et al, Journal of Neuroscience 19, 1952-1958 (1999)<br>- Hahm JH et al, Nature Communications 6: 8919<br>- Liachko NF, Journal of Neuroscience 30, 16208-16219 (2010) |
| Data exclusions | For motility assays, worms were excluded from analysis if they showed less than 0.1 body-bends per second or were not recognized by the program. No data were excluded from other analyses. |

| Replication | At least three independent experiments were conducted for each assay to verify the reproducibility of the findings. If only two independent experiments were performed, this is indicated in the figure legend. All replication attempts yielded similar results. Lifespan assays were conducted at least twice, with 96 animals per condition. Exact sample sizes and the number of independent experiments are specified in the corresponding Figure and Extended Data Figure legends. |
|---|---|
| Randomization | For C. elegans experiments, worm populations were synchronized either by allowing young hermaphrodites to lay eggs for 6 hours or by using the bleaching method (PMID: 22710399). For synchronization via egg laying, young hermaphrodites were randomly picked from maintenance plates. For synchronization via bleaching, random chunks of agar containing mixed-stage animals were transferred from maintenance plates, and larvae were allowed to grow until a sufficient number of young hermaphrodites were available for bleaching. After the 6-hour egg-laying period or bleaching, the resulting larvae were raised to adulthood. For lifespan assays, adult worms were randomly picked and transferred from the synchronized population to the different experimental conditions. For all other experiments, adult worms were randomly distributed into the various experimental groups from single pulls of synchronized populations. The different experimental conditions were collected and lysed in a random order, although data collection and analysis were not randomized.<br><br>In the human cell line experiments, cells with similar confluence were split, and equal numbers of cells were transferred to new plates for the experiments. The plates were randomly assigned to different treatment conditions. Samples were collected and lysed in a random order, although data collection and analysis were not randomized. |
| Blinding | The samples and experimental conditions were not processed in a blinded manner by the researchers involved in this study. However, cells and worms were randomly allocated from from single pulls to different treatment conditions, and key experiments were independently repeated by different researchers to ensure reproducibility.<br>Filter trap, western blot, proteasome activity, and motility assays were not performed under blinded conditions, as these rely on objective instrument-based measurements and/or provide indirect quantitative outputs. Data analysis for these assays was likewise unblinded, as the investigators who performed the analysis also loaded the samples during the experiment, and the corresponding outputs from the measurement equipment were released in sequence.<br>For experiments with direct phenotypic outputs—such as lifespan assays, nose-touch assays, chemotaxis assays, quantification of GABAergic neurodegeneration in C. elegans, and activated caspase-3 levels in human neurons—blinding was not applied during data collection or analysis. This was due to the pronounced and well-characterized phenotypes associated with the disease models and treatment conditions used in this study. The researchers conducting these experiments had extensive prior experience with the specific strains and treatments, which often exhibit obvious phenotypes (e.g., severe defects in polyQ and ALS models, or smaller size in RAC RNAi-treated worms). Nonetheless, worms were randomly assigned to treatment groups, and experimental conditions were assessed in a randomized order. These assays and their analyses were also independently repeated by different researchers to confirm robustness and reproducibility. |

# Reporting for specific materials, systems and methods

We require information from authors about some types of materials, experimental systems and methods used in many studies. Here, indicate whether each material, system or method listed is relevant to your study. If you are not sure if a list item applies to your research, read the appropriate section before selecting a response.

## Materials & experimental systems

| n/a | Involved in the study |
|---|---|
| ☐ | ☒ Antibodies |
| ☐ | ☒ Eukaryotic cell lines |
| ☒ | ☐ Palaeontology and archaeology |
| ☐ | ☒ Animals and other organisms |
| ☒ | ☐ Clinical data |
| ☒ | ☐ Dual use research of concern |
| ☒ | ☐ Plants |

## Methods

| n/a | Involved in the study |
|---|---|
| ☒ | ☐ ChIP-seq |
| ☒ | ☐ Flow cytometry |
| ☒ | ☐ MRI-based neuroimaging |

## Antibodies

| Antibodies used | We used the following antibodies in this study:<br><br>*For western blot:<br>anti-α-tubulin (Sigma, T6199, 1:5,000. RRID: AB_477583). Monoclonal, clone number: DM1A<br>anti-β-actin (Abcam, ab8226, 1:1,000. RRID: AB_306371). Monoclonal, clone number: mAbcam 8226<br>anti-EPS8 (Proteintech, 12455-1-AP, 1:1000). Polyclonal<br>anti-FUS (Abcam, ab154141, 1:1000. RRID: AB_2885092). Monoclonal, clone number: CL0190.<br>anti-TDP43 (Abcam, ab225710, 1:1000). Polyclonal.<br>anti-USP4 (Abcam, ab181105, 1:1000). Monoclonal, clone number: EPR13846<br>anti-HTT (Cell Signaling, #5656, 1:1000). Monoclonal, clone number: D7F7<br>anti-EPS8L2 (Abcam, ab85960, 1:1,000, RRID: AB_1924963). Polyclonal.<br>anti-LGG-1 (PMID: 30910027, 1:2,000).<br>anti-LC3B (Cell Signaling, #2775, 1:1,000). Polyclonal.<br>anti-Phospho-RIP (Ser166) (Cell Signaling, #65746, 1:1,000). Monoclonal, clone number: D1L3S<br>anti-RIP (Cell Signaling, #3493, 1:1,000). Monoclonal, clone number: D94C12 |
|---|---|

*Filter trap of aggregation-prone proteins:
anti-GFP (AMSBIO, 210-PS-1GFP, 1:5,000. RRID: AB_10013682). Polyclonal.
anti-FUS (Abcam, ab154141, 1:1000. RRID: AB_2885092). Monoclonal, clone number: CL0190.
anti-TDP43 (Abcam, ab225710, 1:1000). Polyclonal.

*Immunocytochemistry:
anti-Cleaved Caspase 3 (Cell Signaling, #9661S, 1:300. RRID: AB_2341188). Polyclonal.
anti-MAP2 (2a+2b) (Sigma-Aldrich, #M1406, 1:300. RRID: AB_477171). Monoclonal, clone number: AP-20
Alexa Fluor 488 Goat anti-Mouse IgG (H+L) (ThermoFisher, A-11029, 1:500. RRID: AB_2534088). Polyclonal.
Alexa Fluor 568F(ab')2 Fragment of Goat Anti-Rabbit IgG (H+L) (ThermoFisher, A-21069, 1:500. RRID: AB_141416). Polyclonal.

* Protein immunoprecipitation:
anti-USP-4 antibody (Abcam, ab181105, 1:100)
anti-Normal Rabbit IgG (Cell Signaling, 2729, 1:378)

| Validation | Validation of antibodies were done by the stated manufacturer's, this study, or previous publications and supported by the publications indicated in the manufacturer's website, the Resource Identification Portal (RRID) and other publications using C. elegans and human cells. |

* anti-α-tubulin (Sigma, T6199, 1:5,000. RRID: AB_477583). The antibody was validated as a loading control for western blot analysis in C. elegans in our previous publications: PMID: 32451438; PMID: 27892468; PMID: 34172445; PMID: 34321666; PMID: 37118550
*anti-β-actin (Abcam, ab8226, 1:1,000. RRID: AB_306371) was used according to the manufacturer's instructions and our previous publications: PMID: 27892468; PMID: 30038412; PMID: 32451438; PMID: 37118550
* anti-EPS-8 (Proteintech, 12455-1-AP, 1:1000). The antibody was used according to the manufacturer's instructions, validated in previous studies (PMID: 34391775, PMID: 32147678), and confirmed by the data presented in this study (e.g. western blot of knockdown and overexpression experiments in human cells).
*anti-EPS8L2 (Abcam, ab85960, 1:1,000, RRID: AB_1924963). The antibody was used according to the manufacturer's instructions and validated in our previous publication (PMID: 34321666).
* anti-USP-4 (Abcam, ab181105, 1:1000). The antibody was used according to the manufacturer's instructions, validated in previous studies (PMID: 33038351, PMID: 29542252), and confirmed by the data presented in this study (e.g. western blot of knockdown experiments in human cells).
* anti-LGG-1 (PMID: 30910027, 1:2,000). This antibody was generated and validated in Springhorn, A. & Hoppe, T. Western blot analysis of the autophagosomal membrane protein LGG-1/LC3 in Caenorhabditis elegans. Methods Enzymol 619, 319-336 (2019).
* anti-LC3B (Cell Signaling, #2775, 1:1,000). The antibody was used according to the manufacturer's instructions, validated in previous studies (PMID:  39079530, PMID:39635846).
* anti-Phospho-RIP (Ser166) (Cell Signaling, #65746,  1:1,000). The antibody was used according to the manufacturer's instructions, validated in previous studies (PMID:  39505876, PMID: 39526730).
* anti-RIP (Cell Signaling, #3493, 1:1,000). The antibody was used according to the manufacturer's instructions, validated in previous studies (PMID:  39681571, PMID: 39753884).
*anti-HTT (Cell Signaling, ab#5656, 1:1000). The antibody was used according to the manufacturer's instructions and validated in our previous publications (PMID: 30038412, PMID: 36611004).
* anti-GFP (AMSBIO, 210-PS-1GFP, 1:5,000. RRID: AB_10013682). This antibody has been validated for filter trap and western blot in C. elegans and human cells in our previous publications: PMID: 27892468; PMID: 30038412; PMID: 34172445; PMID: 34321666; PMID: 37118550.
* anti-FUS (Abcam, ab154141, 1:1000. RRID:AB_2885092) was used according to the manufacturer's instructions and our previous publications for filter trap and western blot experiments: PMID: 30038412; PMID: 34172445; PMID: 37118550.
* anti-TDP43 (Abcam, ab225710, 1:1000) was used according to the manufacturer's instructions and our previous publications for filter trap and western blot experiments where it was previously validated:  PMID: 34172445; PMID: 37118550.
*  anti-Cleaved Caspase 3 (Cell Signaling, #9661S, 1:300. RRID:AB_2341188) was used according to the manufacturer's instructions and validated in multiple studies and our previous publication (e.g. PMID:16736467, PMID:17099894, PMID:17299760, PMID:17990272, PMID:19830812, PMID:20235094, PMID:20593360, PMID:20653033, PMID:20653035, PMID: 37118550 etc.).
* anti-MAP2 (2a+2b) (Sigma-Aldrich, #M1406, 1:300. RRID:AB_477171) was used according to the manufacturer's instructions and validated in multiple studies and our previous publication (e.g. PMID:19058188, PMID:19950118, PMID:26509469, PMID: 37118550 etc)
*Alexa Fluor 488 Goat anti-Mouse IgG (H+L) (ThermoFisher, A-11029, 1:500. RRID:AB_2534088) was used according to the manufacturer's instructions and validated in multiple studies (e.g. PMID:34995520, PMID:35194846, PMID:35219381).
*Alexa Fluor 568F(ab')2 Fragment of Goat Anti-Rabbit IgG (H+L) (ThermoFisher, A-21069, 1:500. RRID:AB_141416) was used according to the manufacturer's instructions and validated in multiple studies (e.g. PMID:35111373, PMID:31526765, PMID:29103933).

# Eukaryotic cell lines

Policy information about cell lines and Sex and Gender in Research

| Cell line source(s) | In this study, we used the human HEK293 cell line (HEK293T/17) obtained from the American Type Culture Collection (ATCC). Catalog number: CRL-11268.
ALS-iPSCs (FUSP525L/P525L) were kindly provided by I. Bozzoni and A. Rosa (Sapienza University of Rome). This iPSC line was established and characterized for pluripotency in ref.: Lenzi J et al. ALS mutant FUS proteins are recruited into stress granules in induced pluripotent stem cell-derived motoneurons. Dis Model Mech 8: 755-766  (2015), (PMID: 26035390). ALS-iPSCs were raised from control iPSCs by TALEN (transcription activator-like effector nucleases)-directed mutagenesis and are homozygote for a FUS mutation (P525L) linked with severe ALS (Lenzi J et al. Dis Model Mech 8: 755-766  (2015)). |

| Authentication | The HEK293T/17 cell line commercially obtained from ATTC has not been authenticated in our laboratory. We have authenticated the iPSC lines in the laboratory by performing STR analysis (PMID: 30038412). We confirmed that the STR profile of the ALS-iPSCs used in this study matches with the profile of their isogenic control iPSCs. |
| --- | --- |
| Mycoplasma contamination | All the cell lines used in this study were tested for mycoplasma contamination at least once every 3 weeks. No mycoplasma contamination was detected. |
| Commonly misidentified lines (See ICLAC register) | None of the cell lines used in this paper are listed in the database of commonly misidentified cell lines maintained by ICLAC (version 12, released 16th January 2023) |

# Animals and other research organisms

Policy information about studies involving animals; ARRIVE guidelines recommended for reporting animal research, and Sex and Gender in Research

| Laboratory animals | In this study, we used different Caenorhabditis elegans strains. For all the experiments, we used hermaphrodites worms.

Lifespan analysis was started from day 1 of adulthood. For all the other experiments on C. elegans, the specific age is indicated in the corresponding figures and/or figure legends.

The C. elegans strains used in this study were:

Wild-type (N2)
AM141 (rmIs133[unc-54p::Q40::yellow fluorescent protein (YFP)])
AM23 (rmIs298[F25B3.3p::Q19::CFP])
AM716 (rmIs284[F25B3.3p::Q67::YFP])
MAH602 (sqIs61[vha-6p::Q44::YFP + rol-6(su1006)])
CK405 (Psnb-1::TDP-43WT, myo-2p::dsRED)
CK423 (Psnb-1::TDP-43M337V, myo-2p::dsRED)
ZM5838 (hpIs223[rgef-1p::FUSWT::GFP])
ZM5842 (hpIs228[rgef-1p::FUSR522G::GFP])
ZM5844 (hpIs233[rgef-1p::FUSP525L::GFP])
DVG196 (rmIs284[F25B3.3p::Q67::YFP];sid-1(pk3321)V;uIs69[pCFJ90(myo-2p::mCherry) + unc-119p::sid-1])
VDL05 (eps-8(syb2901)IV))
VDL06 (eps-8(syb2901, syb3149)IV)
DVG365 (rmIs284[pF25B3.3::Q67::YFP]; eps-8(ok539))
DVG344 (rmIs284[pF25B3.3::Q67::YFP]); eps-8(syb2901)IV)
DVG345 (rmIs284[pF25B3.3::Q67::YFP]); eps-8(syb2901 syb3149))
DVG363 (rmIs133[unc-54p::Q40::YFP]); eps-8(syb2901))
DVG364 (rmIs133[unc-54p::Q40::YFP]); eps-8(syb2901, syb3149))
NFB2862 (Psnb-1::TDP-43WT, myo-2p::dsRED; juIs76[unc-25p::GFP + lin-15(+)]II)
NFB2863 (Psnb-1::TDP-43M337V, myo-2p::dsRED; juIs76[unc-25p::GFP + lin-15(+)]II)
NFB2858 (rmIs298[F25B3.3p::Q19::CFP]; otIs549[unc-25p::unc-25(partial)::mChopti::unc-54 3'UTR + pha-1(+)]; him-5(e1490)V)
NFB2859 rmIs284[F25B3.3p::Q67::YFP]; otIs549[unc-25p::unc-25(partial)::mChopti::unc-54 3'UTR + pha-1(+)]; him-5(e1490)V)
NFB2860 (hpIs223[rgef-1p::FUSWT::GFP]); otIs549[unc-25p::unc-25(partial)::mChopti::unc-54 3'UTR + pha-1(+)]; him-5(e1490)V)
NFB2861 (hpIs233[rgef-1p::FUSP525L::GFP]); otIs549[unc-25p::unc-25(partial)::mChopti::unc-54 3'UTR + pha-1(+)]; him-5(e1490)V) |
| --- | --- |
| Wild animals | The study did not involve wild animals. |
| Reporting on sex | In this study, we used hermaphrodites worms for all Caenorhabditis elegans experiments. |
| Field-collected samples | No field collected samples were used in the study. |
| Ethics oversight | We used the invertebrate C. elegans as a model organism and no ethical approval was required. According to the "Zentrale Kommission für die Biologische Sicherheit" (ZKBS), the responsible entity inside the Bundesamt für Verbraucherschutz und Lebensmittelsicherheit to assess the risk of Genetically Modified Organisms (GMO), genetic work with C. elegans is classified as risk group 1 (biological safety level 1: S1). Accordingly, we carried out our work on C. elegans in a S1-laboratory. The use of GMO in Germany is regulated by the "Gentechnik-Gesetz", and we followed the guidelines applying to S1 work with GMO (i.e., documentation of the project and of the, exact description of the creation and maintenance of the genetic modification or correct waste treatment). |

Note that full information on the approval of the study protocol must also be provided in the manuscript.

## Plants

Seed stocks

N/A

Novel plant genotypes

N/A

Authentication

N/A

