## [Peer Review File · Nature Aging]

The aging factor EPS8 induces disease-related protein aggregation through RAC signaling hyperactivation

Corresponding Author: Professor David Vilchez

Version 1:

Reviewer comments:

Reviewer #1

(Remarks to the Author)

General summary of the work

In this manuscript, Koyuncu et al. build on previous observations (from this group) that, with age, reduced ubiquitination of EPS-8 leads to its accumulation, thereby hyperactivating RAC signalling and shortening lifespan. Here the authors extend these findings by showing that knockdown of EPS-8/RAC components (*eps-8*, *mig-2*, *rac-2*) strongly suppresses the age-related aggregation of disease associated proteins (polyglutamine, FUS and TDP-43) in neurons, muscles and intestine. Conversely, ubiquitylation-resistant (Ub-less) mutants of EPS-8 accelerate the aggregation of these proteins, with similar effects observed in HEK293 cells upon depletion or overexpression of EPS8. The authors show that treatment with cytochalasin D (an inhibitor of actin-polymerization) or knockdown of *kgb-1*/JNK blocks the increased protein aggregation that occurs in *C. elegans* and HEK293 cells upon exposure to ub-less EPS-8 or EPS8 overexpression, respectively. Finally, the authors show that these effects are dependent on DUB activity, and pinpoint USP-4 as the key DUB acting on EPS-8/EPS8 to control its degradation and subsequent impact on RAC signalling and protein aggregation.

I like the work and find it to be an interesting and informative extension of the groups previous findings. The effects on protein aggregation are very convincing and the conservation of the core mechanism from worms to HEK293 cells suggests a general importance of the underlying biology. The manuscript is clear and compelling, and I agree with the conclusions presented. The work is appropriate for Nature Aging, but there are some areas of the manuscript that could be improved and/or expanded. In addition, I have some issues with the statistics used in some of the figure panels. While I do not expect this to alter the core conclusions presented, I would still like the authors to address these.

Major comments

1. In Figure 1, the effects of *eps-8*, *mig-2* and *rac-2* RNAi on aggregation in neurons, muscle and intestine are very clear and convincing. Similarly, the effects on thrashing in neuronal Q67 and muscle Q40 animals look good. However, knockdown of *eps-8*, *mig-2* and *rac-2* also increases thrashing in Q19 worms (Fig. 1b). This makes me wonder if knocking down EPS-8/RAC components makes worms generally more hyperactive, rather than specifically suppressing Q67 or Q40 toxicity in neurons/muscles. Does *eps-8*, *mig-2*, *rac-2* RNAi also enhance thrashing in Q0::YFP worms or wildtype worms? If so, the authors should use another measure of neuronal toxicity/function in Q67 worms, such as chemotaxis.

2. Fig 1g and h, the effects of ub-less EPS-8 on polyQ aggregation and toxicity in neurons are very nice. Does this mutant also accelerate polyQ aggregation and toxicity in muscle tissue?

3. In Fig. 2b, the effects of EPS-8/RAC RNAi on thrashing in FUS and TDP-43 mutants look convincing. However, as with the polyQ experiments, the RNAi treatments also enhance thrashing in worms expressing WT FUS. Are there other measures of neuronal health that can be used here to show that the RNAi is really ameliorating toxicity in neurons and not just making worms hyperactive?

4. In Fig. 5c and d, The DUB inhibitor appears to make the Q67 and FUS P525L groups even better than their corresponding Q19 or WT FUS groups. Are these differences statistically significant? If so, can the authors please comment on this in the text (i.e. briefly speculate why this might be the case).

5. The statistical tests used throughout the Figures are incorrect. Two-tailed, paired t-tests are often used even though samples are not paired. In addition, there are many instances where two-tailed t-tests are used to compare > 2 means or

when 2 independent variables (e.g. genotype AND RNAi treatment) are being compared. Moreover, there are several examples where the groups being compared do not have anormal distribution, as values for the control groups have all been set at 1. The authors should apply the correct statistical tests to the hypotheses being tested in all cases. This should be as follows:

One-way ANOVA with Dunnett's comparison to vector RNAi control: Figure 1a,c,d,e, f, g; Figure 2a and c; Figure 3a, b and c; Figure 7c, e and g.

Two-way ANOVA with Fisher's LSD test or equivalent: Figure 5b-e; Figure 6c and f; Figure S2.

Two-way ANOVA with pairwise comparisons of means: Figure 1b; Figure 2b and d;

Non-parametric tests should be used (e.g. Wilcoxon) – Figure 3g; Figure 5a; Figure 6b,d,e and g; Figure 7f.

I should stress that given the differences observed, the number of comparisons being made, and the distributions of the data presented, I do not expect this to alter the authors conclusions significantly. However, it will still be good practice to employ the correct statistical tests.

6. In the Discussion section (page 10), the authors state that: “Our data suggest that excessive actin polymerization triggered by elevated EPS8/RAC activity contributes to disease-related protein aggregation. Additionally, the activation of JNK signaling by hyperactivated EPS8/RAC further promotes protein aggregation.”

Can the authors briefly speculate on how these processes might influence protein aggregation? For example, do the authors think that there are direct interactions with mutant proteins or modification of other factors pathways that suppress or enhance aggregation?

Minor comments

1. Figure 5b appears to be missing the DMSO and DUB inhibitor labels.

2. In the Discussion section (page 11), the authors state that: “In addition to elevated actin polymerization and JNK activity, we cannot discard that other potential mechanisms regulated by EPS8/RAC signaling also contribute to pathological protein aggregation.”

This is nice, but can the authors mention some examples of other RAC signalling targets here?

Reviewer #2

(Remarks to the Author)

A. Summary of the key results

The manuscript by Koyuncu and colleagues explores the cellular biology of protein quality control here in the context of models of Huntington's disease and amyotrophic lateral sclerosis. The authors build a case for the RAC protein EPS8 (and orthologues) being involved in the involved in the neurotoxicity associated with disease proteins in *C. elegans* and cellular iPSC models. RNA silencing and small molecule inhibitors are used to target EPS8.

1. The authors suggest that their treatment may enhance neuronal health. More specifically, the authors refer to neurotoxicity in their work, and in the case of the *C. elegans* models this is in reference to impaired movement phenotypes. Impaired movement can be caused by many things and there is no demonstration of altered neuronal morphology. To substantiate their claims claim, they should conduct experiments to assess neurodegeneration in *C. elegans* neurons. Both mutant and WT worm strains should be crossed with strains expressing a fluorescent marker for GABAergic motor neurons. Neurodegeneration, rather than thrashing behavior, would provide a more direct measure of neuronal health.

2. A statistical analysis of the filter-trap experiments is necessary. The claims made by the authors are difficult to assess without statistical validation of the experimental data.

3. The authors claim that their RNAi/treatment does not affect the total polyQ levels. However, some experiments show trends of increasing or decreasing polyQ levels, though high variability renders these results non-significant. In particular, for Figure 1, a fourth replicate of the SDS-PAGE analysis would enhance the robustness and validity of their claims.

4. The thrashing assay is a controversial and limited measure of *C. elegans* motor function. For scoring motility (body bends), how was this done? If this was done by manual counting then this approach is semiquantitative at best and should be examined with non-biased (automated) approaches. The reference (#2) in the methods refers to a review article. This is important because the range of difference is not huge, it is around 10-15 bends per 30 seconds which is not a huge difference. Also, importantly it does not appear that these experiments were done blind.

The motility data presented by the authors includes some surprising results:

- Figure 1h: The data shows that day 1 neuronal 67Q worms have 40 bends per 30 seconds, which is the same as the thrashing rate observed in day 5 neuronal 67Q worms. This raises two possibilities:
- The thrashing assay may not accurately correlate with worm aging, or

- Day 5 is not a sufficiently advanced time point to detect age-related motor deficits. If neuronal 67Q worms exhibit similar movement at both day 1 and day 5, it makes it hard to argue that the treatment rescues an age-related phenotype.
- Figure 5: Treatment with a DUB inhibitor on day 5 improves the thrashing score of mutant 67Q/FUS/TDP worms to levels exceeding those of their WT counterparts. Given the prior experience with the 19Q and 67Q strains, this seems improbable. This raises concerns about whether the assay is influenced by twitching phenotypes.
- All Figures: Typically, mutant strains display paralysis phenotypes during aging, which should be evident in the assays. However, no data point in the paper shows worms with significantly reduced movement. All data sets fall within a 10-20% range of movement, suggesting a lack of severely affected worms.

The interventions seem to improve the motility of the WT strains (polyQ, FUS, TDP-43) thus it appears to be a general mechanism and not limited to disease proteins. Are these interventions acting as a general motility inducer? What happens in N2 worms if *eps-2* (and others) are targeted?

Since targeting *mig-2* and *rac-2* seems to have positive effects, how does this work mechanistically? Is there no redundancy?

Additionally, all of the work is done using RNAi. Some additional validation using mutations (or auxin approaches) at least for *eps-8* needs to be done to confirm the findings.

Is not the JNK orthologue *jnk-1* in *C. elegans*? Why was *rgb-1* chosen.

The iPSC models are useful tools, but they lack the hallmarks of aging so interpretations should be cautious.

B. Originality and significance: if not novel, please include references

Reduction in protein aggregation does not always correlate with improved disease phenotypes or progression. Some aggregation-reducing therapies have failed to yield clinically relevant results, making this a controversial target. It is therefore crucial to provide compelling evidence that demonstrates reduced neuronal degeneration, improved motor function, or increased lifespan. The study provides a valuable example of translational confirmation, extending findings from *C. elegans* to cellular models, which is a notable strength of the paper. However, given that EPS8 is a known USP-modifier, the results presented in this work are not particularly surprising; it was expected that EPS8 would reduce the aggregation phenotype. Furthermore, considering that targeting protein aggregation has become a frequently tested and somewhat redundant approach in the clinical development of therapies for neurodegenerative diseases, the findings in this study may lack a sense of novelty or excitement of a novel drug target.

PMID: 36926679, PMID: 36158183

C. Data & methodology: validity of approach, quality of data, quality of presentation

There is room for improvement. Details are lacking about the thrashing assay and it appears to be a semiquantitative assay that may be prone to experimenter bias, and the experiments were not blinded. This is important since there are not large differences between the experimental groups. Additional assays looking at neurodegeneration would strengthen the manuscript.

D. Appropriate use of statistics and treatment of uncertainties. There is room for improvement, especially in considering the limitations of the thrashing assay. Also, the protein work (filter-trap assays) lack statistical validation.

E. Conclusions: robustness, validity, reliability.

The conclusions are not fully supported by the approaches, details in section A.

F. Suggested improvements: experiments, data for possible revision

Provide supplementary data on thrashing results at various ages to clearly demonstrate how aging affects the outcomes.

Conduct thrashing experiments (or something better) at older ages where mutant worms exhibit significant age-related motor impairment.

Include additional, unbiased and blinded measures of worm health. The thrashing assay used in this study is not robust regarding data reproducibility expected by journals and current research standards.

G. References: appropriate credit to previous work?

Seems OK, but a little narrow and does not touch upon other work using *C. elegans* for HD and ALS.

H. Clarity and context: lucidity of abstract/summary, appropriateness of abstract, introduction and conclusions. The manuscript is clear and logically presented. There seem to be some grammar syntax errors, so double-checking that would be a good idea.

Reviewer #3

(Remarks to the Author)

Previously the Vilchez lab discovered that the upregulation of EPS-8 hyperactivates RAC in muscle and neurons, and alters actin cytoskeleton and the activity of the protein kinase JNK (Koyuncu et al., Nature 2021). This pathway was also reported to regulate lifespan by modulating actin polymerization. In this study, which is a follow up of the aforementioned work, they focus on the question of whether the EPS-8/RAC pathway also controls toxic protein aggregation in worms and mammalian cultured cells.

In brief, they found that the knockdown of *eps-8*, *mig-2* or of *rac-2* by RNAi reduces the amounts of SDS-resistant aggregates of polyQ67-YFP that is expressed in neurons as well as of polyQ40-YFP which is expressed in muscles and of intestinal

polyQ44-YFP (Fig. 1). The reduced aggregation levels were associated with enhanced thrashing rates indicative of mitigated proteotoxicity. They crossed worms that express polyQ67-YFP in their neurons with animals that bear 3 mutations in the sequence of eps-8. These substitutions of lysine residues prevent the ubiquitination of EPS-8 and thus, stabilize this protein. The stabilization of EPS-8 leads to reduced thrashing rate.

Similarly, the knockdown of eps-8, mig-2 or of rac-2 mitigated the toxicity of two other neurodegeneration-linked, mutated aggregative proteins; FUS and TDP-43 (Fig. 2).

Next, they tested whether the EPS-8/RAC axis also regulates proteostasis in mammalian cells. Using HEK293 they discovered that the knockdown of eps-8 by shRNA leads to reduced aggregation of abnormally long polyQ stretches and of mutated FUS and TDP-43. It also reduced the level of apoptosis of motor neurons (Fig. 3D).

To test whether the modulation of protein aggregation by the EPS-8/RAC axis is associated with its roles in the formation and stabilization of actin filaments they used cytochalasin D. Their results show that the inhibition of actin polymerization and the knockdown of the kinase, KGB-1, reduce the aggregation of several aggregation-prone proteins (Fig. 4). Similar protection from proteotoxicity was seen when worms expressing the aggregative proteins were treated with a DUB inhibitor (Fig. 5). They also found that the deubiquitinase USP-4 regulates the levels of EPS-8 and subsequently, the rate of protein aggregation and level of proteotoxicity in worms (Fig. 6) and in human cells (Fig. 7).

These results culminate to illustrate a proteostasis regulating roles of the EPS-8/RAC pathway that link the integrity of actin filaments with aging-associated increased levels of EPS-8 which result from high levels of de-ubiquitination of this protein. This study is interesting, timely and expand the current knowledge about the EPS-8/RAC pathway and the integrity of the proteome. Yet, several issues should be addressed to enhance clarity and substantiate some aspects of this manuscript.

Major comments:

1. In this work, the authors heavily rely on the “filter trap” assay to monitor protein aggregation. While this technique is widely used it only provides an assessment of SDS-resistant aggregates (probably large aggregates). Since oligomers have been shown to be the most toxic species (Silveira et al., Nature 2005 Vol. 437 Issue 7056 Pages 257-61; Shankar et al., Nat Med. 2008 Aug;14(8):837-42), it is necessary to test whether the knockdown of eps-8 modulates the levels of these conformers – or perhaps the reduced levels of SDS-resistant aggregates are associated with increase in oligomers. This can be done by ultra-centrifugation; however, I think that it would be easier and sufficient to use native gels that were optimized by the Nollen lab to follow different conformers of polyQ-YFP stretches in nematodes (Holmberg and Nollen, Methods Mol Biol 2013 Vol. 1017 Pages 193-9).

2. To test the relevance of their findings to mammals, the authors use HEK293 cells (Fig. 3, except for 3d). Cytoskeletal filaments were reported to be involved in the shuttling and deposition of aggregation-prone proteins, for instance in the formation of aggresomes (Johnston et al., J Cell Biol. 1998 Dec 28;143(7):1883-98). Since the cytoskeleton is expected to be affected by the EPS8/RAC pathway, it is important to test how knocking down the activity of this pathway affects the distribution of toxic aggregates in these cells. Immuno fluorescence can be used to address this.

This is particularly important as it seems that the HEK293 cells were transiently transfected (as explained at the methods section). This method generates very variable cell populations, and some cells may express very high levels of the aggregative proteins.

3. In figure 3e, the authors control for the over-expression of eps-8 but no such control appears in figure 3f. This is an important control as the usage of strong viral promoters (CMV promoter?) could result in the depletion of transcription factors and in the generation of artifacts. In addition, while they state that the expression of TDP-43-A382T was driven by the pLVX-Puro-TDP-43-A382T plasmid, I could not find in the methods section, how TDP-43-M337V, which is shown in figure 3f, was expressed.

4. To test whether the knockdown of eps-8 ameliorates neurodegeneration they derived motor neurons from iPSCs that express mutated FUS (P525L) and concluded that: “However, knockdown of EPS8 ameliorated neurodegeneration in these cells” (page 7 and figure 3d). This statement is based on a single assay in which they followed the levels of cleaved caspase 3, to compare the rate of apoptosis. I do not think that the results presented at figure 3d are sufficient to support their claim. Motor neurons die in ALS patients by additional mechanisms including necroptosis (Neuron. 2014 Mar 5;81(5):1001-1008) and autophagy dependent cell death (see for review: Cell Death Discov. 2024 Jun 19;10(1):291).

Therefore, to support their claim they should at least test whether the knockdown of eps8 induces necroptosis (Methods Mol Biol. 2021;2255:119-134).

5. What are the effects of modulating the EPS8/RAC pathway on protein degradation pathways in the context of constant proteotoxic insult? It has been already shown that the deubiquitinase USP11 controls autophagy (Basic et al., ref. 37). Does the knockdown of eps-8 affect general activity of the ubiquitin proteasome system and/or autophagy when the worm is challenged by a proteotoxic protein?

6. I think that it would enhance the manuscript if they further elaborate in the discussion on the known roles of the cytoskeleton in the maintenance of proteostasis and explain how actin filaments are related. For instance, microtubule have been reported to be critical for the formation of aggresomes and vimentin was shown to cage these structures (Johnston and Kopito J Cell Biol. 1998 Dec 28;143(7):1883-98). Ataxin 2, a protein which is involved in the onset of ALS, is involved in the regulation of cytoskeleton stability (Del Castillo et al., iScience 2021 Dec 3;25(1):103536). In addition, anc-1 which encodes a protein that is important for actin binding and cytoskeleton organization has been found to regulate proteostasis in worms and to modulate proteasome activity (Levine et al., Aging Cell 2019 Dec;18(6):e13047).

Minor comments:

7. While they state that the expression of TDP-43-A382T was driven by the pLVX-Puro-TDP-43-A382T plasmid, I could not find in the methods section, how TDP-43-M337V, which is shown in figure 3f, was expressed.

8. At figure 3g they show that the activation of RAC by a “RAC activator” enhances the aggregation of polyQ100HTT. To assess the specificity of this treatment I wondered what activator was used. Surprisingly, the only information that they provide at the method section is: “For RAC activator experiments, the cells were treated with 2 units/ml RAC activator for 6 hours”. This is unacceptable, as the description MUST be sufficient for other scientists to repeat the reported experiments. Please specify, where the “RAC activator” was obtained this compound from. Vendor and catalogue number of this activator must be mentioned.

Version 2:

Reviewer comments:

Reviewer #1

(Remarks to the Author)

Koyuncu et al have responded thoughtfully and comprehensively to all my comments, questions and criticisms. The resulting revised manuscript is now more insightful and accurate, and the core findings and conclusions are more compelling. Specifically, the authors have performed a high number of additional experiments to measure the effects of EPS-8/RAC/USP-4 on neuronal function in different models of protein aggregation/toxicity. In addition, they have shown that these effects are also relevant to muscle tissues and have revised the statistical tests used in several figures as appropriate. Lastly, the authors have expanded the discussion to provide possible routes by which excessive actin polymerization/JNK may influence protein aggregation. The work is original and important and I recommend that it is accepted for publication in Nature Aging.

Reviewer #2

(Remarks to the Author)

This is an improved manuscript. My major concern last time was the lack of unbiased experimental research methods. Last time, the question of whether worm behavioral experiments were blinded to the investigator was raised. In this regard, I am pleased that they introduced unbiased measures of motility/thrashing.

However, they add the nose-touch assay here, which seems to have been conducted without blinding (Fig. 1b). This raises a concern, given that the effects are small (better at day 7). The assay demonstrates stronger effects in later experiments (Fig. 2f, h); however, conducting these blindly would greatly enhance confidence in the results. The fact that these experiments were not blinded should be explicitly stated, particularly for assays where the experimenter actively participates, such as touching the animal versus using automated recording approaches. Not having all the worm experiments conducted blind to strain/genotype would be a significant improvement. Personally, I believe it should be mandatory for modern publishing standards.

The introduction of scoring for neurodegeneration is good. The results for the TDP-43 strains are convincing. The lack of neurodegeneration phenotypes in the Q67 strain is surprising. I doubt that observation will stand up to scrutiny, but that may be clarified by future research.

Reviewer #3

(Remarks to the Author)

I have read the revised manuscript as well as the rebuttal letter and found that the authors have comprehensively addressed the critique. They tested the physiological relevance of the knockdown of USP-4/EPS-8/RAC signaling using two additional assays (namely, nose-touch response and chemotaxis), conducted control experiments in worms and cells and corrected their statistical analyses.

The text has been also corrected and expanded and missing details in the methods section have been added (i.e catalogue numbers).

In sum, I think that in the current version of the manuscript, the results are more comprehensive, the conclusions are properly supported and the text much clearer. Therefore, I recommend acceptance of the current version of the manuscript.

Reviewers' Comments:

Reviewer #1:

Remarks to the Author:

General summary of the work

In this manuscript, Koyuncu et al. build on previous observations (from this group) that, with age, reduced ubiquitination of EPS-8 leads to its accumulation, thereby hyperactivating RAC signalling and shortening lifespan. Here the authors extend these findings by showing that knockdown of EPS-8/RAC components (*eps-8*, *mig-2*, *rac-2*) strongly suppresses the age-related aggregation of disease associated proteins (polyglutamine, FUS and TDP-43) in neurons, muscles and intestine. Conversely, ubiquitylation-resistant (Ub-less) mutants of EPS-8 accelerate the aggregation of these proteins, with similar effects observed in HEK293 cells upon depletion or overexpression of EPS8. The authors show that treatment with cytochalasin D (an inhibitor of actin-polymerization) or knockdown of *kqb-1*/JNK blocks the increased protein aggregation that occurs in *C. elegans* and HEK293 cells upon exposure to Ub-less EPS-8 or EPS8 overexpression, respectively. Finally, the authors show that these effects are dependent on DUB activity, and pinpoint USP-4 as the key DUB acting on EPS-8/EPS8 to control its degradation and subsequent impact on RAC signalling and protein aggregation.

I like the work and find it to be an interesting and informative extension of the groups previous findings. The effects on protein aggregation are very convincing and the conservation of the core mechanism from worms to HEK293 cells suggests a general importance of the underlying biology. The manuscript is clear and compelling, and I agree with the conclusions presented. The work is appropriate for Nature Aging, but there are some areas of the manuscript that could be improved and/or expanded. In addition, I have some issues with the statistics used in some of the figure panels. While I do not expect this to alter the core conclusions presented, I would still like the authors to address these.

Major comments

1. In Figure 1, the effects of *eps-8*, *mig-2* and *rac-2* RNAi on aggregation in neurons, muscle and intestine are very clear and convincing. Similarly, the effects on thrashing in neuronal Q67 and muscle Q40 animals look good. However, knockdown of *eps-8*, *mig-2* and *rac-2* also increases thrashing in Q19 worms (Fig. 1b). This makes me wonder if knocking down EPS-8/RAC components makes worms generally more hyperactive, rather than specifically suppressing Q67 or Q40 toxicity in neurons/muscles. Does *eps-8*, *mig-2*, *rac-2* RNAi also enhance thrashing in Q0::YFP

worms or wildtype worms? If so, the authors should use another measure of neuronal toxicity/function in Q67 worms, such as chemotaxis.

We appreciate the Reviewer's insightful comment, which was also raised by Reviewer #2. In our initial submission, we focused on thrashing deficits because is the most studied disease-related phenotype in polyQ and FUS-expressing worms (Brignull et al., *J. Neurosci.*, 2006, PMID: 16855087; Murakami et al., 2012, PMID: 21949354). However, we fully acknowledge the Reviewers' concern that this phenotype is not the most specific measure of neuronal dysfunction in the context of USP-4/EPS-8/RAC activity.

In our previous work, we found that EPS-8/RAC signaling plays a role not only in neurons but also in muscle function during aging in wild-type worms. We observed that reducing hyperactivated EPS-8/RAC signaling prevents actin destabilization in muscle cells with age (Koyuncu et al., *Nature* 2021. PMID: 34321666). Consequently, EPS-8/RAC knockdown delays age-related muscle dysfunction and helps maintain motility in wild-type worms during aging (Koyuncu et al., *Nature* 2021. PMID: 34321666). These findings suggest that reducing EPS-8/RAC signaling does not induce general hyperactivity in wild-type worms but rather prevents age-related motility decline. For instance, knockdown of EPS-8/RAC signaling did not affect motility in day-1 adult wild-type worms but significantly improved motility in older worms (**Figure R1**).

Figure R1. Knockdown of *eps-8* ameliorates age-related motility in wild-type worms.

Data are mean \pm s.e.m. thrashing movements over a 30-s period on day 1 ($n = 30$ worms per condition, three independent experiments), day 3 ($n = 30$ worms per condition, three independent experiments) and day 10 ($n = 45$ worms per condition, three independent experiments) of adulthood. Knockdown of *eps-8* after development ameliorates the age-associated decline in motility (day 1 vector RNAi versus day 1 *eps-8* RNAi, $P = 0.2127$; day 3 Vector RNAi versus day 3 *eps-8* RNAi, $P < 0.0001$; day 10 Vector RNAi versus day 10 *eps-8* RNAi, $P < 0.0001$). P values were determined by two-sided t -test. Data from Koyuncu et al, Nature 2021; PMID: 34321666.

In our initial submission, as expected, we observed that knockdown of *eps-8* or RAC orthologs improves motility in control polyQ19 worms at day 5 of adulthood.

Similarly, we found that reducing EPS-8/RAC signaling increased thrashing rates in control day-5 adult worms expressing wild-type FUS and TDP-43. Since knockdown of *eps-8* and *RAC* orthologs restored motility deficits in polyQ67 and ALS worm models to levels comparable to their respective controls, we concluded that these treatments prevent disease-related changes in neurons. After considering the Reviewers' comments, we now recognize that the motility data alone are difficult to interpret, given the beneficial effects of EPS-8/RAC downregulation on aging muscle cells. We apologize for making this claim in our initial submission without additional supporting data.

To further strengthen our conclusions and discard that reducing EPS-8/RAC signaling results in hyperactive worms, we have now performed thrashing assays in day-1 wild-type, control Q19 and Q67 worms. In parallel, we have repeated all the experiments in day-5 animals to perform automated quantification as suggested by Reviewer #2. We confirmed that polyQ67 worms exhibited reduced motility compared to control Q19 worms at day 5 but not at day 1 (**Extended Data Fig. 2a**). Furthermore, knockdown of *eps-8* and *RAC* orthologs had no effect at day 1 but improved motility in polyQ67, Q19, and wild-type worms at day 5 (**Extended Data Fig. 2a,b**). Since EPS-8/RAC knockdown restored motility to levels comparable to control Q19 worms under the same treatment (**Extended Data Fig. 2a**), these results indicate an effect on disease-related motility deficits. However, we now explicitly acknowledge that these results are difficult to interpret due to the general benefits of EPS-8/RAC downregulation on muscle cells during aging (PMID: 34321666). The revised text now says: "The accumulation of polyQ aggregates in *C. elegans* neurons impairs neuronal function^{24, 29}. The most studied phenotype is loss of motility, which correlates with aggregate levels and age^{9, 24, 25, 30}. Indeed, polyQ67 worms exhibited a decline in motility compared to control polyQ19 worms at day 5 of adulthood, but not at day 1 (**Extended Data Fig. 2a**). While knockdown of either *eps-8* or *RAC* orthologs had no effect in young worms, it reduced motility deficits in aged polyQ67 worms (**Extended Data Fig. 2a**). Previously, we found that lowering EPS-8/RAC signaling not only has effects in neurons but also delays age-related muscle dysfunction in wild-type animals, preventing motility decline during aging¹⁴. Consistently, loss of *eps-8* and *RAC* orthologs improved motility in control polyQ19 and wild-type animals at day 5 of adulthood (**Extended Data Fig. 2a,b**). Although *eps-8* and *RAC* knockdown rescued motility deficits to levels similar to control Q19 worms under the same treatment (**Extended Data Fig. 2a**), these results were difficult to interpret due to the beneficial effects of EPS-8/RAC downregulation in aging control animals (**Extended Data Fig. 2a,b**)".

Similarly, we have now observed that knockdown of either *eps-8* or *RAC* orthologs does not affect motility in worms expressing mutant TDP-43 at day 1 of adulthood (**Extended Data Fig. 3c**). However, by day 5 of adulthood, it rescues motility deficits in these worms, restoring motility to levels similar to those of age-matched wild-type worms or control TDP-43 worms under the same treatment (**Extended Data Fig. 3c**).

Moreover, we have repeated the thrashing assays using automated quantification in polyQ67 worms expressing mutant Ub-less EPS-8, including two different ages (day 1 and day 3). We confirmed that Ub-less EPS-8 induces disease-related motility decline in young worms as early as day 1 of adulthood (**Extended Data Fig. 2f**).

We have also repeated the motility experiments using automated analysis in polyQ, TDP-43 and FUS models following DUB inhibition and *usp-4* knockdown, now monitoring worms at different ages (**Extended Data Fig. 9c-f, 10c-e**). For instance, we assessed worms under *usp-4* RNAi treatment at days 1, 5, and 7 of adulthood. Similar to *eps-8* knockdown, *usp-4* RNAi had no effect on motility at day 1 but improved motility only at older ages. We found that loss of *usp-4* not only improves age-related motility deficits in control worms but also rescues the deficits caused by disease-related mutant proteins, restoring motility to levels similar to those of control worms under the same treatment (**Extended Data Fig. 10c-e**). Since the thrashing assays provide valuable information, we have included these data in the revised manuscript (now presented as **Extended Data Figures**).

In addition to motility deficits, we have now performed lifespan assays. Worms expressing polyQ67 repeats exhibited a shorter lifespan compared to control polyQ19 worms (**Extended Data Fig. 2c, 10b**). We observed that either *eps-8* or *usp-4* knockdown extends lifespan in polyQ67-expressing worms (**Extended Data Fig. 2c, 10b**). However, since loss of *eps-8* and *usp-4* also extends lifespan in control polyQ19 worms (**Extended Data Fig. 2c, 10b**), we could not attribute this effect specifically to disease-related changes. In summary, given that aging hastens disease-related phenotypes and *usp-4/eps-8* knockdown delays aging, it is difficult to ascribe a specific effect of lowering USP-4/EPS-8/RAC signaling on preventing disease-related phenotypes such as shortened lifespan and motility deficits.

Given that lifespan and thrashing assays alone are insufficient to support a specific role of USP-4/EPS-8/RAC signaling in disease-related neuronal dysfunction, we have now performed two different behavioral assays (i.e. nose-touch response and chemotaxis) in polyQ and ALS models, under various conditions (*eps-8/mig-2/rac-2/usp-4* knockdown, Ub-less EPS-8 expression, loss-of-function *eps-8* mutant, DUB inhibition) and at different ages (day 1, 5, and 7).

Importantly, the disease models exhibited normal behavioral responses on day 1 but developed significant deficits in nose-touch response and chemotaxis with age. Our treatments did not affect behavioral responses in control worms at the different ages tested, but they specifically rescued or hastened the age-related decline in nose-touch response and chemotaxis observed in Q67 and ALS models (**Fig. 1b,c,e,f; Fig. 2f-i; Fig. 6c-e; Fig. 7c-g; Extended Data Fig. 2d-e, Extended Data Fig. 3d-e, Extended Data Fig. 9a-b; Extended Data Fig. 10a**). These findings provide strong evidence that USP-4/EPS-8/RAC signaling influences disease-related neuronal dysfunction in *C. elegans* during aging. The revised text now says: "Nose-touch avoidance behavior is mediated by sensory neurons located in the head of the worm.

On the first day of adulthood, polyQ67-expressing worms responded to nose touch similarly to control polyQ19 and wild-type worms (**Fig. 1b**). However, polyQ67 worms exhibited a decline in nose-touch response compared to control animals at older ages (**Fig. 1b**). Notably, knockdown of *eps-8* or *RAC* orthologs rescued this age-related functional decline in polyQ67 worms, but had no effect on aging control animals (**Fig. 1b**). PolyQ aggregation also induces neurotoxicity in chemosensory neurons, leading to impaired chemotaxis responses³¹. While polyQ67 worms exhibited normal chemotaxis towards benzaldehyde on day 1 of adulthood, they developed chemotaxis deficits with age (**Fig. 1c**). However, knockdown of *eps-8* and *RAC* orthologs mitigated this decline in polyQ67 worms without affecting chemotaxis behavior in control animals (**Fig. 1c**). Similarly, *eps-8* knockout mutation ameliorated the age-related decline in nose-touch responses and chemotaxis caused by polyQ67 expression (**Extended Data Fig. 2d-e**). (...) We observed that the expression of ubiquitin-less EPS-8 accelerates polyQ67 aggregation and disease-related behavioral changes from day 1 of adulthood (**Fig. 1d-f and Extended Data Fig. 2f**). (...) Importantly, we observed that DUB inhibitor also attenuates protein aggregation and behavioral deficits in *C. elegans* disease models (**Fig. 6a-e and Extended Data Fig. 9a-f**). (...) In addition to polyQ-expanded peptides, *usp-4* knockdown also prevented the aggregation of ALS-related mutant proteins (**Fig. 7a,b**). Consistent with this reduction in aggregation, *usp-4* knockdown rescued both nose-touch response and chemotaxis deficits in polyQ and ALS models during aging, while having no effect on these behavioral responses in control worms (**Fig. 7c-g and Extended Data Fig. 10a**)”.

2. Fig 1g and h, the effects of ub-less EPS-8 on polyQ aggregation and toxicity in neurons are very nice. Does this mutant also accelerate polyQ aggregation and toxicity in muscle tissue?

We have now tested whether Ub-less mutant EPS-8 accelerates polyQ aggregation and toxicity in worms expressing polyQ-expanded peptides specifically in muscle cells. We found that Ub-less EPS-8 expression accelerates both aggregation and motility deficits in these worms (please see **Extended Data Fig. 3a,b**). However, unlike neuronal polyQ-expanded models (**Fig. 1d and Extended Data Fig. 2c**), where we observed effects from day 1 of adulthood, the detrimental impact of Ub-less mutant EPS-8 on aggregation and motility in muscle cells started from day 3 of adulthood (**Extended Data Fig. 3a,b**).

3. In Fig. 2b, the effects of EPS-8/RAC RNAi on thrashing in FUS and TDP-43 mutants look convincing. However, as with the polyQ experiments, the RNAi treatments also enhance thrashing in worms expressing WT FUS. Are there other measures of neuronal health that can be used here to show that the RNAi is really ameliorating toxicity in neurons and not just making worms hyperactive?

As discussed in Comment #1, we completely agree with the Reviewer that thrashing assays alone are insufficient to assess a specific role of EPS-8/RAC signaling in disease-related neuronal dysfunction. To address this, we have now conducted two additional behavioral assays (i.e. nose-touch response and chemotaxis) in the ALS FUS and TDP-43 models, under various treatments (*eps-8/mig-2/rac-2/usp-4* knockdown, Ub-less EPS-8 expression, DUB inhibition) and at different ages (day 1, 5, and 7).

Similar to polyQ67 worms, the ALS disease models exhibited normal behavioral responses on day 1 but developed significant deficits in nose-touch response and chemotaxis with age. Importantly, our treatments did not affect these behavioral responses in control worms or disease models at day 1 of adulthood. Moreover, they did not affect behavioral responses in control worms at older ages. However, our treatments specifically rescued the age-related decline in nose-touch response and chemotaxis in polyQ and ALS models (**Fig. 1b-c; Fig. 2f-i; Fig. 6c-e; Fig. 7c-g; Extended Data Fig. 2d-e, Extended Data Fig. 3d-e, Extended Data Fig. 9a-b; Extended Data Fig. 10a**). These findings support a role of USP-4/EPS-8/RAC signaling in disease-related neuronal dysfunction during aging.

4. In Fig. 5c and d, The DUB inhibitor appears to make the Q67 and FUS P525L groups even better than their corresponding Q19 or WT FUS groups. Are these differences statistically significant? If so, can the authors please comment on this in the text (i.e. briefly speculate why this might be the case).

As discussed above, we have now repeated the motility assays in control, Q67, FUS^{P525L}, and TDP-43^{M337V} worms treated with DUB inhibitor to perform automated quantification analysis (**Extended Data Fig. 9c-f**). In these experiments, the motility of Q67 worms treated with DUB inhibitor was not higher compared to their corresponding controls under the same treatment (**Extended Data Fig. 9c,d**). In the case of FUS models, although the motility in FUS^{P525L} worms treated with DUB inhibitor appeared to be higher compared to FUS^{WT} worms under the same treatment (**Extended Data Fig. 9e**), the difference was not significant: FUS^{WT} + DUB inhibitor vs FUS^{P525L} + DUB inhibitor, P value = 0.1116 (two-way ANOVA with Fisher's LSD test). Likewise, the motility in TDP-43^{M337V} worms treated with DUB inhibitor was not significantly higher compared to TDP-43^{WT} under the same treatment (**Extended Data Fig. 9f**): TDP-43^{WT} + DUB inhibitor vs TDP-43^{M337V} + DUB inhibitor, P value = 0.1828 (two-way ANOVA with Fisher's LSD test). Given the lack of statistical significance, we have not commented on this in the text.

5. The statistical tests used throughout the Figures are incorrect. Two-tailed, paired t-tests are often used even though samples are not paired. In addition, there are many instances where two-tailed t-tests are used to compare > 2 means or when 2 independent variables (e.g. genotype AND RNAi treatment) are being compared.

Moreover, there are several examples where the groups being compared do not have a normal distribution, as values for the control groups have all been set at 1. The authors should apply the correct statistical tests to the hypotheses being tested in all cases. This should be as follows:

One-way ANOVA with Dunnett's comparison to vector RNAi control: Figure 1a,c,d,e, f, g; Figure 2a and c; Figure 3a, b and c; Figure 7c, e and g.

Two-way ANOVA with Fisher's LSD test or equivalent: Figure 5b-e; Figure 6c and f; Figure S2.

Two-way ANOVA with pairwise comparisons of means: Figure 1b; Figure 2b and d;

Non-parametric tests should be used (e.g. Wilcoxon) – Figure 3g; Figure 5a; Figure 6b,d,e and g; Figure 7f.

I should stress that given the differences observed, the number of comparisons being made, and the distributions of the data presented, I do not expect this to alter the authors' conclusions significantly. However, it will still be good practice to employ the correct statistical tests.

We sincerely thank the Reviewer for not only identifying this issue but also for providing the appropriate tests for each figure. Following the Reviewer's detailed guidance, we have now applied the correct statistical tests to all the figures from our initial submission, as well as the new experiments performed during the revision. Importantly, these adjustments did not alter the conclusions of our manuscript.

Regarding the Wilcoxon test, this analysis always yields a *P* value greater than 0.05 when applied to five or fewer values, regardless of the sample median's deviation from the hypothetical median. Thus, we have now conducted additional independent experiments for all figures in our initial submission that required a Wilcoxon test, ensuring a minimum of six data points per condition.

6. In the Discussion section (page 10), the authors state that: "Our data suggest that excessive actin polymerization triggered by elevated EPS8/RAC activity contributes to disease-related protein aggregation. Additionally, the activation of JNK signaling by hyperactivated EPS8/RAC further promotes protein aggregation."

Can the authors briefly speculate on how these processes might influence protein aggregation? For example, do the authors think that there are direct interactions with mutant proteins or modification of other factors' pathways that suppress or enhance aggregation?

We have now speculated on various direct and indirect mechanisms by which excessive actin polymerization and JNK activity may influence protein aggregation. The Discussion section now says: “Our data indicate that EPS8/RAC signaling contributes to protein aggregation through different pathways. EPS8/RAC modulates cellular processes such as actin polymerization and JNK signaling, both of which have been implicated in neurodegenerative diseases⁵⁵⁻⁵⁸. We found that excessive actin polymerization, driven by hyperactivated EPS8/RAC signaling, contributes to disease-related protein aggregation. Additionally, elevated EPS8/RAC hyperactivates JNK signaling, further promoting protein aggregation. However, the precise mechanisms by which excessive actin polymerization and JNK activity drive protein aggregation remain unknown.

While HTT and ALS-related proteins, including the polyQ-containing protein ataxin-2, regulate actin dynamics^{59, 60}, previous studies have indicated that actin filaments and actin-binding factors may also influence pathological protein aggregation^{55, 61}. For instance, distinct familial ALS cases that exhibit wild-type TDP-43 aggregates are associated with mutations in actin cytoskeleton regulators such as profilin 1⁵⁵. We speculate that age-related destabilization of actin filaments may affect protein aggregation by impairing essential cellular processes, thereby reducing the cellular capacity to prevent protein aggregation. In *C. elegans*, knockdown of *anc-1*, which encodes a protein involved in actin binding and cytoskeleton organization, alters the expression of transcription factors and E3 ubiquitin ligases, leading to polyQ aggregation⁶².

Importantly, mutant HTT aggregates can co-localize with actin filaments⁵⁷. Although we did not observe changes in the intracellular distribution of mutant HTT aggregates following EPS8 knockdown in human cells (**Extended Data Fig. 12**), we cannot exclude the possibility that the actin cytoskeleton directly influences aggregation through its interaction with disease-related proteins. For instance, redistribution of the intermediate protein vimentin contributes to the assembly of aggresomes containing cystic fibrosis transmembrane conductance regulator (CFTR), whereas disruption of microtubules blocks aggresome formation⁶³. In previous work, we observed that age-related changes in actin filaments lead to aggregation of actin protein itself¹⁴. This raises the intriguing possibility that actin aggregates may act as a niche for the accumulation of disease-related proteins. Alternatively, actin aggregates could sequester molecular chaperones and other components of the proteostasis network, leading to its collapse and subsequent aggregation of pathological proteins.

Likewise, hyperactivation of JNK may influence pathological protein aggregation through different mechanisms. The JNK pathway is involved in the response to proteotoxic stresses, such as heat and oxidative stress. Moreover, JNK triggers phosphorylation cascades that modulate distinct regulatory proteins in the mitochondria and nucleus, including SMAD4, p53, c-JUN, ATF2, ELK1 and HSF1^{64, 65}. Thus, JNK hyperactivation during aging may lead to cellular alterations, promoting protein aggregation. In addition, these downstream targets of JNK signaling could directly affect the activity of proteostasis mechanisms”.

Minor comments

1. Figure 5b appears to be missing the DMSO and DUB inhibitor labels.

We thank the Reviewer for bringing this to our attention. We have now added the labels for DMSO and the DUB inhibitor (please see **Fig. 6b**).

2. In the Discussion section (page 11), the authors state that: “In addition to elevated actin polymerization and JNK activity, we cannot discard that other potential mechanisms regulated by EPS8/RAC signaling also contribute to pathological protein aggregation.”

This is nice, but can the authors mention some examples of other RAC signalling targets here?

This is an important point. We have now included examples of other targets and pathways regulated by RAC signaling. The text now says: “Beyond elevated actin polymerization and JNK activity, we cannot discard that other mechanisms regulated by EPS8/RAC signaling contribute to protein aggregation. For instance, RAC regulates additional pathways, including p38 MAPK, PI3K/Akt/mTOR, and STAT signaling^{41, 66, 67}. Moreover, RAC influences reactive oxygen species (ROS) production⁶⁸, which could play a role in pathological aggregation”.

Reviewer #2:

Remarks to the Author:

A. Summary of the key results

The manuscript by Koyuncu and colleagues explores the cellular biology of protein quality control here in the context of models of Huntington's disease and amyotrophic lateral sclerosis. The authors build a case for the RAC protein EPS8 (and orthologs) being involved in the neurotoxicity associated with disease proteins in *C. elegans* and cellular iPSC models. RNA silencing and small molecule inhibitors are used to target EPS8.

1. The authors suggest that their treatment may enhance neuronal health. More specifically, the authors refer to neurotoxicity in their work, and in the case of the *C. elegans* models this is in reference to impaired movement phenotypes. Impaired movement can be caused by many things and there is no demonstration of altered neuronal morphology. To substantiate their claims, they should conduct experiments to assess neurodegeneration in *C. elegans* neurons. Both mutant and WT worm strains should be crossed with strains expressing a fluorescent marker for GABAergic motor neurons. Neurodegeneration, rather than thrashing behavior, would provide a more direct measure of neuronal health.

We appreciate the Reviewer's thoughtful comment, which was also raised by Reviewer #1. In our initial submission, we focused on thrashing deficits because it is the most studied disease-related phenotype in polyQ- and FUS-expressing worms (Brignull et al., *J. Neurosci.*, 2006, PMID: 16855087; Murakami et al., 2012, PMID: 21949354). However, we fully acknowledge the Reviewers' concern that this phenotype is not the most specific measure of neuronal dysfunction in the context of USP-4/EPS-8/RAC activity.

In our previous work, we found that EPS-8/RAC signaling plays a role not only in neurons but also in muscle function during aging in wild-type worms. We reported that reducing EPS-8/RAC signaling prevents actin destabilization in muscle cells of wild-type animals as they age (Koyuncu et al., *Nature* 2021. PMID: 34321666). Consequently, *EPS-8/RAC* knockdown delays age-related muscle dysfunction and helps maintain motility in wild-type worms during aging (Koyuncu et al., *Nature* 2021. PMID: 34321666). For example, *EPS-8/RAC* knockdown did not affect motility in day 1 adult wild-type worms but significantly improved motility in older worms (**Figure R1**). Together, these findings suggest that reducing EPS-8/RAC signaling can prevent age-related motility decline in wild-type animals.

Figure R1. Knockdown of *eps-8* ameliorates age-related motility in wild-type worms. Data are mean \pm s.e.m. thrashing movements over a 30-s period on day 1 ($n = 30$ worms per condition, three independent experiments), day 3 ($n = 30$ worms per condition, three independent experiments) and day 10 ($n = 45$ worms per condition, three independent experiments) of adulthood. Knockdown of *eps-8* after development ameliorates the age-associated decline in motility (day 1 vector RNAi versus day 1 *eps-8* RNAi, $P = 0.2127$; day 3 Vector RNAi versus day 3 *eps-8* RNAi, $P < 0.0001$; day 10 Vector RNAi versus day 10 *eps-8* RNAi, $P < 0.0001$). P values were determined by two-sided t -test. Data from Koyuncu et al, Nature 2021; PMID: 34321666.

In our initial submission, as expected, we observed that knockdown of *eps-8* or RAC orthologs improves motility in control polyQ19 worms at day 5 of adulthood. Similarly, we found that reducing EPS-8/RAC signaling increased thrashing rates in control day-5 adult worms expressing wild-type FUS and TDP-43. Since knockdown of *eps-8* and RAC orthologs restored motility deficits in polyQ67 and ALS worm models to levels comparable to their respective controls, we concluded that these treatments prevent disease-related changes in neurons. After considering the Reviewers' comments, we now recognize that the motility data alone are difficult to interpret, given the beneficial effects of EPS-8/RAC downregulation on aging muscle cells. We apologize for making this claim in our initial submission without additional supporting data.

We have now repeated all the motility experiments in day-5 adult animals to perform automated quantification as suggested by the Reviewer in their comments below. To further strengthen our conclusions and discard that reducing EPS-8/RAC signaling results in hyperactive worms, we have also performed thrashing assays in day-1 wild-type, control Q19 and Q67 worms. We confirmed that polyQ67 worms exhibit reduced motility compared to control Q19 worms at day 5 but not at day 1 (**Extended Data Fig. 2a**). Furthermore, knockdown of *eps-8* and RAC orthologs had no effect at day 1 but improved motility in wild-type, Q19, and Q67 worms at day 5 (**Extended Data Fig. 2a,b**). Since EPS-8/RAC knockdown restored motility to levels comparable to control Q19 worms under the same treatment (**Extended Data Fig. 2a**), these results indicate an effect on disease-related motility deficits. However, we now explicitly acknowledge that these results are difficult to interpret due to the general benefits of EPS-8/RAC downregulation on muscle cells during aging (PMID:

34321666). The revised text now says: “The accumulation of polyQ aggregates in *C. elegans* neurons impairs neuronal function^{24, 29}. The most studied phenotype is loss of motility, which correlates with aggregate levels and age^{9, 24, 25, 30}. Indeed, polyQ67 worms exhibited a decline in motility compared to control polyQ19 worms at day 5 of adulthood, but not at day 1 (**Extended Data Fig. 2a**). While knockdown of either *eps-8* or RAC orthologs had no effect in young worms, it reduced motility deficits in aged polyQ67 worms (**Extended Data Fig. 2a**). Previously, we found that lowering EPS-8/RAC signaling not only has effects in neurons but also delays age-related muscle dysfunction in wild-type animals, preventing motility decline during aging¹⁴. Consistently, loss of *eps-8* and *RAC* orthologs improved motility in control polyQ19 and wild-type animals at day 5 of adulthood (**Extended Data Fig. 2a,b**). Although *eps-8* and RAC knockdown rescued motility deficits to levels similar to control Q19 worms under the same treatment (**Extended Data Fig. 2a**), these results were difficult to interpret due to the beneficial effects of EPS-8/RAC downregulation in aging control animals (**Extended Data Fig. 2a,b**)”.

Similarly, we have now observed that knockdown of either *eps-8* or RAC orthologs does not affect motility in worms expressing mutant TDP-43 at day 1 of adulthood (**Extended Data Fig. 3c**). However, by day 5 of adulthood, it rescues motility deficits in these worms, restoring motility to levels similar to those of age-matched wild-type worms or control TDP-43 worms under the same treatment (**Extended Data Fig. 3c**).

Moreover, we have repeated the thrashing assays using automated quantification in polyQ67 worms expressing mutant Ub-less EPS-8, including two different ages (day 1 and day 3). We confirmed that Ub-less EPS-8 induces disease-related motility decline in young worms as early as day 1 of adulthood (**Extended Data Fig. 2f**).

We have also repeated the motility experiments to perform automated analysis in polyQ, FUS, and TDP-43 models following DUB inhibition and *usp-4* knockdown, now monitoring worms at different ages (**Extended Data Fig. 9c-f, 10c-e**). For instance, we assessed worms under *usp-4* RNAi treatment at days 1, 5, and 7 of adulthood. Similar to *eps-8* knockdown, *usp-4* RNAi had no effect on motility at day 1 but improved motility only at older ages. We found that loss of *usp-4* not only improves age-related motility deficits in control worms but also rescues the deficits caused by disease-related mutant proteins, restoring motility to levels similar to those of control worms under the same treatment (**Extended Data Fig. 10c-e**). Since the thrashing assays provide valuable information, we have included these data in the revised manuscript (now presented as **Extended Data Figures**).

In addition to motility deficits, we have now performed lifespan assays. Worms expressing polyQ67 repeats exhibited a shorter lifespan compared to control polyQ19 worms (**Extended Data Fig. 2c, 10b**). We observed that either *eps-8* or *usp-4* knockdown extends lifespan in polyQ67-expressing worms (**Extended Data Fig. 2c, 10b**). However, since loss of *eps-8* and *usp-4* also extends lifespan in control polyQ19

worms (**Extended Data Fig. 2c, 10b**), we could not attribute this effect specifically to disease-related changes. In summary, given that aging hastens disease-related phenotypes and *eps-8/usp-4* knockdown delays aging, it is difficult to ascribe a specific effect of lowering USP-4/EPS-8/RAC signaling on preventing disease-related phenotypes such as shortened lifespan and motility deficits.

As suggested by Reviewer #2, we have now assessed neurodegeneration in GABAergic neurons. To this end, we collaborated with Nuria Flames' group (IBV-CSIC Valencia), which has extensive expertise in neurogenesis and neurodegeneration. While loss of motility is the most studied phenotype in polyQ67- and FUS-expressing worm models, neurodegeneration of GABAergic neurons has not been reported in these strains (Brignull et al., *J. Neurosci.*, 2006, PMID: 16855087; Murakami et al., 2012, PMID: 21949354). In contrast, neurodegeneration phenotypes in GABAergic neurons have been previously documented in worms expressing mutant *TDP-43* (Liachko et al., *J. Neurosci.*, 2010, PMID: 21123567). We have now crossed polyQ, FUS, and TDP-43 worm models, along with their respective controls, with strains expressing fluorescent markers for GABAergic neurons (PMID: 27740909). We did not detect neurodegeneration phenotypes in the GABAergic neurons of polyQ67 or mutant FUS worms, even at day 10 of adulthood (**Extended Data Fig. 4b-g**). In contrast, as previously reported, we observed a strong neurodegeneration phenotype in worms expressing mutant TDP-43 (**Fig. 2j,k and Extended Data Fig. 4a**). Notably, knockdown of either *eps-8* or *RAC* orthologs reduced GABAergic degeneration in TDP-43^{M337V} worms (**Fig. 2j,k and Extended Data Fig. 4a**). The beneficial effects of reducing EPS8/RAC signaling on disease-related neurodegeneration are further supported by analysis of cell death in human iPSC-derived neurons expressing mutant FUS (**Fig. 3d-e, 8g**).

Nevertheless, neuronal function can still be altered even in the absence of pronounced degeneration phenotypes. Given that lifespan and thrashing assays alone are insufficient to support a specific role of USP-4/EPS-8/RAC signaling in disease-related neuronal dysfunction, we have now performed two different behavioral assays (i.e. nose-touch response and chemotaxis) in polyQ and ALS models, under various conditions (*eps-8/mig-2/rac-2/usp-4* knockdown, Ub-less EPS-8 expression, loss-of-function *eps-8* mutant, DUB inhibition) and at different ages (day 1, 5, and 7).

Importantly, the disease models exhibited normal behavioral responses on day 1 but developed significant deficits in nose-touch response and chemotaxis with age. Our treatments did not affect behavioral responses in control worms at the different ages tested, but they specifically rescued or hastened the age-related decline in nose-touch response and chemotaxis observed in Q67 and ALS models (**Fig. 1b,c,e,f; Fig. 2f-i; Fig. 6c-e; Fig. 7c-g; Extended Data Fig. 2d-e, Extended Data Fig. 3d-e, Extended Data Fig. 9a-b; Extended Data Fig. 10a**). These findings provide strong evidence that USP-4/EPS-8/RAC signaling influences disease-related neuronal dysfunction in *C. elegans* during aging. The revised text now says: "Nose-touch avoidance behavior is mediated by sensory neurons located in the head of the worm.

On the first day of adulthood, polyQ67-expressing worms responded to nose touch similarly to control polyQ19 and wild-type worms (**Fig. 1b**). However, polyQ67 worms exhibited a decline in nose-touch response compared to control animals at older ages (**Fig. 1b**). Notably, knockdown of *eps-8* or *RAC* orthologs rescued this age-related functional decline in polyQ67 worms, but had no effect on aging control animals (**Fig. 1b**). PolyQ aggregation also induces neurotoxicity in chemosensory neurons, leading to impaired chemotaxis responses³¹. While polyQ67 worms exhibited normal chemotaxis towards benzaldehyde on day 1 of adulthood, they developed chemotaxis deficits with age (**Fig. 1c**). However, knockdown of *eps-8* and *RAC* orthologs mitigated this decline in polyQ67 worms without affecting chemotaxis behavior in control animals (**Fig. 1c**). Similarly, *eps-8* knockout mutation ameliorated the age-related decline in nose-touch responses and chemotaxis caused by polyQ67 expression (**Extended Data Fig. 2d-e**). (...) We observed that the expression of ubiquitin-less EPS-8 accelerates polyQ67 aggregation and disease-related behavioral changes from day 1 of adulthood (**Fig. 1d-f and Extended Data Fig. 2f**). (...) Importantly, we observed that DUB inhibitor also attenuates protein aggregation and behavioral deficits in *C. elegans* disease models (**Fig. 6a-e and Extended Data Fig. 9a-f**). (...) In addition to polyQ-expanded peptides, *usp-4* knockdown also prevented the aggregation of ALS-related mutant proteins (**Fig. 7a,b**). Consistent with this reduction in aggregation, *usp-4* knockdown rescued both nose-touch response and chemotaxis deficits in polyQ and ALS models during aging, while having no effect on these behavioral responses in control worms (**Fig. 7c-g and Extended Data Fig. 10a**)”.

2. A statistical analysis of the filter-trap experiments is necessary. The claims made by the authors are difficult to assess without statistical validation of the experimental data.

We have now quantified and performed statistical analysis on the filter trap experiments, including those presented in our initial submission as well as the new experiments conducted for this revision. We thank Reviewer #2 for raising this point, as our conclusions regarding the effects on disease-related aggregation are now further strengthened by the statistical analysis of the filter trap assays.

3. The authors claim that their RNAi/treatment does not affect the total polyQ levels. However, some experiments show trends of increasing or decreasing polyQ levels, though high variability renders these results non-significant. In particular, for Figure 1, a fourth replicate of the SDS-PAGE analysis would enhance the robustness and validity of their claims.

We have now performed additional replicate experiments for the SDS-PAGE analysis in many figures. For instance, all the panels presented in Figure 1 of our initial submission now include statistical analysis from at least four independent experiments. Since several figures have been redistributed in the revised manuscript,

here we provide a summary of all current figures containing additional independent replicates: one additional replicate for **Fig. 1a**, **Fig. 1d**, **Fig. 2a**, **Fig. 3c**, **Fig. 3h**, and **Extended Data Fig. 1g**; two additional replicates for **Fig. 6a**; three additional replicates for **Fig. 3f**, **Fig. 3g**, **Fig. 4d**, **Fig. 5a**; and four additional replicates for **Fig. 4c**.

Alongside the SDS-PAGE analyses, we have also performed the corresponding filter trap experiments for all the new replicates. These new replicate experiments, together with the statistical analysis of both filter trap and SDS-PAGE assays, strengthen our conclusion that lowering USP-4/EPS8/RAC signaling reduces the aggregation of distinct disease-related mutant proteins without decreasing their total levels.

4. The thrashing assay is a controversial and limited measure of *C. elegans* motor function. For scoring motility (body bends), how was this done? If this was done by manual counting then this approach is semiquantitative at best and should be examined with non-biased (automated) approaches. The reference (#2) in the methods refers to a review article. This is important because the range of difference is not huge, it is around 10-15 bends per 30 seconds which is not a huge difference. Also, importantly it does not appear that these experiments were done blind.

In our initial submission, the motility assays were not conducted blind, and counts were obtained manually. We have now repeated the experiments in day-5 adult polyQ and TDP-43 worms upon *EPS-8/RAC* knockdown to perform non-biased (automated) quantification (**Extended Data Fig. 2a, 3c**). In parallel, we have now performed thrashing assays with automated quantification in day-1 polyQ and TDP-43 worms, along with their corresponding control strains (**Extended Data Fig. 2a, 3c**). We confirmed that polyQ67 and mutant TDP-43 worms exhibit a statistically significant reduction in motility compared to control Q19 and wild-type TDP-43 worms at day 5, but not at day 1 (**Extended Data Fig. 2a, 3c**). Likewise, mutant FUS worms exhibited motility deficits when compared to wild-type FUS worms at day 5 or 7, but not at day 1 of adulthood (**Extended Data Fig. 9e, 10d**)

Additionally, we repeated thrashing assays in polyQ67 worms expressing mutant Ub-less EPS-8 at two different ages (day 1 and day 3) using automated quantification. Similar to the effects on nose-touch response and chemotaxis (**Fig. 1e,f**), our results confirm that Ub-less EPS-8 induces disease-related motility decline as early as day 1 of adulthood (**Extended Data Fig. 2f**). Moreover, we repeated the motility experiments in polyQ, TDP-43, and FUS models following DUB inhibition and *usp-4* knockdown, now implementing automated analysis and monitoring worms at different ages (**Extended Data Fig. 9c-f, 10c-e**). For instance, we assessed worms under *usp-4* RNAi treatment at days 1, 5, and 7 of adulthood.

Since USP-4/EPS-8/RAC downregulation and DUB inhibition restored motility in disease models to levels comparable to control worms under the same treatment

(**Extended Data Fig. 2a, 3c, 9c-f, 10c-e**), our results suggest an effect on disease-related motility deficits. However, we now explicitly acknowledge that these results are difficult to interpret due to the general benefits of USP-4/EPS-8/RAC downregulation and DUB inhibition on motility during aging (**Extended Data Fig. 2a, 3c, 9c-f, 10c-e**). In the revised manuscript, our primary conclusions regarding the specific effects of lowering USP-4/EPS-8/RAC signaling on disease-related changes are based on nose-touch response and chemotaxis experiments at different (**Fig. 1b,c,e,f; Fig. 2f-i; Fig. 6c-e; Fig. 7c-g; Extended Data Fig. 2d-e, Extended Data Fig. 3d-e, Extended Data Fig. 9a-b; Extended Data Fig. 10a**), which correlate with the changes in protein aggregation induced by these treatments.

We have now provided a more detailed explanation in the Methods section of how the motility assays were conducted, including the automated analysis. Additionally, we apologize for citing an incorrect reference in the initial submission and have now cited the appropriate sources.

The motility data presented by the authors includes some surprising results:

- Figure 1h: The data shows that day 1 neuronal 67Q worms have 40 bends per 30 seconds, which is the same as the thrashing rate observed in day 5 neuronal 67Q worms. This raises two possibilities:
- The thrashing assay may not accurately correlate with worm aging, or
- Day 5 is not a sufficiently advanced time point to detect age-related motor deficits. If neuronal 67Q worms exhibit similar movement at both day 1 and day 5, it makes it hard to argue that the treatment rescues an age-related phenotype.

We thank the Reviewer for this comment. However, it might be difficult to draw conclusions by comparing the motility data from day-1 adults in Figure 1h of our initial submission with day-5 adults from other figures. In former Figure 1h, data were obtained from polyQ67 worms outcrossed with VDL05 and VDL06 strains. VDL05 was generated by tagging endogenous EPS-8 with 3xHA in the N2 wild-type stock from SunyBiotech, while VDL06 was derived from VDL05 by mutating the ubiquitination sites of EPS-8::3xHA. Additionally, in former Figure 1h, worms were fed OP50 *E. coli*, whereas in other figures, they were fed HT115 *E. coli* for RNAi treatment. This has now been clarified in the *Motility Assay* section of the Methods: “*C. elegans* were synchronized on *E. coli* (OP50) bacteria using the egg-laying technique until L4 stage, and then randomly transferred to plates with *E. coli* (HT115) bacteria containing either control or RNAi for the rest of the experiment. For experiments with Ub-less EPS-8 mutants or DUB inhibitor experiments, L4 larvae were instead transferred to fresh plates containing *E. coli* (OP50) bacteria”.

We have now repeated the thrashing assays in polyQ67 worms expressing EPS-8::3xHA or Ub-less EPS-8::3xHA using automated quantification (**Extended Data Fig. 2f**). In addition to analyzing day-1 adult worms, we have now included day-3 adult worms for direct age-based comparisons (**Extended Data Fig. 2f**). At day 3 of

adulthood, the control Q67;EPS-8::3xHA strain exhibited a significant decline in motility compared to day-1 adults, supporting that day 3 is a sufficiently advanced time point to detect age-related motor deficits. Moreover, expression of Ub-less EPS-8 accelerates motility decline in polyQ67 worms from day 1 of adulthood, with even stronger effects observed at day 3 (**Extended Data Fig. 2f**).

Additionally, for other motility experiments involving Q67 and ALS models (FUS, TDP-43), we have now performed automated analysis at different ages. We confirmed that polyQ67, mutant FUS, and mutant TDP-43 worms do not exhibit motility differences at day 1 of adulthood compared to their respective age-matched controls. While control worms displayed an age-related decline in motility, this decline was more pronounced in disease models. Consequently, disease models exhibited a statistically significant reduction in motility compared to controls at days 5 and 7 (**Extended Data Fig. 2a, 3c, and Extended Data Fig. 9d-f, 10c-f**). These results further support that the thrashing assay correlates with aging-related motor decline.

Importantly, USP-4/EPS-8/RAC downregulation had no effect on motility at day 1 of adulthood but restored motility deficits at older ages in disease models to levels comparable to control worms under the same treatment (**Extended Data Fig. 2a, 3c, 9c-f, 10b-d**). Together, these findings suggest that USP-4/EPS-8/RAC downregulation mitigates age-related motility deficits exacerbated by disease-related protein aggregation.

- Figure 5: Treatment with a DUB inhibitor on day 5 improves the thrashing score of mutant 67Q/FUS/TDP worms to levels exceeding those of their WT counterparts. Given the prior experience with the 19Q and 67Q strains, this seems improbable. This raises concerns about whether the assay is influenced by twitching phenotypes.

As indicated above, we have now repeated the motility assays in control, Q67, FUS^{P525L}, and TDP-43^{M337V} worms treated with DUB inhibitor to perform automated quantification analysis (**Extended Data Fig. 9c-f**). In these experiments, the motility of Q67 worms treated with DUB inhibitor was not higher compared to their corresponding controls under the same treatment (**Extended Data Fig. 9c,d**). In the case of FUS models, although the motility in FUS^{P525L} worms treated with DUB inhibitor appeared to be slightly higher compared to FUS^{WT} worms under the same treatment (**Extended Data Fig. 9e**), the difference was not significant: FUS^{WT} + DUB inhibitor vs FUS^{P525L} + DUB inhibitor, *P* value = 0.1116 (two-way ANOVA with Fisher's LSD test). Likewise, the motility in TDP-43^{M337V} worms treated with DUB inhibitor was not significantly higher compared to TDP-43^{WT} under the same treatment (**Extended Data Fig. 9f**): TDP-43^{WT} + DUB inhibitor vs TDP-43^{M337V} + DUB inhibitor, *P* value = 0.1828 (two-way ANOVA with Fisher's LSD test).

- All Figures: Typically, mutant strains display paralysis phenotypes during aging, which should be evident in the assays. However, no data point in the paper shows

worms with significantly reduced movement. All data sets fall within a 10-20% range of movement, suggesting a lack of severely affected worms.

Other disease models such as worms expressing amyloid- β ($A\beta_{1-42}$ peptide) in muscle tissues display a severe paralysis phenotype (Cohen et al, *Science*, 2006, PMID: 16902091). In contrast, the disease models used in our study display a decrease in thrashing rates but do not develop severe paralysis (for instance: Murakami et al., 2012, PMID: 21949354).

Using automated analysis, we confirmed that polyQ67, mutant FUS, and mutant TDP-43 worms do not show motility differences at day 1 of adulthood compared to their respective age-matched controls. However, these disease models exhibit a statistically significant reduction in motility at days 5 and 7 compared to controls (**Extended Data Fig. 2a, 3c, and Extended Data Fig. 9d-f, 10c-e**). Importantly, they also exhibit statistically significant deficits in nose-touch and chemotaxis responses with aging (**Fig. 1b-c, 1e-f, 2f-i, 6c-e, 7c-g and Extended Data Fig. 2d-e, 3d-e, 9a-b, 10a**).

While we acknowledge that these disease models are not severely affected, they display statistically significant behavioral impairments that can be ameliorated by our treatments. Based on our experience, these strains represent a more physiologically relevant model of diseases than those exhibiting severe phenotypes and early-onset sickness.

The interventions seem to improve the motility of the WT strains (polyQ, FUS, TDP-43) thus it appears to be a general mechanism and not limited to disease proteins. Are these interventions acting as a general motility inducer? What happens in N2 worms if *eps-2* (and others) are targeted?

As discussed in Comment #1, we previously found that EPS-8/RAC signaling plays a role in both neuronal and muscular function in wild-type worms during aging. Specifically, we reported that reducing EPS-8/RAC signaling prevents actin destabilization in muscle cells of aging wild-type animals (Koyuncu et al., *Nature* 2021. PMID: 34321666). Consequently, EPS-8/RAC knockdown delays age-related muscle dysfunction and helps maintain motility during aging (Koyuncu et al., *Nature* 2021. PMID: 34321666). These findings indicate that reducing EPS-8/RAC signaling does not induce general hyperactivity in wild-type worms but instead prevents age-related motility decline (**Figure R1**).

Using automated analysis, we have now confirmed that knockdown of *eps-8* or *RAC* orthologues does not affect motility in young wild-type worms, but mitigates age-related motility deficits at older ages (please see **Extended Data Fig. 2b**). Consistently, we observed the USP-4/EPS-8/RAC downregulation ameliorates age-related motility deficits in control polyQ19, wild-type FUS and wild-type TDP-43 worms (**Extended Data Fig. 2a, 3c, 9c-f, 10c-e**). Additionally, we confirmed that polyQ67, mutant FUS, and mutant TDP-43 worms exhibit significant motility decline at day 5

and 7 compared to their respective age-matched controls, but not at day 1 (**Extended Data Fig. 2a, 3c, 9c-f, 10c-e**). Importantly, USP-4/EPS-8/RAC downregulation restored motility deficits in these disease models at older ages to levels comparable to control worms under the same treatment (**Extended Data Fig. 2a, 3c, 9c-f, 10c-e**).

Together, these findings suggest that USP-4/EPS-8/RAC downregulation can mitigate age-related motility deficits exacerbated by disease-related protein aggregation. However, we now explicitly discuss in the main text that these results are difficult to interpret due to the general beneficial effects of USP-4/EPS-8/RAC downregulation in the motility of control worms during aging (**Extended Data Fig. 2a-b, 3c, 9c-f, 10c-e**).

Since targeting *mig-2* and *rac-2* seems to have positive effects, how does this work mechanistically? Is there no redundancy?

To assess whether the RAC orthologs *mig-2* and *rac-2* have redundant effects on polyQ aggregation, we applied diluted RNAi treatments. We observed that the combination of diluted RNAi against *mig-2* and *rac-2* further decreases polyQ67 aggregation compared to diluted *rac-2* alone, suggesting that both RAC orthologs have at least partially redundant effects on polyQ aggregation (**Extended Data Fig. 1c**).

Additionally, all of the work is done using RNAi. Some additional validation using mutations (or auxin approaches) at least for *eps-8* needs to be done to confirm the findings.

We appreciate the Reviewer's concern. However, we respectfully clarify that not all experiments in our initial submission were conducted using RNAi. Our conclusions were drawn from a comprehensive set of experiments across different disease worm models, including:

1. RNAi-mediated knockdown of *eps-8*, as well as upstream (*usp-4*) and downstream (RAC orthologs) factors;
2. analyses of Ub-less EPS-8 mutants; and
3. treatments with pharmacological agents (e.g., DUB inhibitors, CytoD).

The experiments involving Ub-less EPS-8 mutants are particularly relevant, as they establish a direct link between EPS-8 ubiquitination and disease-related protein aggregation. In addition to our work in *C. elegans*, we supported our findings with experiments in human cells, using independent shRNA constructs against *EPS8*, *EPS8* overexpression, and a RAC activator.

To further strengthen our conclusions, we have now conducted additional experiments using loss-of-function *eps-8(ok539)* mutants. We observed that *eps-8(ok539)* mutants exhibit reduced polyQ67 aggregation (**Extended Data Fig. 1a**).

Furthermore, the *eps-8* knockout mutation ameliorated the age-related decline in nose-touch response and chemotaxis caused by polyQ67 expression (**Extended Data Fig. 2d-e**).

Is not the JNK orthologue *jnk-1* in *C. elegans*? Why was *rgb-1* chosen.

KGB-1 is a well-studied *C. elegans* JNK homolog (e.g., PMID: 22554143, 27864060, 34726729, 33755114). Similar to EPS-8/RAC signaling (PMID: 34321666), previous studies have shown that KGB-1 activity becomes detrimental in adulthood, leading to a shortened lifespan (PMID: 22554143, 33755114). In light of these findings, our previous work focused on the interplay between hyperactivation of EPS-8/RAC and KGB-1 signaling in the regulation of *C. elegans* lifespan (PMID: 34321666). Specifically, we demonstrated that knockdown of *kgb-1* after development mitigates the shortened lifespan phenotype observed in mutant worms expressing ubiquitin-less EPS-8 (PMID: 34321666). These results suggest that KGB-1 activity contributes to the detrimental effects of hyperactivated EPS-8/RAC signaling during aging.

Building on this, our initial submission explored the role of *kgb-1* in disease-related protein aggregation. However, we agree with the Reviewer's insightful comment that the involvement of *jnk-1*, another *C. elegans* JNK homolog, cannot be ruled out. To address this, we have now tested the effects of *jnk-1* on disease-related protein aggregation (please see **Extended Fig. 8a-d**). Similar to *kgb-1* (**Fig. 5a,b**), loss of *jnk-1* also decreased polyQ67 aggregation in day-5 adult worms and mitigated the accelerated aggregation of polyQ67 in Ub-less EPS-8 mutants at younger ages (**Extended Data Fig. 8a,b**). Besides polyQ peptides, *kgb-1* knockdown effectively prevented aggregation of FUS (R552G, PR522G) and TDP-43 (M337V) mutant variants in day-5 adult worms (**Fig. 5c,d**). Although to a lesser extent, loss of *jnk-1* also decreased aggregation of TDP-43^{M337V} and the severe FUS^{P525L} mutant variant, but it did not significantly prevent aggregation of mutant FUS^{R522G} (**Extended Data Fig. 8c,d**). These results suggest that the JNK homolog *kgb-1* has stronger effects on disease-related aggregation than *jnk-1*.

The iPSC models are useful tools, but they lack the hallmarks of aging so interpretations should be cautious.

The Reviewer is absolutely right, and we have now discussed this limitation to interpret our results using iPSC-derived motor neurons in the context of aging. The revised manuscript now says: "Similar to *C. elegans*, we found that lowering EPS8/RAC signaling reduces disease-related changes in human cell lines and iPSC-derived neurons, highlighting the evolutionary conservation of these effects. While ALS iPSC-derived motor neurons exhibit disease-related alterations, such as increased cell death⁵⁰⁻⁵², they lack hallmarks of aging^{53, 54}. This limitation arises because the reprogramming process to generate iPSCs resets cellular age to an embryonic-like state^{53, 54}. Therefore, although our results demonstrate a role for

EPS8/RAC activity in protein aggregation and neurodegeneration in human cells, they cannot provide a direct link between aging and EPS8/RAC signaling in these cellular models”.

B. Originality and significance: if not novel, please include references

Reduction in protein aggregation does not always correlate with improved disease phenotypes or progression. Some aggregation-reducing therapies have failed to yield clinically relevant results, making this a controversial target. It is therefore crucial to provide compelling evidence that demonstrates reduced neuronal degeneration, improved motor function, or increased lifespan. The study provides a valuable example of translational confirmation, extending findings from *C. elegans* to cellular models, which is a notable strength of the paper. However, given that EPS8 is a known USP-modifier, the results presented in this work are not particularly surprising; it was expected that EPS8 would reduce the aggregation phenotype. Furthermore, considering that targeting protein aggregation has become a frequently tested and somewhat redundant approach in the clinical development of therapies for neurodegenerative diseases, the findings in this study may lack a sense of novelty or excitement of a novel drug target.

PMID: 36926679, PMID: 36158183

We sincerely thank the Reviewer for their critical comments and suggestions, which have significantly improved our manuscript and strengthened our conclusions. In response to their feedback, we have now conducted nose-touch and chemotaxis experiments, providing compelling evidence demonstrating that USP-4/EPS-8/RAC signaling not only modulates protein aggregation in Huntington’s disease and ALS worm models but also subsequent neuronal deficits (i.e., decline in nose-touch response and chemotaxis) (**Fig. 1b,c,e,f; Fig. 2f-i; Fig. 6c-e; Fig. 7c-g; Extended Data Fig. 2d-e, Extended Data Fig. 3d-e, Extended Data Fig. 9a-b; Extended Data Fig. 10a**). Additionally, we have now showed that this pathway alleviates short lifespan phenotypes (**Extended Data Fig. 2c, 10b**) and neurodegeneration in worms (**Fig. 2j-h and Extended Data Fig. 4a**). Moreover, we have included new data demonstrating that EPS8 knockdown reduces not only apoptosis but also necroptosis in ALS iPSC-derived motor neurons (**Fig. 3e**).

Regarding the novelty and significance of our findings, we respectfully disagree with the Reviewer’s comment. While several clinical trials targeting amyloid- β for Alzheimer’s disease treatment—such as those involving Gantenerumab, Crenezumab, and Aamilomotide—have been unsuccessful, substantial evidence continues to support a role of amyloid- β and other aggregates (e.g., tau) in Alzheimer’s disease pathology. Along these lines, recent clinical trials targeting amyloid- β in the early stages of Alzheimer’s disease (e.g., Lecanemab, Donanemab, Aducanumab) have shown promising results, suggesting that reducing protein aggregation remains

a viable therapeutic strategy (PMID: 37784171). However, it is important to emphasize that our study focuses on Huntington's disease and ALS, not Alzheimer's disease.

We also thank the Reviewer for providing two recent review articles on ubiquitination and proteasomal degradation of mutant HTT and TDP-43. We have now cited these reviews in our manuscript. However, it is important to note that these review articles primarily focus on the direct ubiquitination and degradation of mutant HTT and TDP-43. In this regard, our previous work has contributed to identifying E3 ubiquitin ligases that promote mutant HTT degradation (e.g., PMID: 30038412), and this work is appropriately cited in the Huntington's disease review. In contrast, our present study takes a different approach. Instead of focusing on direct degradation mechanisms of disease-related mutant proteins, we investigated EPS8, a regulatory protein that becomes less ubiquitinated and degraded with age. Given the association between EPS8 accumulation and aging, we explored whether reducing its levels could be a broader strategy to prevent age-related diseases.

In response to the Reviewer's comment: "*However, given that EPS8 is a known USP-modifier, the results presented in this work are not particularly surprising; it was expected that EPS8 would reduce the aggregation phenotype*", we sincerely apologize if we misunderstood the concern. To our knowledge, no prior study has demonstrated a role for EPS8 as a UPS-modifier. Our findings indicate that EPS8 knockdown decreases aggregation of disease-related proteins without affecting their degradation. To clarify this point, we have now included data showing that *EPS8* knockdown does not alter global proteasome or autophagy activity in either worm or human disease models (**Extended Data Fig. 6a-f**).

Finally, we recognize that our initial submission may not have clearly conveyed the novelty and significance of our findings. We have revised the manuscript to clarify these aspects. For instance, we have rewritten the introduction to better highlight the novelty of our approach. The revised text now says: "With age, animals undergo alterations in proteolytic systems, including the ubiquitin-proteasome system^{2, 11-16}. Since aggregation-prone proteins like mutant HTT and TDP-43 can be ubiquitinated, extensive research focuses on how ubiquitinating and deubiquitinating enzymes directly influence their proteasomal degradation^{17, 18}. In this study, we explored a different approach to define common mechanisms that prevent pathological aggregation across distinct disorders. Beyond directly influencing the levels of disease-related proteins, age-related downregulation of targeted degradation also leads to the accumulation of regulatory proteins, affecting pathways required for normal cell function^{2, 11-16}. An intriguing question is whether the accumulation of regulatory proteins that escape proteasomal clearance contributes to disease-related protein aggregation during aging".

C. Data & methodology: validity of approach, quality of data, quality of presentation

There is room for improvement. Details are lacking about the thrashing assay and it appears to be a semiquantitative assay that may be prone to experimenter bias, and the experiments were not blinded. This is important since there are not large differences between the experimental groups. Additional assays looking at neurodegeneration would strengthen the manuscript.

We appreciate the Reviewer's constructive feedback and have taken steps to strengthen our methodology and conclusions. In our initial submission, motility assays were not conducted blind, and body bends were counted manually. As discussed above, we have now repeated the motility assays including different ages and using non-biased automated quantification (**Extended Data Fig. 2a-b, 2f, 3b-c, 9c-f, 10b-d**). These new experiments confirm that polyQ67, mutant FUS, and mutant TDP-43 worms exhibit statistically significant deficits in motility compared to their respective controls at older ages, while no deficits are observed at day 1. Additionally, automated thrashing analysis demonstrates that USP-4/EPS-8/RAC downregulation and DUB inhibition restore motility deficits in disease models to levels comparable to control worms under the same treatment.

D. Appropriate use of statistics and treatment of uncertainties.

There is room for improvement, especially in considering the limitations of the thrashing assay. Also, the protein work (filter-trap assays) lack statistical validation.

We have now applied the correct statistical tests to all the figures from our initial submission, as well as the new experiments performed during the revision. Importantly, these adjustments did not alter the conclusions of our manuscript. As discussed above, we have also quantified and performed statistical analysis on the filter trap experiments.

E. Conclusions: robustness, validity, reliability.

The conclusions are not fully supported by the approaches, details in section A.

We acknowledge the Reviewer's concern and have comprehensively addressed these points throughout our response letter. In our revised manuscript, we have conducted multiple experiments and included new data to further support our conclusions. These new experiments strengthen the validity and robustness of our study, reinforcing our conclusion regarding the role of USP-4/EPS-8/RAC downregulation in mitigating disease-related protein aggregation and subsequent neuronal dysfunction.

F. Suggested improvements: experiments, data for possible revision
Provide supplementary data on thrashing results at various ages to clearly demonstrate how aging affects the outcomes.

Conduct thrashing experiments (or something better) at older ages where mutant worms

Include additional, unbiased and blinded measures of worm health. The thrashing assay used in this study is not robust regarding data reproducibility expected by journals and current research standards.

As we have discussed thoroughly this response letter, we have now addressed this concern in the revised manuscript, strengthening our conclusions. For instance, we have repeated the motility assays at different ages using non-biased automated quantification, conducted lifespan experiments, neurodegeneration analysis, and performed behavioral analyses to assess neuronal function.

G. References: appropriate credit to previous work?

Seems OK, but a little narrow and does not touch upon other work using *C. elegans* for HD and ALS.

We have now expanded the number of references from 45 in our initial submission to 85. Among the newly added references, we have discussed other studies that used HD and ALS *C. elegans* models. For instance, the revised manuscript now includes the following text: “PolyQ aggregation also induces neurotoxicity in chemosensory neurons, leading to impaired chemotaxis responses³¹”. (...) “*C. elegans* models of Huntington’s and ALS have proven to be invaluable tools for identifying modifiers of disease-related protein aggregation and its physiological consequences, including components of the proteostasis network and environmental interventions^{9, 16, 24-27, 33, 42-49}. (...) In *C. elegans*, knockdown of *anc-1*, which encodes a protein involved in actin binding and cytoskeleton organization, alters the expression of transcription factors and E3 ubiquitin ligases, leading to polyQ aggregation⁶²”.

H. Clarity and context: lucidity of abstract/summary, appropriateness of abstract, introduction and conclusions. The manuscript is clear and logically presented. There seem to be some grammar syntax errors, so double-checking that would be a good idea.

We thank the Reviewer for their positive comments about the structure of our paper and apologize for any grammar syntax error in our initial submission. We have double checked the revised manuscript multiple times to correct these errors.

Reviewer #3:

Remarks to the Author:

Previously the Vilchez lab discovered that the upregulation of EPS-8 hyperactivates RAC in muscle and neurons, and alters actin cytoskeleton and the activity of the protein kinase JNK (Koyuncu et al., Nature 2021). This pathway was also reported to regulate lifespan by modulating actin polymerization. In this study, which is a follow up of the aforementioned work, they focus on the question of whether the EPS-8/RAC pathway also controls toxic protein aggregation in worms and mammalian cultured cells.

In brief, they found that the knockdown of *eps-8*, *mig-2* or of *rac-2* by RNAi reduces the amounts of SDS-resistant aggregates of polyQ67-YFP that is expressed in neurons as well as of polyQ40-YFP which is expressed in muscles and of intestinal polyQ44-YFP (Fig. 1). The reduced aggregation levels were associated with enhanced thrashing rates indicative of mitigated proteotoxicity. They crossed worms that express polyQ67-YFP in their neurons with animals that bear 3 mutations in the sequence of *eps-8*. These substitutions of lysine residues prevent the ubiquitination of EPS-8 and thus, stabilize this protein. The stabilization of EPS-8 leads to reduced thrashing rate. Similarly, the knockdown of *eps-8*, *mig-2* or of *rac-2* mitigated the toxicity of two other neurodegeneration-linked, mutated aggregative proteins; FUS and TDP-43 (Fig. 2). Next, they tested whether the EPS-8/RAC axis also regulates proteostasis in mammalian cells. Using HEK293 they discovered that the knockdown of *eps-8* by shRNA leads to reduced aggregation of abnormally long polyQ stretches and of mutated FUS and TDP-43. It also reduced the level of apoptosis of motor neurons (Fig. 3D).

To test whether the modulation of protein aggregation by the EPS-8/RAC axis is associated with its roles in the formation and stabilization of actin filaments they used cytochalasin D. Their results show that the inhibition of actin polymerization and the knockdown of the kinase, KGB-1, reduce the aggregation of several aggregation-prone proteins (Fig. 4). Similar protection from proteotoxicity was seen when worms expressing the aggregative proteins were treated with a DUB inhibitor (Fig. 5). They also found that the deubiquitinase USP-4 regulates the levels of EPS-8 and subsequently, the rate of protein aggregation and level of proteotoxicity in worms (Fig. 6) and in human cells (Fig. 7).

These results culminate to illustrate a proteostasis regulating roles of the EPS-8/RAC pathway that link the integrity of actin filaments with aging-associated increased levels of EPS-8 which result from high levels of de-ubiquitination of this protein.

This study is interesting, timely and expand the current knowledge about the EPS-8/RAC pathway and the integrity of the proteome. Yet, several issues should be addressed to enhance clarity and substantiate some aspects of this manuscript.

Major comments:

1. In this work, the authors heavily rely on the “filter trap” assay to monitor protein aggregation. While this technique is widely used it only provides an assessment of SDS-resistant aggregates (probably large aggregates). Since oligomers have been shown to be the most toxic species (Silveira et al., Nature 2005 Vol. 437 Issue 7056 Pages 257-61; Shankar et al., Nat Med. 2008 Aug;14(8):837-42), it is necessary to test whether the knockdown of *eps-8* modulates the levels of these conformers – or perhaps the reduced levels of SDS-resistant aggregates are associated with increase in oligomers. This can be done by ultra-centrifugation; however, I think that it would be easier and sufficient to use native gels that were optimized by the Nollen lab to follow different conformers of polyQ-YFP stretches in nematodes (Holmberg and Nollen, Methods Mol Biol 2013 Vol. 1017 Pages 193-9).

We appreciate the Reviewer's insightful suggestion regarding the importance of analyzing oligomers, as they can also contribute to disease-related changes. While our manuscript primarily focuses on SDS-resistant aggregates, we acknowledge the relevance of studying oligomers as well.

We are grateful to the Reviewer for recommending the methodology described in Holmberg & Nollen (Methods Mol Biol 2013, PMID: 23719917). This Methods paper provides a detailed explanation of the approach used in Figure 2 of a previous study from the same group (PMID: 20723760). Although we could not find more studies using this native gel electrophoresis method followed by fluorescent visualization of Q40::YFP in the gel, we agree that it represents a valuable technique for analyzing oligomers. In the original study (PMID: 20723760), the authors applied native gel electrophoresis to detect polyQ oligomers in worms expressing polyQ40 repeats in muscle cells.

However, using this approach, we were unable to detect oligomers in worms expressing Q67::YFP in neurons. The methodology paper also suggests that this technique could be adapted for cultured cells (PMID: 23719917). Accordingly, we tested native gel electrophoresis in HEK293 cells expressing polyQ repeats with a fluorescent tag. In this human cell model, we successfully detected soluble polyQ assemblies (**Extended Data Fig. 5**). However, knockdown of EPS8 did not lead to a reduction in oligomer levels (**Extended Data Fig. 5**). The revised manuscript now says: “In addition to high-molecular-weight insoluble aggregates, the accumulation of small, soluble assemblies of pathological proteins also contributes to disease-related changes^{35, 36}. Using native gel electrophoresis, we did not detect changes in the levels of polyQ-expanded soluble oligomers upon EPS8 knockdown (**Extended Data Fig. 5**), indicating that this pathway specifically influences the assembly of insoluble aggregates”.

To further validate our filter trap results regarding the effect of *eps-8* knockdown on polyQ aggregation, we have now used a western blot approach capable of detecting both polyQ monomers and SDS-insoluble polyQ aggregates retained at the

top of the gel (Nollen et al, PNAS 2004, PMID: 15084750). Using this method, we confirmed that *eps-8* knockdown reduces insoluble polyQ67 levels in worms (**Extended Data Fig. 1b**).

2. To test the relevance of their findings to mammals, the authors use HEK293 cells (Fig. 3, except for 3d). Cytoskeletal filaments were reported to be involved in the shuttling and deposition of aggregation-prone proteins, for instance in the formation of aggresomes (Johnston et al., J Cell Biol. 1998 Dec 28;143(7):1883-98). Since the cytoskeleton is expected to be affected by the EPS8/RAC pathway, it is important to test how knocking down the activity of this pathway affects the distribution of toxic aggregates in these cells. Immunofluorescence can be used to address this.

This is particularly important as it seems that the HEK293 cells were transiently transfected (as explained at the methods section). This method generates very variable cell populations, and some cells may express very high levels of the aggregative proteins.

We thank the Reviewer for suggesting this important experiment. We have now tested whether EPS8 knockdown affects the intracellular distribution of mutant HTT in human cells. However, we did not observe changes in aggregate distribution following EPS8 knockdown (please see **Extended Data Fig. 12**).

3. In figure 3e, the authors control for the over-expression of *eps-8* but no such control appears in figure 3f. This is an important control as the usage of strong viral promoters (CMV promoter?) could result in the depletion of transcription factors and in the generation of artifacts. In addition, while they state that the expression of TDP-43-A382T was driven by the pLVX-Puro-TDP-43-A382T plasmid, I could not find in the methods section, how TDP-43-M337V, which is shown in figure 3f, was expressed.

The Reviewer is absolutely right that confirming EPS8 overexpression in Fig. 3f (now updated as **Fig. 3g**) is crucial to support our conclusions. We have now verified overexpression of EPS8 in these experiments and updated the figure accordingly.

Regarding mutant TDP-43 expression, we did not use the TDP-43-M337V variant in human cell experiments. In all experiments assessing mutant TDP-43 aggregation in human cells, we tested the ALS-related TDP-43-A382T mutant variant. Unfortunately, the figure was mistakenly labeled as M337V instead of A382T. We sincerely apologize for this oversight and thank the Reviewer for bringing it to our attention. We have now corrected this labeling error.

4. To test whether the knockdown of *eps-8* ameliorates neurodegeneration they derived motor neurons from iPSCs that express mutated FUS (P525L) and concluded that: "However, knockdown of EPS8 ameliorated neurodegeneration in these cells" (page 7 and figure 3d). This statement is based on a single assay in which they followed the levels of cleaved caspase 3, to compare the rate of apoptosis. I do not

think that the results presented at figure 3d are sufficient to support their claim. Motor neurons die in ALS patients by additional mechanisms including necroptosis (Neuron. 2014 Mar 5;81(5):1001-1008) and autophagy dependent cell death (see for review: Cell Death Discov. 2024 Jun 19;10(1):291). Therefore, to support their claim they should at least test whether the knockdown of eps8 induces necroptosis (Methods Mol Biol. 2021;2255:119-134).

Following the Reviewer's suggestion, we have now tested necroptotic cell death. Our results indicate that knockdown of EPS8 reduces necroptosis in ALS iPSC-derived motor neurons (please see **Fig. 3e**). We are grateful to the Reviewer for suggesting this experiment, as it strengthens our conclusion regarding the protective effects of lowering EPS8 in preventing neurodegeneration. The revised text now says: "However, EPS8 knockdown ameliorated apoptosis in these cells (**Fig. 3d**). Besides apoptosis, other mechanisms, such as necroptosis, contribute to motoneuronal death in ALS³⁸. Notably, we observed that EPS8 knockdown also reduces phosphorylation and subsequent activation of RIP kinase in ALS motor neurons (**Fig. 3e**), a marker of necroptotic cell death³⁹. Together, these results suggest that lowering EPS8 levels mitigates ALS-related neurodegeneration".

5. What are the effects of modulating the EPS8/RAC pathway on protein degradation pathways in the context of constant proteotoxic insult? It has been already shown that the deubiquitinase USP11 controls autophagy (Basic et al., ref. 37). Does the knockdown of eps-8 affect general activity of the ubiquitin proteasome system and/or autophagy when the worm is challenged by a proteotoxic protein?

The Reviewer is absolutely right that testing the effects of EPS8 on global proteasome and autophagy activity is critical for our conclusions. As suggested by the Reviewer, we have now performed these experiments in *C. elegans* expressing polyQ67. We found that *eps-8* knockdown does not induce proteasome activity or autophagy in these worms (please see **Extended Data Fig. 6a,b**). In addition, we have tested proteasome activity and autophagy in both naïve and Q100-HTT HEK293 human cells. Similar to worms, knockdown of EPS8 did not affect the general activity of these major proteolytic systems in human cells (**Extended Data Fig. 6c-f**). The text now says: "Our findings suggest that lowering EPS8 levels attenuates pathological protein aggregation without decreasing the levels of disease-related proteins. Consistent with this, knockdown of EPS8 in both worms and human cells did not enhance proteasome activity or LC3 lipidation, a marker of autophagosomes (**Extended Data Fig. 6a-f**)".

6. I think that it would enhance the manuscript if they further elaborate in the discussion on the known roles of the cytoskeleton in the maintenance of proteostasis and explain how actin filaments are related. For instance, microtubule have been reported to be critical for the formation of aggresomes and vimentin was shown to cage these

structures (Johnston and Kopito J Cell Biol. 1998 Dec 28;143(7):1883-98). Ataxin 2, a protein which is involved in the onset of ALS, is involved in the regulation of cytoskeleton stability (Del Castillo et al., iScience 2021 Dec 3;25(1):103536). In addition, *anc-1* which encodes a protein that is important for actin binding and cytoskeleton organization has been found to regulate proteostasis in worms and to modulate proteasome activity (Levine et al., Aging Cell 2019 Dec;18(6):e13047).

We thank the Reviewer for providing these references. We have now further elaborated on the links between cytoskeleton and disease-related aggregation, incorporating the references suggested by Reviewer #3, as well as other relevant studies. Additionally, as recommended by both Reviewers #1 and #3, we have speculated on potential mechanisms by which excessive actin polymerization may influence protein aggregation. The updated Discussion section now says: “While HTT and ALS-related proteins, including the polyQ-containing protein ataxin-2, regulate actin dynamics^{59, 60}, previous studies have indicated that actin filaments and actin-binding factors may also influence pathological protein aggregation^{55, 61}. For instance, distinct familial ALS cases that exhibit wild-type TDP-43 aggregates are associated with mutations in actin cytoskeleton regulators such as profilin 1⁵⁵. We speculate that age-related destabilization of actin filaments may affect protein aggregation by impairing essential cellular processes, thereby reducing the cellular capacity to prevent protein aggregation. In *C. elegans*, knockdown of *anc-1*, which encodes a protein involved in actin binding and cytoskeleton organization, alters the expression of transcription factors and E3 ubiquitin ligases, leading to polyQ aggregation⁶².

Importantly, mutant HTT aggregates can co-localize with actin filaments⁵⁷. Although we did not observe changes in the intracellular distribution of mutant HTT aggregates following EPS8 knockdown in human cells (**Extended Data Fig. 12**), we cannot exclude the possibility that the actin cytoskeleton directly influences aggregation through its interaction with disease-related proteins. For instance, redistribution of the intermediate protein vimentin contributes to the assembly of aggresomes containing cystic fibrosis transmembrane conductance regulator (CFTR), whereas disruption of microtubules blocks aggresome formation⁶³. In previous work, we observed that age-related changes in actin filaments lead to aggregation of actin protein itself¹⁴. This raises the intriguing possibility that actin aggregates may act as a niche for the accumulation of disease-related proteins. Alternatively, actin aggregates could sequester molecular chaperones and other components of the proteostasis network, leading to its collapse and subsequent aggregation of pathological proteins”.

Minor comments:

7. While they state that the expression of TDP-43-A382T was driven by the pLVX-Puro-TDP-43-A382T plasmid, I could not find in the methods section, how TDP-43-M337V, which is shown in figure 3f, was expressed.

As noted in point #3, the Figure was mistakenly labeled as TDP-43-M337V instead of TDP-43-A382T. We have now corrected this labeling error.

In all experiments assessing mutant TDP-43 aggregation in human cells, we tested the ALS-related TDP-43-A382T variant. We did not express TDP-43-M337V in human cells. Once again, we sincerely apologize for this mistake and thank the Reviewer for identifying the issue.

8. At figure 3g they show that the activation of RAC by a “RAC activator” enhances the aggregation of polyQ100HTT. To assess the specificity of this treatment I wondered what activator was used. Surprisingly, the only information that they provide at the method section is: “For RAC activator experiments, the cells were treated with 2 units/ml RAC activator for 6 hours”. This is unacceptable, as the description MUST be sufficient for other scientists to repeat the reported experiments. Please specify, where the “RAC activator” was obtained this compound from. Vendor and catalogue number of this activator must be mentioned.

We sincerely apologize for not including this information in our initial submission and thank the Reviewer for bringing it to our attention. We have now provided the vendor and catalog number for the RAC activator. Upon reviewing other reagents, we also realized that information regarding the vendor and catalog number for cytochalasin D and the catalog number for the DUB inhibitor was missing. This essential information has now been added to the Methods section. Once again, we apologize for this oversight in our initial submission and appreciate the Reviewer’s help.

Reviewer #1:

Remarks to the Author:

Koyuncu et al have responded thoughtfully and comprehensively to all my comments, questions and criticisms. The resulting revised manuscript is now more insightful and accurate, and the core findings and conclusions are more compelling. Specifically, the authors have performed a high number of additional experiments to measure the effects of EPS-8/RAC/USP-4 on neuronal function in different models of protein aggregation/toxicity. In addition, they have shown that these effects are also relevant to muscle tissues and have revised the statistical tests used in several figures as appropriate. Lastly, the authors have expanded the discussion to provide possible routes by which excessive actin polymerization/JNK may influence protein aggregation. The work is original and important and I recommend that it is accepted for publication in Nature Aging.

We sincerely thank Reviewer #1 for their thoughtful and constructive feedback throughout the review process.

Reviewer #2:

Remarks to the Author:

This is an improved manuscript. My major concern last time was the lack of unbiased experimental research methods. Last time, the question of whether worm behavioral experiments were blinded to the investigator was raised. In this regard, I am pleased that they introduced unbiased measures of motility/thrashing. However, they add the nose-touch assay here, which seems to have been conducted without blinding (Fig. 1b). This raises a concern, given that the effects are small (better at day 7). The assay demonstrates stronger effects in later experiments (Fig. 2f, h); however, conducting these blindly would greatly enhance confidence in the results. The fact that these experiments were not blinded should be explicitly stated, particularly for assays where the experimenter actively participates, such as touching the animal versus using automated recording approaches. Not having all the worm experiments conducted blind to strain/genotype would be a significant improvement. Personally, I believe it should be mandatory for modern publishing standards. The introduction of scoring for neurodegeneration is good. The results for the TDP-43 strains are convincing. The lack of neurodegeneration phenotypes in the Q67 strain is surprising. I doubt that observation will stand up to scrutiny, but that may be clarified by future research.

We thank Reviewer #2 for their feedback. As noted in the Reporting Summary of our previous submission, the nose-touch assay was not performed blind. In

response to the reviewer's comment, we have now also explicitly stated this in the Methods section with the sentence: "Data collection and analysis were not performed blind to the conditions of the experiments."

Reviewer #3:

Remarks to the Author:

I have read the revised manuscript as well as the rebuttal letter and found that the authors have comprehensively addressed the critique. They tested the physiological relevance of the knockdown of USP-4/EP5-8/RAC signaling using two additional assays (namely, nose-touch response and chemotaxis), conducted control experiments in worms and cells and corrected their statistical analyses. The text has been also corrected and expanded and missing details in the methods section have been added (i.e catalogue numbers). In sum, I think that in the current version of the manuscript, the results are more comprehensive, the conclusions are properly supported and the text much clearer. Therefore, I recommend acceptance of the current version of the manuscript.

We thank Reviewer #3 for for their thoughtful and constructive feedback throughout the review process.